# Importance of reactive halogens in the tropical marine atmosphere: A regional modelling study using WRF-Chem

Alba Badia[1,a], Claire E. Reeves[1], Alex R. Baker[1], Alfonso Saiz-Lopez[2], Rainer Volkamer[3,4], Theodore K. Koenig[3,4], Eric C. Apel[5], Rebecca S. Hornbrook[5], Lucy J. Carpenter[6], Stephen J. Andrews[6], Tomás Sherwen[6,7], and Roland von Glasow[1,*]

[1]Centre for Ocean and Atmospheric Sciences, School of Environmental Sciences, University of East Anglia, Norwich, United Kingdom
[2]Department of Atmospheric Chemistry and Climate, Institute of Physical Chemistry Rocasolano, CSIC, Madrid, Spain
[3]Department of Chemistry, University of Colorado, Boulder, CO, USA
[4]Cooperative Institute for Research in Environmental Sciences (CIRES), University of Colorado, Boulder, CO, USA
[5]Earth System Laboratory, Atmospheric Chemistry Division, National Center for Atmospheric Research (NCAR), Boulder, CO, USA
[6]Wolfson Atmospheric Chemistry Laboratories (WACL), Department of Chemistry, University of York, York, United Kingdom
[7]National Centre for Atmospheric Science (NCAS), Department of Chemistry, University of York, York, United Kingdom
[*]deceased, 6th September 2015
[a]now at: Department of Atmospheric Chemistry and Climate, Institute of Physical Chemistry Rocasolano, CSIC, Madrid, Spain

Correspondence: a.badia-moragas@uea.ac.uk

**Abstract.**

This study investigates the impact of reactive halogen species (RHS, containing chlorine (Cl), bromine (Br) or iodine (I)) on atmospheric chemistry in the tropical troposphere and explores the sensitivity to uncertainties in the fluxes of RHS to the atmosphere and their chemical processing. To do this the regional chemistry transport model WRF-Chem has been extended to include Br, I as well as Cl chemistry for the first time, including heterogeneous recycling reactions involving sea-salt aerosol and other particles, reactions of Br and Cl with volatile organic compounds (VOCs), along with oceanic emissions of halocarbons, VOCs and inorganic iodine. The study focuses on the tropical East Pacific using field observations from the TORERO campaign (January-February 2012) to evaluate the model performance.

Including all the new processes, the model does a reasonable job reproducing the observed mixing ratios of BrO and IO, albeit with some discrepancies, some of which can be attributed to difficulties in the model's ability to reproduce the observed halocarbons. This is somewhat expected given the large uncertainties in the air-sea fluxes of the halocarbons in a region where there are few observations of their seawater concentrations.

We see a considerable impact on the inorganic bromine ($Br_y$) partitioning when heterogeneous chemistry is included, with a greater proportion of the $Br_y$ in active forms such as BrO, HOBr and dihalogens. Including debromination of sea-salt increases BrO slightly throughout the free troposphere, but in the tropical marine boundary layer, where the sea-salt particles are plentiful and relatively acidic, debromination leads to overestimation of the observed BrO. However, it should be noted that the modelled BrO was extremely sensitive to the inclusion of reactions between Br and the oxygenated VOCs (OVOCs), which convert Br to

HBr, a far less reactive form of $Br_y$. Excluding these reactions leads to modelled BrO mixing ratios greater than observed. The reactions between Br and aldehydes were found to be particularly important, despite the model underestimating the amount of aldehydes observed in the atmosphere. There are only small changes to the inorganic iodine ($I_y$) partitioning and IO when the heterogeneous reactions, primarily on sea-salt, are included.

Our model results show that tropospheric $O_x$ loss due to halogens ranges between 25-60%. Uncertainties in the heterogeneous chemistry accounted for a small proportion of this range (25% to 31%). This range in good agreement with other estimates from state-of-the-art atmospheric chemistry models. The upper bound is found when reactions between Br and Cl with VOCs are not included and, consequently, $O_x$ loss by $BrO_x$, $ClO_x$ and $IO_x$ cycles is high (60%). With the inclusion of halogens in the troposphere $O_3$ is reduced by 7 ppbv on average. However, when reactions between Br and Cl with VOCs are not included $O_3$ is much lower than observed. Therefore, the tropospheric $O_x$ budget is highly sensitive to the inclusion of halogen reactions with VOCs and to the uncertainties in current understanding of these reactions and the abundance of VOCs in the remote marine atmosphere.

## 1   Introduction

RHS cause ozone ($O_3$) destruction, change the $HO_x$ ($HO_2$ + OH) and $NO_x$ ($NO_2$ + NO) partitioning, affect the oxidation of VOCs and mercury, and take part in new particle formation (Chameides and Davis, 1980; von Glasow et al., 2004; Saiz-Lopez and von Glasow, 2012). Moreover, reactive chlorine reduces the lifetime of methane ($CH_4$). Halogen species are known to play an important role in the oxidising capacity of the troposphere. The atmospheric oxidation capacity is to a large extent determined by budgets of the hydroxyl radical (OH) and $O_3$; globally most tropospheric OH is found in the tropics (Bloss et al., 2005). Therefore a quantitative understanding of the composition and chemistry of the tropical marine atmosphere is essential to examine the atmospheric oxidative capacity and climate forcing.

In the troposphere, reactive halogen species catalyse ozone destruction cycles:

$$O_3 + X \quad \rightarrow \quad XO + O_2 \tag{R1}$$

$$HO_2 + XO \quad \rightarrow \quad HOX + O_2 \tag{R2}$$

$$HOX + hv \quad \rightarrow \quad OH + X \tag{R3}$$

where X= Cl, Br, I.

In the past, tropospheric halogen chemistry has been studied using a number of box models and 1D models (Sander and Crutzen, 1996; von Glasow et al., 2002a; Saiz-Lopez et al., 2006; Simpson et al., 2015; Lowe et al., 2009; Sommariva and von Glasow, 2012; Surl et al., 2015). Currently, there are several global models that have been used to study tropospheric halogens (Hossaini et al., 2010; Ordóñez et al., 2012; Saiz-Lopez et al., 2012a; Fernandez et al., 2014; Saiz-Lopez et al., 2015; Sherwen et al., 2016b; Schmidt et al., 2016). Numerical models predict that reactive halogen compounds account for 30% of $O_3$ destruction in the MBL (von Glasow et al., 2002b, 2004; Saiz-Lopez et al., 2015; Sherwen et al., 2016b) and 5-20%

globally (Yang et al., 2005; Saiz-Lopez et al., 2015, 2012a; Sherwen et al., 2016b). Up to 34% of $O_3$ loss is calculated to be due to I and Br combined in the tropical East Pacific (Wang et al., 2015).

However, there are only a few regional models that have studied tropospheric halogens. Chlorine chemistry was implemented into the WRF-Chem model (Lowe et al., 2015; Li et al., 2016) and into the CMAQ model (Sarwar et al., 2014) to study the formation of nitryl chloride ($ClNO_2$) from the uptake of dinitrogen pentoxide ($N_2O_5$) on aerosols containing chloride. Moreover, bromine and iodine chemistry was implemented in CMAQ in Gantt et al. (2017) and Sarwar et al. (2015), where the impact of iodide-mediated $O_3$ deposition on surface ozone concentrations was studied, and in the recent work of Muñiz-Unamunzaga et al. (2018), that concluded that oceanic halogens and dimethyl sulfide (DMS) emissions need to be included into the regional models to accurately reproduce the air quality in coastal cities.

Oceanic emissions provide a source of Very Short Lived Halocarbons (VSLH) to the atmosphere, defined as trace gases with chemical lifetimes generally under six months, mainly in the form of bromoform ($CHBr_3$), dibromomethane ($CH_2Br_2$) and methyl iodide ($CH_3I$). Once in the atmosphere, VSLH (and their degradation products) can ascend into the lower stratosphere (LS) where they can contribute to the $Br_y$ and lead to ozone depletion. Several emissions inventories for the VSLH have been evaluated at a global scale (Bell et al., 2002; Ziska et al., 2013; Ordóñez et al., 2012; Hossaini et al., 2013; Lennartz et al., 2015; Wales et al., 2018). Recent measurements constrain the stratospheric injection of bromine from VSLH as $\sim$5pptv $Br_y$ (Wales et al., 2018), confirming recent WMO estimates. About 40-50% of the bromine (2.1-2.6 pptv $Br_y$) is injected into stratosphere as product gases (Koenig et al., 2017; Wales et al., 2018). Lennartz et al. (2015) presents a comparison of two simulations using the chemistry climate model EMAC. The first simulation computes the oceanic emissions online, mainly driven by the surface water concentrations and modelled meteorological variables, and the second uses prescribed emissions. These results reveal that calculating the air-sea fluxes online leads, in most cases, to more accurate atmospheric mixing ratios in the model in comparison with the simulation using prescribed emissions. Emissions of inorganic iodine compounds (HOI and $I_2$) have been recognised as a significant source required to reproduce iodine oxide (IO) measurements over the open ocean (Mahajan et al., 2012; Carpenter et al., 2013) and have been included in some global models (Saiz-Lopez et al., 2014; Sherwen et al., 2016b).

There are indications that the chemistry of reactive halogens and oxygenated VOCs (OVOCs) in the tropics are inter-related. Model calculations suggest aldehydes are an important sink for bromine atoms and hence compete with the formation of BrO (Br + $O_3 \rightarrow$ BrO). This illustrates a link between the cycles of halogens and OVOCs in the marine atmosphere (Sommariva and von Glasow, 2012; Toyota et al., 2004).

Recent studies have highlighted the key role that heterogeneous chemistry plays in explaining observations of BrO and IO abundances in the tropical troposphere. Cycling of Br and I through HOBr, $BrNO_3$, HOI and $INO_3$ is very slow in the gas-phase, making it necessary to include heterogeneous reactions involving reactive halogen species to reproduce observed BrO and IO abundances (von Glasow et al., 2004; Saiz-Lopez et al., 2015; Sherwen et al., 2016a).

Another source of reactive inorganic bromine in the troposphere is the release of bromide ($Br^-$) from sea-salt aerosols into the gas-phase. This is known as debromination and occurs through the uptake of a gaseous species in sea-salt and the subsequent reaction with $Br^-$. Debromination has been included as a source of gas-phase bromine in several atmospheric models (Yang et al., 2005; Parrella et al., 2012; Ordóñez et al., 2012; Schmidt et al., 2016; Long et al., 2014). However, this

process is poorly understood and its inclusion into the models can cause inconsistent high levels of bromine species (Schmidt et al., 2016).

Halogen chemistry in atmospheric models remain largely untested due to lack of field observations of halogen species. However, during the last few years there have been four campaigns that provided vertically resolved measurements of halogen radicals: the Tropical Ocean tRoposphere Exchange of Reactive halogen species and Oxygenated VOC (TORERO; Volkamer et al., 2015; Wang et al., 2015; Dix et al., 2016), the CONvective TRansport of Active Species in the Tropics (CONTRAST; Pan et al., 2017; Koenig et al., 2017), the Coordinated Airborne Studies in the Tropics (CAST; Harris et al., 2017) and Airborne Tropical TRopopause EXperiment (ATTREX; Jensen et al., 2017).

The main objective of this study is to investigate the atmospheric chemistry in the tropical East Pacific with a focus on reactive halogens using the Weather Research and Forecasting model coupled with Chemistry (WRF-Chem; Grell et al., 2005) and field data from the TORERO campaign (Volkamer et al., 2015; Wang et al., 2015). Our reaction mechanism in WRF-Chem is based on the MOZART-4 mechanism (Emmons et al., 2010; Knote et al., 2014) and has been extended to include halogen chemistry. Heterogeneous recycling reactions involving halogens have been included into the model, along with oceanic emissions of relevant VOCs and halocarbons. The observational data is described in Sec. 2. Model developments are described in Sec. 3. The model setup and the description of different sensitivity runs are in Sec. 4. The results of the model performance are discussed in Sec. 5. The last section summarizes the conclusions of this work.

## 2 Observational data

The TORERO campaign (Volkamer et al., 2015; Wang et al., 2015), from 15 January to 1 March 2012, was used to evaluate the model. Data on halocarbons are available from the TORERO ship cruise (Andrews et al., 2015) and flights of the NSF/NCAR GV aircraft, whilst observations of $O_3$, BrO, IO and OVOCs are available from the flights. The TORERO cruise aboard the NOAA RV Ka'imimoana (KA-12-01) took place from Honolulu, HI, to Puntarenas, Costa Rica, between 27 January and 1 March 2012. Air samples from the TORERO ship cruise were taken from a 10m bow mast and surface water samples were taken from the underway supply. Halocarbons in air and water phases were measured using two automated on- line GC-MS systems (Andrews et al., 2015) and calibrated using NOAA standard SX-3570. Ozone was measured by UV absorption (Coburn et al., 2014), OVOCs by the Trace Organic Gas Analyzer (TOGA) (Apel and UCAR/NCAR, 2016), and bromine oxide (BrO) and iodine oxide (IO) radicals were measured by the University of Colorado Airborne Multi-AXis Differential Optical Absorption Spectroscopy (CU AMAX-DOAS) instrument with typical detection limits of 0.5 pptv for BrO, and 0.05 pptv for IO (Volkamer et al., 2015; Dix et al., 2016). 13 flights provide $O_3$ data and 16 flights provide BrO and IO data. Fig. 1 displays the location of all the observational data with an orange line for the cruise track, red lines for the flights in the tropics and green lines for the flights in the sub-tropics.

## 3 Model description

WRF-Chem (Grell et al., 2005) is a highly flexible community model for atmospheric research where aerosol-radiation-cloud feedback processes are taken into account. Version 3.7.1 is used in this study.

The sources of halogen atoms considered in this study are: inorganic I from the ocean (HOI and $I_2$), oceanic source of organic halogens ($CHBr_3$, $CH_2Br_2$, $CH_3I$, $CH_2BrCl$, $CHBrCl_2$, $CHBr_2Cl$, $CH_2I_2$, $CH_2IBr$ and $CH_2ICl$) and debromination ($Br_2$, IBr). These sources are explained in detail below in the following sections.

### 3.1 Oceanic fluxes

The oceanic emission of inorganic iodine (HOI and $I_2$) follows the deposition of $O_3$ to the surface ocean and reaction with iodide ($I^-$) (Carpenter et al., 2013). We use Eqs. 19 and 20 in Carpenter et al. (2013) for the calculation of these emissions. Ocean surface $I^-$ is parameterized using MacDonald et al. (2014) (see Fig. S1 in the supplementary information). Fig. 2 shows the average oceanic emission for inorganic iodine ($I_2$ in the left panel and HOI in the middle panel) during January and February 2012. Higher emissions for inorganic iodine occurs in the tropics with HOI being the dominant species.

Two different approaches for the marine emissions of the halocarbons ($CHBr_3$, $CH_2Br_2$, $CH_3I$, $CH_2BrCl$, $CHBrCl_2$, $CHBr_2Cl$, $CH_2I_2$, $CH_2IBr$ and $CH_2ICl$) are examined in this model. The first approach uses prescribed monthly average oceanic fluxes from Ziska et al. (2013) and the second computes the oceanic fluxes online. Prescribed monthly average oceanic fluxes from Ziska et al. (2013) were calculated using 6-hourly means of wind speed and sea surface temperature from the ERA-Interim meteorological assimilation database (Dee et al., 2011) for the years 1989-2011 ($1°$ x $1°$). Computing the emissions online accounts for an interaction between the modelled atmosphere and the ocean at each time step. Thus, this approach can respond to changes in meteorological parameters, like surface temperature and surface wind speed. A two-layer model (Liss and Slater, 1974) is used to calculate the halocarbons air-sea fluxes:

$$F = -K_a \cdot (C_g - K_H \cdot C_l) \tag{1}$$

where $K_a$ is the transfer velocity of the gas ($s^{-1}$), $C_g$ (ppm) and $C_l$ (nM) are the bulk gas and liquid-phase concentrations and $K_H$ is the Henry's law constant. $K_a$ is parameterized following Johnson (2010) which is mainly a function of wind speed and sea surface temperature (SST) taken from the model at each time-step. $C_g$ is also taken from the model. Halocarbon sea-water concentrations $C_l$ are taken from Ziska et al. (2013). Fig. 3 shows the average air-sea fluxes for $CHBr_3$, $CH_2Br_2$ and $CH_3I$ during January and February 2012 for the two approaches. Note that, the online calculation could increase, decrease or even reverse the fluxes in comparison with the prescribed emissions. This is the case for the online fluxes of $CHBr_3$ over the tropics where the model calculates negative fluxes whereas the prescribed fluxes are positive.

Recent studies suggest that the ocean is an important source of OVOCs such as acetaldehyde, ethanol and methanol (Coburn et al., 2014; Lawson et al., 2015; Mahajan et al., 2014; Myriokefalitakis et al., 2008; Sinreich et al., 2010; Volkamer et al., 2015; Yang et al., 2014; Fischer et al., 2012) that models do not generally consider or are not able to capture (Millet et al.,

2010; Sherwen et al., 2016a). Oceanic fluxes of several VOCs have been included into the WRF-Chem as part if this study. For the three OVOCs (acetaldehyde ($CH_3CHO$), ethanol ($C_2H_6O$) and methanol ($CH_3OH$)), the same online approach for the VSLH is used to calculate the marine fluxes where their sea-water concentrations are taken from Yang et al. (2014).

Emissions for alkenes and alkanes ($C_2H_4$, $C_3H_6$, $C_2H_6$, $C_3H_8$) are prescribed and based on the POET (Granier et al., 2005)
global inventory.

Deposition over the ocean for the halocarbons and OVOCs is included in the air-sea fluxes described above. For the rest of the species, dry deposition is calculated with the Wesely scheme (Wesely, 1989), which is used over land for several species. Washout of gases by precipitation is simulated using the scheme included in WRF-Chem (Grell and Dévényi, 2002; Zaveri et al., 2008) which was modified to include the Henry's law constants for the RHS shown in Table 1.

The sea-salt aerosol emissions parameterization used in this study is described in Archer-Nicholls et al. (2014). This parameterization is mainly a function of wind speed from the model and uses the emissions scheme from Gong et al. (1997) for particles with dry diameters of 0.45nm or more and for smaller particles uses Fuentes et al. (2010).

## 3.2  Gas-phase chemistry scheme

Our reaction mechanism is based on the MOZART-4 mechanism (Emmons et al., 2010; Knote et al., 2014). This mechanism
has been extended to include bromine, chlorine and iodine chemistry and has been coupled with the MOSAIC 4-bin aerosol module (Zaveri et al., 2008). A total of 48 species and 159 halogen reactions have been included (see Tables 2, 3 and 4 for details). Inorganic, organic and inter-halogens reactions come from the 1D model MISTRA (Sommariva and von Glasow, 2012). Production and loss reactions of the higher order of iodine oxides ($I_2O_X$, where X=2,3,4) reactions have been included into the model. Photochemistry of $I_2O_X$ species is still an area of high uncertainty in atmospheric iodine chemistry (Sommariva et al.,
2012; Saiz-Lopez et al., 2012b). Chemical loss of VSLH through oxidation by the hydroxyl radical (OH) and by photolysis is included using data from Sander et al. (2011b).

A schematic representation of the main bromine and iodine chemistry implemented in the model is shown in Fig 4. Chlorine chemistry is also included into the model, however, since our results are mainly focused on reactive bromine and iodine, we do not include chlorine chemistry in Fig 4.

Photolysis reactions included in the mechanism are listed in Table 4. To compute the photolysis rates the Fast Tropospheric Ultraviolet-Visible (FTUV) online scheme (Tie et al., 2003) is used. The quantum yields and cross section for the photolytic reactions of halogens are from JPL 10-6 (Sander et al., 2011b) and have been linearly interpolated onto the 17 wavelength bins used by FTUV. For $I_2O_X$ we use the quantum yield and cross section data from Gómez Martín et al. (2005).

### 3.2.1  Halogens and VOCs reactions

Reactions between halogens and VOCs can be important for regulating reactive halogen chemistry in the MBL by promoting the conversion of Cl and Br atoms into HCl and HBr or more stable organic halogenated intermediates. The oxidation of methane ($CH_4$), formaldehyde ($CH_2O$), acetaldehyde ($CH_3CHO$), methanol ($CH_3OH$), methyl hydroperoxide ($CH_3OOH$), methylperoxy ($CH_3O_2$), ethane ($C_2H_6$), ethene ($C_2H_4$) and propene ($C_3H_6$) by Cl is included in the chemical mechanism. In

addition, the oxidation of $CH_2O$, $CH_3CHO$, $C_2H_4$ and $C_3H_6$ by bromine is also included in the chemical mechanism with a simplified version of the chemical scheme presented in Toyota et al. (2004) used for reactions of bromine with alkenes:

$$Br + C_2H_4 + O_2 \quad \rightarrow \quad BrRO_2 \tag{R4}$$

$$Br + C_3H_6 + O_2 \quad \rightarrow \quad BrRO_2 \tag{R5}$$

where $BrRO_2$ is a brominated peroxy radical.

The loss of $BrRO_2$ is represented by the following reactions:

$$BrRO_2 + NO \quad \rightarrow \quad xHBr + (1-x)Br + CH_3CO_3 + NO_2 + 0.5CH_2O + HO_2 \tag{R6}$$

$$BrRO_2 + CH_3O_2 \quad \rightarrow \quad xHBr + (1-x)Br + CH_3CO_3 + HO_2 + CH_2O \tag{R7}$$

$$BrRO_2 + HO_2 \quad \rightarrow \quad BrOR + H_2O \tag{R8}$$

The loss of BrOR is represented by the following reactions:

$$BrOR + OH \quad \rightarrow \quad 0.5(xHBr + (1-x)Br) + 0.5BrRO2 + 0.5OH + 0.5CH_3CHO \tag{R9}$$

$$BrOR + h\nu \quad \rightarrow \quad xHBr + (1-x)BrOH + HO_2 + CH_3CO_3 + 0.5CH_2O \tag{R10}$$

where BrOR is a brominated organic specie and $x$ is a number between 0 and 1.

Reaction rates for these reactions and deposition velocities are taken from Toyota et al. (2004). Kinetic data for these
reactions is poor, and the partitioning of the products (HBr/Br) is not clear. Based on the Toyota et al. (2004) description, it is assumed that $x = 0.2$ such that the partitioning for HBr/Br is 1/4 (Toyota, pers. comm., 2017).

### 3.3   Heterogeneous chemistry

Heterogeneous reactions on particle surfaces involving halogens are summarised in Table 5. The heterogeneous chemistry is assumed to take place between a gas-phase species and an adsorbed species. The bulk aqueous phase chemistry in sea-salt
aerosols is not treated. Uptake coefficients are used to calculate first-order rate constants for heterogeneous loss of the gas-phase to the adsorbing surface (Jacob, 2000). This follows the approach use by McFiggans et al. (2000), which assumes a free molecular transfer regime approximation. The reaction rate constants, $K$ ($s^{-1}$), are given by:

$$K = \frac{\gamma}{4} \cdot S \cdot A \tag{2}$$

where $\gamma$ is the uptake coefficient, $S$ is the root-mean-square molecular speed (m $s^{-1}$) and $A$ is the total available aerosol
surface area density ($cm^2$ $cm^{-3}$). Equation 2 does not take account of any diffusion limitation (i.e. the rate at which gases can diffuse towards the aerosol surface).

We test the sensitivity of our results by adding the diffusion term, $D_g$ (cm$^2$ s$^{-1}$), following Brasseur and Jacob (2017), where the reaction rate constant, $K$ (s$^{-1}$), is given by:

$$K = \sum_{i=1}^{n_{bin}} \left[ \frac{4}{\gamma} + \frac{r_i}{D_g} \right] \cdot A_i \tag{3}$$

where $n_{bin}$ is the number of particle-size bins, $r_i$ is the particle radius for bin $i$ (cm) and $A_i$ is the available aerosol surface area density for bin $i$ (cm$^2$ cm$^{-3}$). $K$ is integrated over the aerosol size distribution in order to resolve the dependence of the rate constant for the particle radius. Following Brasseur and Jacob (2017), $D_g$ is given by:

$$D_g = 1.53 \cdot \times 10^{18} \left[ \frac{1}{m_g} + \frac{1}{m_{air}} \right]^{\frac{1}{2}} \cdot \frac{T^{\frac{1}{2}}}{n_a} \tag{4}$$

where $m_g$ is the mean molecular mass of the gas-phase specie (g/mol), $m_{air}$ is the molecular mass of air (g/mol), $n_a$ is the total air number density (molecules cm$^{-3}$), and $T$ is the temperature (K).

Second-order reaction rate constants are calculated by dividing the first-order rate constant by the concentrations of the adsorbed species. Heterogeneous halogen activation is very efficient under cold or stratospheric conditions as compared to moderate temperatures. For this reason, we have made a distinction between moderate (> 243.15 K) and cold temperatures (< 243.15 K) in some reactions. Uptake coefficients for reactions in Table 5 are based on literature values where available (Jacob, 2000; Sander et al., 2006; Ordóñez et al., 2012).

There are 6 reactions implemented for sea-salt particles. The sea-salt surface area is calculated in the following way: 1) using the mass of Na and Cl and the associated H$_2$O for each bin and the individual dry densities (for Na, Cl and H$_2$O) the total volume of those particles for each bin is calculated and then, 2) assuming that sea-salt aerosols are spheres, the total surface area is calculated for each bin using this volume and the radius of aerosols in each bin.

It is known that the chemistry involving the release of bromine from the sea-salt aerosol (debromination) is strongly pH dependent, being more efficient for acidified aerosol especially with a pH < 5.5 (Keene et al., 1998). Therefore, the pH value of the aerosol particles is calculated in the model for each size bin (see Zaveri et al. (2008) for further description of the pH calculation). We then apply a pH dependence to the heterogeneous reactions that occur on the surface of the sea-salt. When the pH < 5.5 debromination reactions occur with the release of Br$_2$ and IBr resulting from the uptake of BrNO$_3$, BrNO$_2$, HOBr, INO$_3$, INO$_2$ and HOI (R11-R16). When the pH > 5.5 no debromination reactions occur, although uptake of INO$_3$, INO$_2$ and HOI on the sea-salt still occurs (R17-R19) leading to a change in iodine speciation but no release of Br. See also Table 5.

If the pH < 5.5:

$$\text{BrNO}_3 \quad \rightarrow \quad 0.6\text{Br}_2 + \text{HNO}_3 \tag{R11}$$

$$\text{BrNO}_2 \quad \rightarrow \quad 0.6\text{Br}_2 + \text{HNO}_3 \tag{R12}$$

$$\text{HOBr} \quad \rightarrow \quad 0.6\text{Br}_2 \tag{R13}$$

$$\text{INO}_3 \quad \rightarrow \quad 0.5\text{IBr} + 0.5\text{ICl} + \text{HNO}_3 \tag{R14}$$

$$\text{INO}_2 \quad \rightarrow \quad 0.5\text{IBr} + 0.5\text{ICl} + \text{HNO}_3 \tag{R15}$$

$$\text{HOI} \quad \rightarrow \quad 0.5\text{IBr} + 0.5\text{ICl} \tag{R16}$$

If the pH > 5.5:

$$\text{INO}_3 \quad \rightarrow \quad 0.5\text{I}_2 + \text{HNO}_3 \tag{R17}$$

$$\text{INO}_2 \quad \rightarrow \quad 0.5\text{I}_2 + \text{HNO}_3 \tag{R18}$$

$$\text{HOI} \quad \rightarrow \quad 0.5\text{I}_2 \tag{R19}$$

Due to the high uncertainty in the debromination process, the fraction of $\text{Br}_2$ formed by reactions R11-R13 was chosen arbitrarily in order to add an extra bromine source in a simple way. A value of 0.6 was chosen. Fig. 2 shows the column-integrated fluxes for inorganic bromine ($\text{Br}_2$, right panel) during January and February 2012.

In addition, the heterogeneous uptake of $\text{N}_2\text{O}_5$ onto aerosol particles that contain $\text{Cl}^-$ to form $\text{ClNO}_2$ is considered in the model. After uptake $\text{N}_2\text{O}_5$ is taken up onto the particle, it reacts reversibly with liquid water to form protonated nitric acid intermediate ($\text{H}_2\text{ONO}^{+2}$). This then reacts with either liquid water, to form aqueous nitric acid ($\text{HNO}_3$), or with chloride ions to form $\text{ClNO}_2$. See Archer-Nicholls et al. (2014) for further description of this chemistry. In Archer-Nicholls et al. (2014) $\text{ClNO}_2$ was considered as an inert specie, however in our study $\text{ClNO}_2$ is not treated as an inert specie but is broken down via photolysis and reaction with OH (see Tables 2 and 4).

## 4 Model setup

The model is set up with a horizontal grid spacing of 30 km x 30 km and 30 vertical layers up to 50 hPa. Simulations that study the oxidation of VOCs by Br over the tropical area (described in Sec. 4.1) are performed with more vertical layers than the standard case in order to capture the vertical mixing in this area. Thus, 52 vertical layers up to 50h Pa are used in this case. The meteorological initial and lateral boundary conditions were determined using the ERA-Interim (Dee et al., 2011) data and the meteorology was reinitialized every 3 days to reproduce the observed transport. Chemical initial and boundary conditions (IC/BCs) are from the global atmospheric model GEOS-Chem described in Sherwen et al. (2016b). We conducted WRF-Chem simulations for January and February 2012 covering the TORERO domain (see Fig. 1). We performed a spin-up of 20 days. Table 6 describes the main configuration of the model.

## 4.1 Sensitivity studies

Ten different simulations were performed in this study. Our base simulation, WRF-DEBROM, considered all main processes involving halogen chemistry (sea-salt debromination, heterogeneous chemistry and reactions between halogens and VOCs) and computes the oceanic halocarbons fluxes online. The WRF-ZIS simulation is the same as WRF-DEBROM but uses prescribed oceanic emissions for the halocarbons. To test the sensitivity to the heterogeneous reaction rate constants, two runs were performed: the first one, in which the values for the uptake coefficient ($\gamma$ from Eq. 2) from Table 5 have been divided by two, WRF-GAMMADV2, and the second one, where equation 3 that has a diffusion term, is used, WRF-DIFF. To account for the importance of the debromination in sea-salt particles, we performed the simulation WRF-NODEBROM which is the same as the WRF-DEBROM simulation but without debromination. The WRF-NOHET simulation is the same as WRF-NODEBROM but without heterogeneous chemistry. A simulation with no halogen chemistry, WRF-NOHAL, is performed to study the effect of halogens on the tropospheric chemistry. All simulations except WRF-NOHAL use IC/BCs from the GEOS-Chem model that include halogens. WRF-NOHAL simulation uses IC/BCs from the GEOS-Chem model with no halogen chemistry. Finally, to study the oxidation of VOCs by halogens four simulations have been performed: 1) a simulation without the reactions of bromine reactions with alkenes (WRF-NOBRALKE), 2) a simulation without the reactions of bromine with aldehydes (WRF-NOBRALD), 3) a simulation without the reactions of bromine with VOCs, therefore neither alkenes nor aldehydes (WRF-NOBRVOCS) and 4) a simulation without reactions of bromine and chlorine with VOCs (WRF-NOHALVOCS). See Table 7 for a summary of all these simulations.

## 5 Model results

This section presents the model evaluation with observations of relevant trace gases. The model output is sampled at the nearest timestamp and grid box to the measurements. An ocean mask neglecting grid-boxes above land was applied to compute all model results.

### 5.1 Oceanic emissions: Halocarbons

Fig. 5 shows the time series of $CHBr_3$ (top-left panel), $CH_2Br_2$ (top-right panel) and $CH_3I$ (bottom-left panel) mixing ratios (in pptv) for the WRF-ZIS (green line) and WRF-DEBROM (black line) runs. In addition, the modelled wind speed (black line) is also shown in Fig. 5 (bottom-right panel). Measurements for the halocarbons and wind speed are represented by the solid red lines. Fig. 6 presents the time series of $CHBr_3$, $CH_2Br_2$ and $CH_3I$ water concentration (in pmol/L) from the measurements (dashed red lines) and from the Ziska et al. (2013) climatology (dashed blue lines) used to compute both the prescribed and online fluxes.

In general, both simulations reproduce the concentrations of the halocarbons to the right order of magnitude, although there are specific periods with a negative bias. We see a tendency to underestimate $CHBr_3$ for both model simulations during most of the period. This result is similar to the study of Hossaini et al. (2016) who compared eleven global models using different

emissions inventories. The majority of the models do not reproduce the observed concentrations in the tropical marine boundary layer. Over the tropics, high emissions observed are associated with tropical upwelling and active planktonic production (Class and Ballschmiter, 1988; Atlas et al., 1993). One reason for low $CHBr_3$ concentrations in our model simulations might be that the sea-water concentrations are too low in this area (see Fig. 6 for $CHBr_3$). The fluxes are also low (see Fig. S2 in the supplementary information). Note that Ziska et al. (2013) used only a very limited amount of data to derive the sea-water concentration for the halocarbons in our domain, which leads to uncertainty in the calculated fluxes. The modelled $CHBr_3$ is underestimated throughout the troposphere when is compared with aircraft observations (see Fig. S3 in the supplementary information). Atmospheric concentrations of $CH_2Br_2$ are in good agreement with the observations although the model underestimates the observed values by ∼0.5 pptv during the periods 6-10 and 22-25 February. Bromocarbon concentrations agree better with the measurements, when the oceanic fluxes are calculated online (WRF-DEBROM), in particular the underestimation is less for specific periods (e.g. 20th February for $CHBr_3$ and 10th and 22nd February for $CH_2Br_2$) in comparison with WRF-ZIS. Moreover, the correlation coefficients between the observations and the simulations are better for the WRF-DEBROM compared to WRF-ZIS: 0.48 and 0.3 for $CH_2Br_2$ and $CHBr_3$, respectively, in the case of WRF-ZIS and 0.65 and 0.43 for $CH_2Br_2$ and $CHBr_3$ in the case of WRF-DEBROM. Modelled $CH_3I$ concentrations show a similar trend to the observations, although, like the bromocarbons, both simulations underestimate the observations during specific periods (days 6-10 and 18-28 of February). This underestimation is more prominent in the WRF-DEBROM simulation. One reason for that could be that the wind speed from WRF-Chem is lower than the wind speed used to calculate the prescribed emissions, producing lower online fluxes. Nevertheless, the correlation coefficients between the observed and simulated $CH_3I$ atmospheric concentrations are better for WRF-DEBROM than for WRF-ZIS: 0.19 is calculated for the WRF-ZIS and 0.40 for the WRF-DEBROM simulation.

Specific periods of negative bias for both simulations demand further attention. A possible explanation for the underestimation in halocarbon atmospheric concentrations might be due to the input data (e.g. wind speed, SST, sea-water concentration) that we used to compute these fluxes. In the case of the online fluxes, between 6-8 of February the model underestimates wind speed and this is directly accompanied by an underestimation for all three halocarbons atmospheric concentrations. Ziska et al. (2013) demonstrate that changes in the input parameters, especially wind speed and SST, affect the fluxes calculation. The same study suggests that $CH_3I$ emissions are mainly influenced by variations of the wind speed. Moreover, the study of Lennartz et al. (2015), that uses the same sea-water concentration as our study, suggests that the negative bias in the modelled atmospheric concentrations could indicate regions where the sea-water concentration from the climatologies lacks hotspots, thus, missing an oceanic source regions. This is clearly seen for the sea-water concentrations of $CHBr_3$ (during most of the period), $CH_2Br_2$ (peaks around 15th February) and $CH_3I$ (peaks around 20th February) used in this study that seem to be too low in comparison with the observations (see Fig. 6). More data on the sea-water concentrations of these halocarbons in this region are required to better constrain the oceanic flux data sets available to models and so to improve the representation of these gases in the atmosphere.

## 5.2 Gas phase and heterogeneous chemistry: bromine and iodine partitioning

Fig. 7 compares model results sampled along 16 flight tracks with the observations for BrO (pptv) separating tropical from subtropical flights for the five simulations WRF-NOHET, WRF-NODEBROM, WRF-GAMMADV2, WRF-DIFF and WRF-DEBROM. Results indicate that there is an improvement of the modelled BrO throughout the troposphere in both the tropics and subtropics when the heterogeneous chemistry is included in both tropics and subtropics.

In the subtropics, higher values of BrO are found in the altitude range 11-13 km due to the lower altitude of the tropopause. Some data points in this altitude range will be in the lower stratosphere. There is really good agreement with the observations particularly in the middle and upper troposphere where the model is able to capture the higher values of BrO. Within the model, aerosols over the subtropical area tend to be alkaline, thus, BrO does not increase in this area when sea-salt debromination is included. Over the tropics, where the aerosol is more acidic and where the sea salt aerosols are mostly located (see emissions of $Br_2$ in Fig. 2), elevated BrO is seen with the inclusion of the debromination (WRF-DEBROM) in the MBL. Debromination improves the simulation of BrO concentrations in the middle troposphere although it excessively increases BrO levels up to 1 pptv in the MBL. Higher values are also seen in other modelling studies that include this process (Schmidt et al., 2016).

Areas, such as the tropics, where debromination dominates, the impact of halving gamma (WRF-GAMMADV2 run) is approximately half of the impact of including heterogeneous chemistry (i.e. the difference between the WRF-DEBROM run and the WRF-NOHET run) at least for the lower troposphere. Very little impact is seen in the UT, a slight decrease in BrO, when gamma is halved (WRF-GAMMADV2). The simulation in which diffusion limitation is considered in the heterogeneous reactions (WRF-DIFF) gives values of BrO that are generally between the results from the WRF-GAMMADV2 and WRF-DEBROM simulations. They are similar to the WRF-GAMMADV2 values in the MBL, but over the subtropics, where debromination is less, WRF-DIFF is very close to WRF-DEBROM values. Significant uncertainties still exist in the sea-salt debromination processes and the parameterisations used here might be too simple to represent them.

In addition, the conversion of BrO to HBr is dominated by the reaction between Br and OVOCs, such that the BrO overestimation seen in the MBL could be reduced if the modelled aldehydes concentrations were increased (discussed in section 5.3). However, a reduction in the debromination would also reduce BrO concentrations. Thus, in order to capture the BrO concentrations in the MBL the right balance between these two chemical processes is needed.

BrO is underestimated in the model by 1 pptv in the upper troposphere over the tropics. The breakdown of bromocarbons, such as $CHBr_3$, contributes to BrO concentrations in the UT, thus, a good representation of bromocarbons is needed. $CHBr_3$ is underestimated in the middle and upper troposphere especially over the tropics (see Fig. S3 in the supplementary information). The reason for that could be a combination of different factors: underestimation of the boundary conditions used in this study for $CHBr_3$, underestimation in the oceanic fluxes (see Fig. S2 in the supplementary information) and overestimation of the loss rates. Moreover, an underestimation in the heterogeneous chemistry or uncertainties in the reactions between the halogens and VOCs (discussed in section 5.3) can also contribute to the underestimation of BrO in the UT over the tropics.

Fig. 8 shows the vertical profile distribution for inorganic bromine ($Br_y$ in pptv) for the three simulations WRF-NOHET (left panels), WRF-NODEBROM (middle panels) and WRF-DEBROM (right panels) over the subtropics (top panels) and

tropics (bottom panels). Inorganic bromine concentrations increase with altitude with a maximum of 8 pptv at 14 km in the subtropical area for all three simulations. This reflects the lifetime of the bromocarbon species that breakdown and release Br in the UT and LS. Over the tropical area, inorganic bromine concentrations have a peak in the middle troposphere at 6 km and then decrease until 12 km then start to increase again. A big impact on the vertical $Br_y$ partitioning is seen between the three

simulations. With the inclusion of the heterogeneous chemistry, there is a decrease of HBr and an increase of more reactive species: di-halogens (BrCl, $Br_2$ and BrI) and BrO. HOBr increases and $BrNO_3$ decreases in the UT due to $BrNO_3$ hydrolysis. Over the tropics, $Br_y$ increases in the MBL ($\sim$ 4 pptv) when debromination is included (WRF-DEBROM). This enhancement is seen for all inorganic species with a maximum in the surface where the concentration of sea-salt aerosols is highest. Over the subtropical area, little difference is seen between WRF-NODEBROM and WRF-DEBROM.

Fig. 9 compares model results sampled along 16 flight tracks with the observations for IO separating tropical from subtropical flights for the five simulations WRF-NOHET, WRF-NODEBROM, WRF-GAMMADV2 ,WRF-DIFF and WRF-DEBROM. No clear impact is seen with the inclusion of the heterogeneous chemistry. At the surface, simulations with heterogeneous chemistry (WRF-DEBROM, WRF-GAMMADV2 WRF-DIFF, and WRF-NODEBROM) have slightly lower IO concentrations than the simulation without heterogeneous chemistry (WRF-NOHET). The main reason for that reduction is the sink for

the iodine oxides ($I_2O_X$, where X=2,3,4) included in the heterogeneous chemistry. Over the tropical region, the model overestimated surface IO. This overestimation might be explained by the large modelled inorganic iodine oceanic fluxes in this area. The biggest uncertainty in the inorganic iodine emissions parameterization is the calculation of the iodide concentration in the sea water. Over the subtropics, IO enhancements observed below 4 km are not captured by the model. Some studies suggest that there is abiotic $CH_3I$ production when dust contacts seawater containing iodide (Williams et al., 2007; Puentedura et al.,

2012). Implementing this chemistry into the model is out of the scope of this paper and further investigation is needed to explain whether the production of $CH_3I$ enhances the IO concentration or if there are other missing IO precursors. Gómez Martín et al. (2013) presented an analysis of observations of several gas-phase iodine species made during a field campaign in the eastern Pacific marine boundary layer and suggested that the presence of elevated $CH_3I$ does not have a big impact on the $IO_x$ concentrations due to $CH_3I$ in the MBL having a long lifetime ($\sim$2 days at the equator). An overestimation of modelled IO in

the UT needs further investigation. This overestimation is similar to other modelling studies Sherwen et al. (2016a). Changing the heterogeneous rate constants (difference between the WRF-DEBROM, WRF-GAMMADV2 and WRF-DIFF runs) has very little impact on IO.

      Fig. 10 shows the vertical profile distribution for inorganic iodine ($I_y$) for the three simulations WRF-NOHET (left panels), WRF-NODEBROM (middle panels) and WRF-DEBROM (right panels) over the subtropics (top panels) and tropics (bottom

panels). $I_y$ is higher in the MBL where it is emitted, especially in the tropical region, with HOI being the dominant species. Concentrations start to decrease above the MBL due to the removal of soluble species by the wet deposition. Unlike $Br_y$, we do not see a big impact on the vertical profile of $I_y$ partitioning with the inclusion of the heterogeneous chemistry. The only differences are the $I_y$ decreases in the surface with the inclusion of the heterogeneous chemistry, due to the removal of the iodine oxides, and the production of more di-halogens in the MBL, specially when debromination is included. Heterogenous

iodine reactions (reactions R11-R19) compete with the photolysis. Iodine species are more readily photolyzed, so less is taken up into the aerosol and the impact of heterogeneous chemistry is less.

## 5.3 Impact on VOCs

Several VOCs oceanic fluxes have been included in the model (see section 3.2.1) as well as the oxidation of VOCs by halogens. In order to see the impact of halogen reactions with the VOCs, average loss rates of all organic compounds due to the Cl and Br families are calculated as % of the total tropospheric losses over the ocean for the WRF-DEBROM simulation. Bromine accounts for 9.2% of the oxidation of $CH_3CHO$, 1.4% of $CH_2O$, 0.8% of $C_2H_4$ and 4.1% of $C_3H_6$. Chlorine accounts for 0.6% of the oxidation of $CH_3CHO$, 0.3% of $CH_2O$, 7.7% of $CH_3OH$, 0.8% of $CH_3OOH$, 0.6% of $CH_3O_2$, 35.5% of $C_2H_6$ and 10.5% of $C_3H_8$.

A sub-set of 9 flights from the TORERO campaign over the tropics is compared with the WRF-DEBROM, WRF-NOBRVOCS, WRF-NOBRALKE and WRF-NOBRALD simulations for BrO (pptv) in Fig. 11. Comparisons between WRF-DEBROM and WRF-NOBRVOCS simulations show a clear difference (1-4 pptv) throughout the whole troposphere. VOCs play an important role in the MBL regulating the reactive halogens. Without the bromine reactions with the VOCs, BrO concentrations are higher than observed in the MBL. In the middle and upper troposphere, where VOCs emitted from the ocean and forests are transported by convection, the model underestimates the amounts of BrO when these reactions are considered. The results obtained indicate that BrO is highly sensitive to the conversion of reactive bromine into more stable species by these reactions. The partitioning of the products of these reactions (HBr/Br), and thus the conversion of reactive bromine to more stable species, is highly uncertain (see section 3.2.1) and the results suggest that it might be too effective in these upper layers of the model.

In order to understand which families of VOCs have a higher impact on the BrO concentrations, the oxidation of alkenes and aldehydes by Br have been studied separately in WRF-NOBRALKE and WRF-NOBRALD simulations. Differences between WRF-DEBROM and WRF-NOBRALD are seen in the whole troposphere with higher differences up to 2 pptv in the MBL, where the concentrations of both bromine and aldehydes are high. The concentrations of the aldehydes are underestimated by the model, especially for $CH_3CHO$, meaning that BrO modelled concentrations would be even lower if the modelled concentrations of the aldehydes were reconciled with the observations. The model also seriously underestimates the observed glyoxal mixing ratios. The modelled values are typically ~1 pptv, whilst the observed values are around 30-40 pptv in the MBL decreasing to around 5-10 pptv in the upper troposphere (Volkamer et al., 2015; Sinreich et al., 2010). This illustrates that there are large gaps in our understanding of OVOCs in the remote marine atmosphere. Small differences are observed between WRF-DEBROM and WRF-NOBRALKE. However, differences up to 2 pptv between WRF-NOBRVOCS and WRF-NOBRALD are clearly seen especially in the MBL.

These findings suggest that when aldehyde oxidation by Br is included, reactive Br is reduced considerable, thus, limiting the amount of alkenes oxidation by Br (difference between WRF-DEBROM and WRF-NOBRALKE). However, when the oxidation of aldehydes is included, there is sufficient $Br_y$ present for the oxidation of alkenes by Br to have an impact on the BrO (difference between WRF-NOBRALD and WRF-NOBRVOCS).

Fig. 11 also shows the vertical profile distribution for inorganic bromine ($Br_y$ in pptv) for the WRF-NOBRVOCS run over the tropics (top-right panel). When reactions of bromine with VOCs are not included the amount of $Br_y$ increases considerably (difference between WRF-DEBROM and WRF-NOBRVOCS from Figs. 8 and 11), reaching values of 14 pptv in the MBL over the tropics. Moreover, when this chemistry is included, the partitioning of $Br_y$ shifts to more stable bromine species such as HBr.

Fig. 12 shows the vertical profile distribution for inorganic chlorine ($Cl_y$ in pptv) for the two simulations WRF-NOHALVOCS (left panels), WRF-DEBROM (middle panels) over the subtropics (top panels) and tropics (bottom panels). Regional average vertical partitioning of reactive chlorine species ($Cl^*$) is also shown (right panels) where $Cl^*$ is defined as $Cl_y$ gases other than HCl. When the VOCs react with Cl (WRF-DEBROM), almost all the inorganic Cl is in the form of HCl (see Fig. 12). When these reactions are not considered (WRF-NOHALVOCS), $Cl_y$ increases and there is a shift in the partitioning to more reactive chlorine increases, in particular HOCl, but also ClO and the di-halogens.

From this, we concluded that VOCs play an important role in the reactive bromine and chlorine concentrations. Therefore marine emissions of VOCs as well as halogen reactions with VOCs need to be included in models. However, large uncertainties still exist in some of these reactions (see Sec. 3.2.1).

## 5.4 Impact on $O_3$ and $O_x$

Fig. 13 (left panel) presents a comparison of modelled $O_3$ from 7 simulations (WRF-DEBROM, WRF-GAMMADV2, WRF-DIFF, WRF-NODEBROM, WRF-NOHET, WRF-NOHALVOCS, and WRF-NOHAL) sampled along 13 flight tracks with the observed $O_3$ (ppbv). $O_3$ is overestimated when halogens are not included (WRF-NOHAL) except in the upper troposphere. When halogens are included, the model (WRF-DEBROM) is in line with the observations, capturing the $O_3$ gradient and variability of data throughout the troposphere. The average difference between WRF-DEBROM and WRF-NOHAL simulations throughout the troposphere is 7 ppbv. In the MBL, high concentrations of halogens due to ocean emissions destroy $O_3$ and contribute to a negative bias up to 8 ppbv for WRF-DEBROM run. In the middle troposphere, the model results (WRF-DEBROM) improve with the inclusion of halogens, where the average underestimation is reduced from 4.0 to 2.4 ppbv. In the upper troposphere, where the differences between the the simulations (WRF-DEBROM and WRF-NOHAL) are mainly driven by the boundary conditions used for each simulation, both simulations underestimate the ozone concentrations. The heterogeneous halogen chemistry has an impact on $O_3$ concentrations where a difference of up to 3 ppbv of $O_3$ is seen between the simulation with and without heterogeneous chemistry (WRF-DEBROM run WRF-NOHET run, respectively) mainly in the MBL. Dividing gamma by 2 (WRF-GAMMADV2) and considering the diffusion limitation (WRF-DIFF) reduces this difference to around 2 ppbv. The modelled $O_3$ is highly sensitive to the inclusion of the reactions of the halogens with the VOCs (WRF-NOHALVOCS) where $O_3$ concentrations are much lower (between 12-7 ppbv) than in the WRF-DEBROM run.

Fig. 13 (middle and right panels) shows the regional effects of halogen chemistry on simulated $O_3$ concentrations at the surface. Surface mean bias (ppbv) and relative mean bias (%) between the simulation with no halogen chemistry (WRF-NOHAL) and with halogen chemistry (WRF-DEBROM) for the simulation period are presented. We find that the regional $O_3$ concentrations are reduced by 2-18 ppbv, corresponding to 25-70 %, with the inclusion of the halogens. Over the tropics, there

is a substantial decrease of $O_3$ (> 8ppbv, > 40%). As we see in Fig. 2 and 3, there are high iodine and bromocarbon emissions and especially large amounts of bromine produced from debromination over this area. These destroy ozone and contribute to higher difference in $O_3$ concentrations in this area.

The odd oxygen $O_x$ is defined as:

$$O_x = O(^3P) + O(^1D) + O_3 + NO_2 + 2\times NO_3 + HNO_3 + HO_2NO_2 + 3\times N_2O_5 + PAN + MPAN + ONIT + ONITR + ISOPNO_3$$
$$+ PBZNIT + MBONO_3O_2 + XO + HOX + XNO_2 + 2\times XNO_3 + 2\times OIO + 2\times I_2O_2 + 3\times I_2O_3 + 4\times I_2O_4 + 2\times OClO,$$

where X=Cl, Br and I; PAN= peroxyacetyl nitrate, MPAN= methacryloyl peroxynitrate; ONIT= organic nitrate; ONITR= lumped isoprene nitrate; ISOPNO$_3$= peroxy radical from NO$_3$+ISOP; PBZNIT= peroxybenzoyl nitrate; MBONO$_3$O$_2$= peroxy radical from NO$_3$ + 2 methyl-3-buten-2-ol.

The $O_x$ loss resulting from reactions with each of the ozone depleting families ($O_x$ , HO$_x$, NO$_y$, VOCs, Br, Cl and I) is calculated. Note that to calculate the $O_x$ loss due to the $O_x$ depleting family we only consider reactions involving O($^3$P), O($^1$D) and $O_3$. The average tropospheric vertical profile of $O_x$ loss grouped by ozone depleting families for the WRF-DEBROM simulation is given in Fig. 14. Fig. 15 summarises the relative contribution of each halogen family averaged at different altitude intervals for the WRF-DEBROM , WRF-GAMMADV2, WRF-DIFF, WRF-NODEBROM, WRF-NOHET and WRF-NOHALVOCS simulations.

The regional average $O_x$ percentage loss due to the halogens in our model domain is 34%, 18% and 40% in the MBL (p>900hPa), FT (350<p<900hPa) and UT (350hPa<p<trop), respectively for the WRF-DEBROM simulation. The MBL $O_x$ loss is in good agreement with Sherwen et al. (2016b) that reported 33% and Prados-Roman et al. (2015) reported 31%. The tropospheric $O_x$ loss due to the BrO$_x$, IO$_x$ and ClO$_x$ cycles is 14%, 16%, 1% throughout the troposphere, respectively for the WRF-DEBROM simulation. The very fast catalytic reactions of iodine species make the iodine loss higher than for bromine and chlorine, especially in the MBL for all simulations that include halogens-VOCs reactions (19-23%). With the inclusion of the sea-salt debromination, $O_x$ loss due to the bromine is 14% in the MBL. In the upper troposphere, iodine contributes 18-23% and bromine 14-19% to the total $O_x$ loss. The impact of halogen chemistry on the tropospheric $O_x$ loss is 31 % for the WRF-DEBROM simulation. This value is comparable with other studies that reported 28% over the tropics (Saiz-Lopez et al., 2015) and 21.4% at the global scale (Sherwen et al., 2016b). Moreover, our results are in agreement with Wang et al. (2015), that used a box model and concluded that bromine and iodine are responsible for 34% of the column-integrated loss of tropospheric $O_3$. The tropospheric $O_x$ loss due to the iodine is higher than the box model study of Dix et al. (2013), that concluded that the fraction of iodine-induced ozone loss generally is around 10%. When comparing different simulations with the WRF-DEBROM run, the biggest difference is seen with WRF-NOHALVOCS simulation, where around 60% of $O_x$ is removed by halogens. BrO is much higher when the VOC reactions are not included (see Fig. 11), which explains why the amount of $O_x$ loss by BrO$_x$ reactions is much larger (20.5%). Moreover, the big change though is for the ClO$_x$ which increases from < 1% to 26%. Cl is very important in the oxidation of the alkanes. When this chemistry is not included the concentrations of Cl$_y$ increases and there is an impact on the partitioning increasing reactive species (see Fig. 12), hence, the ClO$_x$ cycles play an important role in $O_x$ loss. It should be noted that very little is known about the abundance and distribution of Cl$_y$ so this is a large uncertainty. Therefore, a large uncertainty in the impact of halogen cycling on the $O_3$ budget are the reactions of

halogens with VOCs. In the model runs performed, excluding these reactions doubled the percentage contribution of halogens to $O_x$ loss (i.e. increase it from 31% to 60%) in the troposphere. Heterogeneous chemistry (including debromination) has the effect of increasing the $O_x$ loss by halogen cycling from 25 to 31% for the whole troposphere (i.e. comparision between WRF-NOHET and WRF-DEBROM runs). For the UT the equivalent values are 37% to 40%, for the FT 13% to 18% and for the

MBL 23% to 34%. Hence, heterogeneous chemistry increases the percentage of the $O_x$ loss that is attributable to the halogens by about 6% for the troposphere ranging from 3% to 11% depending on the region of the troposphere. Dividing gamma by 2 (WRF-GAMMADV2) and considering the diffusion limitation (WRF-DIFF) reduces the $O_x$ loss in the troposphere by the halogens to 3% and 2%, respectively. Note that the gas phase halogen chemistry makes a bigger contribution of around 25% (WRF-NOHET run) to the $O_x$ loss for the troposphere ranging from 13% to 37% depending on the region of the troposphere.

Therefore, the overall impact of the halogen chemistry on $O_x$ loss appears not to be very sensitive to the treatment of the heterogeneous chemistry.

## 6   Conclusions

We have presented a regional 3D tropospheric model that includes halogen chemistry (bromine, iodine and chlorine). A comprehensive description has been provided for the halogen gas-phase chemistry, the heterogeneous recycling reactions in sea-salt

aerosol and other particles, reactions of reactive halogens with volatile organic compounds (VOCs) and the oceanic emissions of halocarbons, inorganic iodine and several VOCs. It is the first time that a comprehensive halogen chemistry mechanism has been added into the online WRF-Chem model. Our results provide useful insight regarding the potential importance of reactive halogens in the tropical marine atmosphere and the many uncertainties that remain. Field data from the TORERO campaign (Jan-Feb 2012) has been used in the model evaluation.

Two different approaches to compute marine emissions, online and prescribed, for the VSLH are discussed here. There is an improvement using online fluxes, WRF-DEBROM, in comparison with prescribed fluxes, WRF-ZIS, especially for $CH_2Br_2$ and $CHBr_3$ atmospheric concentrations, where the overestimation seen for the model in comparison with ship measurements is decreased for specific periods. During the whole period, an underestimation is seen for both simulations for $CHBr_3$. This underestimation is similar to other modelling studies, which indicates the oceanic fluxes for $CHBr_3$ in this region are not well

determined. Results indicate that the input data (especially wind-speed and water concentrations) used in this study to calculate marine fluxes underestimate halocarbon concentrations. Large underestimation of $CHBr_3$ and $CH_3I$ concentrations throughout the troposphere is seen when compared to the aircraft observations.

Four sensitivity studies are compared in order to understand the impact of the heterogeneous chemistry for bromine and iodine species. Results show that the inclusion of heterogeneous chemistry on marine aerosol has a considerable impact on

the $Br_y$ partitioning, increasing reactive species like BrO. An increase of $Br_y$ is seen in the tropical MBL when debromination processes are included, due to the presence of relatively acidic particles.

The oxidation of alkenes and aldehydes by bromine have been studied in three different sensitivity runs. These runs suggest that reactions of bromine with OVOCs have a big impact on the BrO concentrations. The reactions between Br and alde-

hydes were found to be particularly important, despite the model underestimating the amount of aldehydes observed in the atmosphere.

The model shows an overall good agreement with the observed IO vertical profile. Higher modelled concentrations in the surface are seen over the tropics indicating that inorganic iodine emissions might be too high in this area. The model is not able to capture the IO enhancements sometimes seen below 4 km over the subtropical area. Unlike, $Br_y$, the $I_y$ partitioning is found to be relatively insensitive to inclusion of the heterogeneous chemistry.

The model captures the $O_3$ vertical profile in the free troposphere. The simulation with halogens (WRF-DEBROM) underestimates the observed $O_3$ values in the MBL, where the oceanic emissions of the halogenated species are higher. Over the tropics, the regional surface $O_3$ concentrations are reduced between 2-18 ppbv with the inclusion of the halogens. When heterogeneous chemistry is included $O_3$ concentrations is reduced by up to 3 ppbv in the MBL. The biggest difference (7-12 ppbv) in $O_3$ values is seen when reactions between Br and Cl and VOCs are not considered (WRF-NOHALVOCS run).

In our simulations, halogens constitute 25-60% of the overall tropospheric $O_x$ loss. This range of values is comparable with other studies. Uncertainties in the heterogeneous chemistry accounted for only a small proportion of this range (25% to 31% of the $O_x$ loss). When reactions between Br and Cl with VOCs are not considered (WRF-NOHALVOCS), $O_x$ loss by $BrO_x$, $ClO_x$ and $IO_x$ cycles is high (60%) which accounts for the upper limit of the overall range. The model results are clearly very sensitive to the VOCs and this is a large uncertainty given that their emissions over these remote areas are poorly known.

Our model results suggest that including halogen chemistry has a large affect on $O_3$ (7 ppbv) and contributes typically about 25-30% of $O_x$ loss. Including heterogeneous halogen chemistry has a big impact on the $Br_y$ partitioning, but not on the $I_y$ partitioning. However, it does not have a large impact on the $O_3$ concentrations or the percentage of $O_x$ loss via halogen chemistry. Therefore, although the uncertainties in the heterogeneous chemistry are large the $O_x$ appears to be relatively insensitive to these uncertainties. However, the modelled $O_3$ and $O_x$ loss are very sensitive to the reactions between the halogens and the VOCs. Excluding these reactions leads to greater amounts of the reactive halogen species (Figs. 11 and 12) less $O_3$ (Fig. 13) and greater $O_x$ loss from halogens (60%) (Fig. 15), in particularly from $ClO_x$. Very little is known about the abundance and distribution of $Cl_y$ so this is a large uncertainty. There are also large uncertainties in the degree to which Br is recycled or converted to the more stable product HBr in the reactions following Br reactions with the alkenes. Moreover, there is considerably uncertainty in the emissions and distributions of the VOCs in the remote marine atmosphere.

More data is required at the process level from laboratory studies along with field observations of, for example, more $Br_y$, $I_y$ and $Cl_y$ species, to better constrain the modelled representation of these processes and to verify if halogens really do have such a large impact on $O_x$ in the tropical troposphere. This is important given that the oxidising capacity of this region of the atmosphere has a large impact on the lifetime of many pollutants including methane, a key greenhouse gas.

*Code availability.* The WRF-Chem model code is available from https://github.com/wrf-model/WRF, with the specific code used in this study available from the authors upon request (alba.badia.moragas@gmail.com).

*Data availability.* The TORERO data are available from the TORERO data archive: https://www.eol.ucar.edu/field_projects/torero. The TORERO data set is open for use by the public, subject to the data policy: https://www.eol.ucar.edu/content/torero-data-policy.

*Author contributions.* AB carried out all the model simulations and data analysis, and led the interpretation of the results and prepared the manuscript with contributions from all co-authors. CER contributed to the interpretation of the results and provided extensive comments on manuscript. ARB and AS made several comments and suggestions. RV, TKK, ECA, RSH, LJC and SJA conducted and provided the TORERO measurements. TS provided input data to run the model. RvG provided the initial motivation to this study, designed the research and secured the funding.

*Competing interests.* The authors declare that they have no conflict of interest.

*Acknowledgements.* This work is funded by the National Environmental Research Council (NERC) grant NE/L005271/1. The authors wish to thank the TORERO team, specially Barbara Dix and Theodore Konstantinos. TORERO was supported by NSF under award AGS-1104104 (PI: R.Volkamer). The involvement of the NSF-sponsored Lower Atmospheric Observing Facilities, managed and operated by the National Center for Atmospheric Research (NCAR) Earth Observing Laboratory (EOL), is acknowledged. R.V. acknowledges funding from NSF award AGS-1620530. L.J.C. acknowledges support from NERC (award NE/J00619X/1). Also, thanks to Carlos Cuevas, Douglas Lowe, Gordon McFiggans, Kenjiro Toyota, Peter Braüer, Luke Surl, Deanna Donohoue and Roberto Sommariva for their constructive suggestions and feedback during this study. Finally, this work is specially dedicated to the friendship and memory of Professor Roland von Glasow.

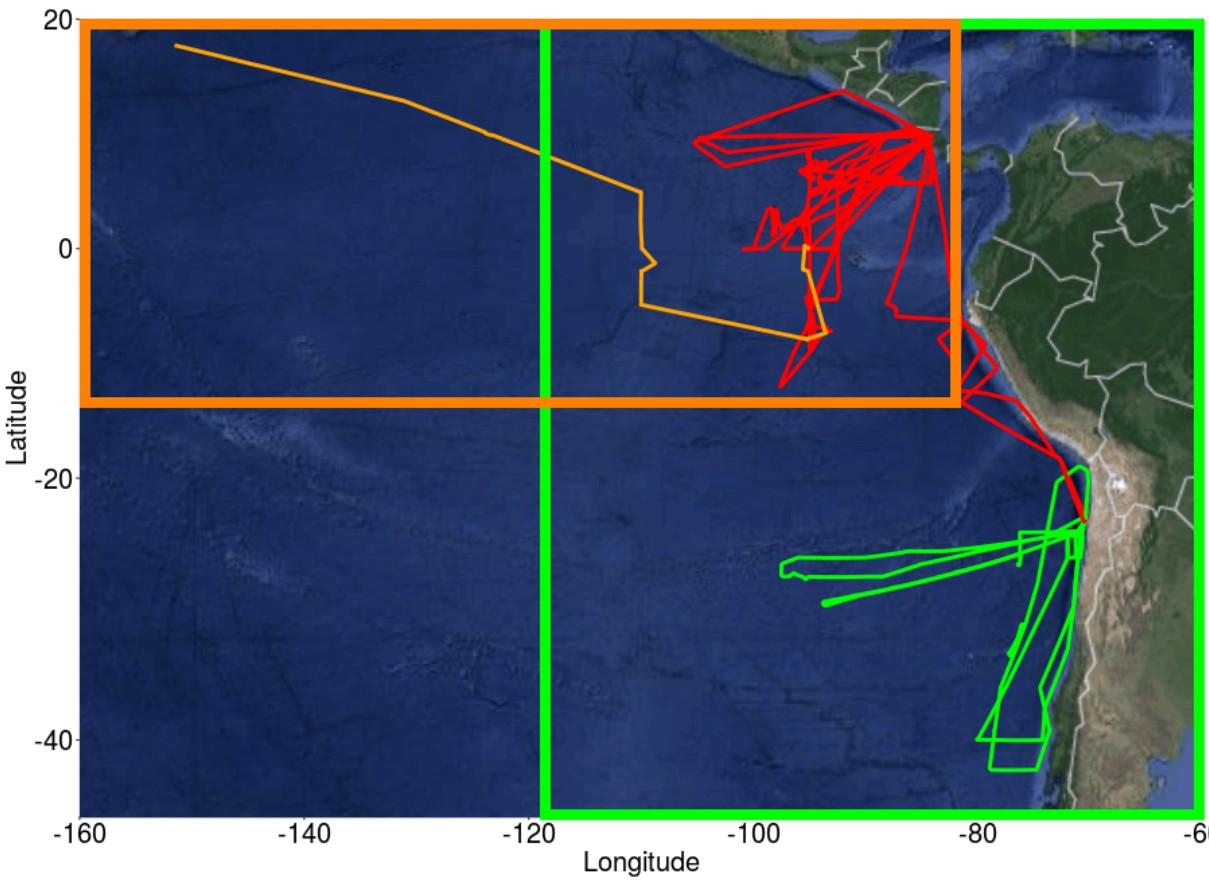

**Figure 1.** Flight and cruise tracks from the TORERO campaign (January-February 2012). Cruise track is represented by an light orange line. Flights are grouped by the following regions: tropical (red lines) and subtropical (green lines). Two different domains where defined: domain to evaluate the cruises (dark orange square) and domain to evaluate aircrafts (green square).

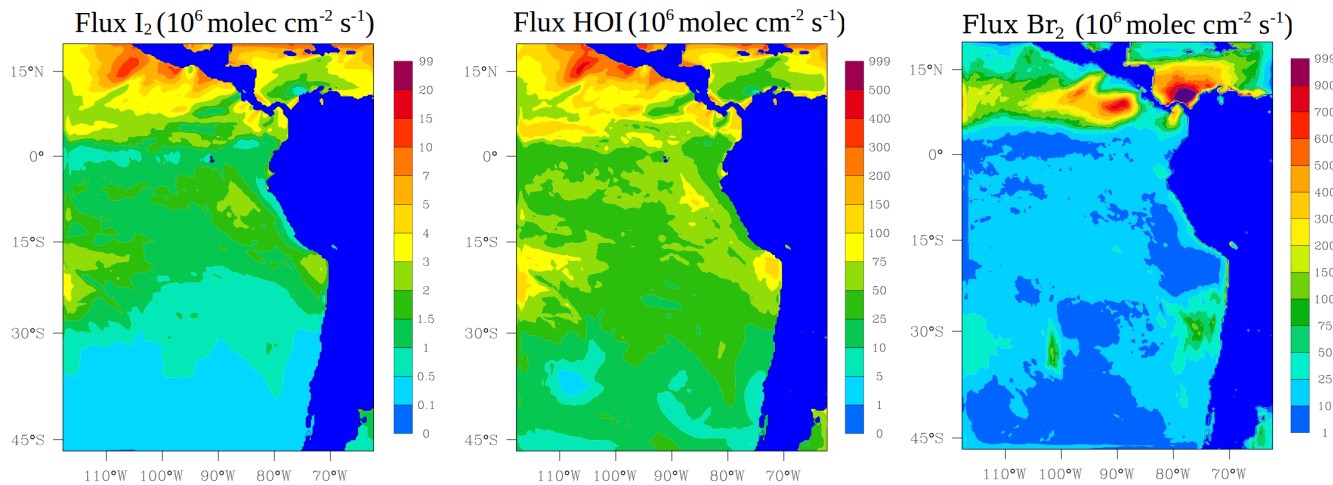

**Figure 2.** Mean oceanic surface fluxes for inorganic iodine: $I_2$ (left panel) and HOI (middle panel). The column-integrated fluxes for inorganic bromine ($Br_2$, right panel) from the debromination process during January and February 2012 are also shown. Values are given in $10^6$ molec $cm^{-2}$ $s^{-1}$.

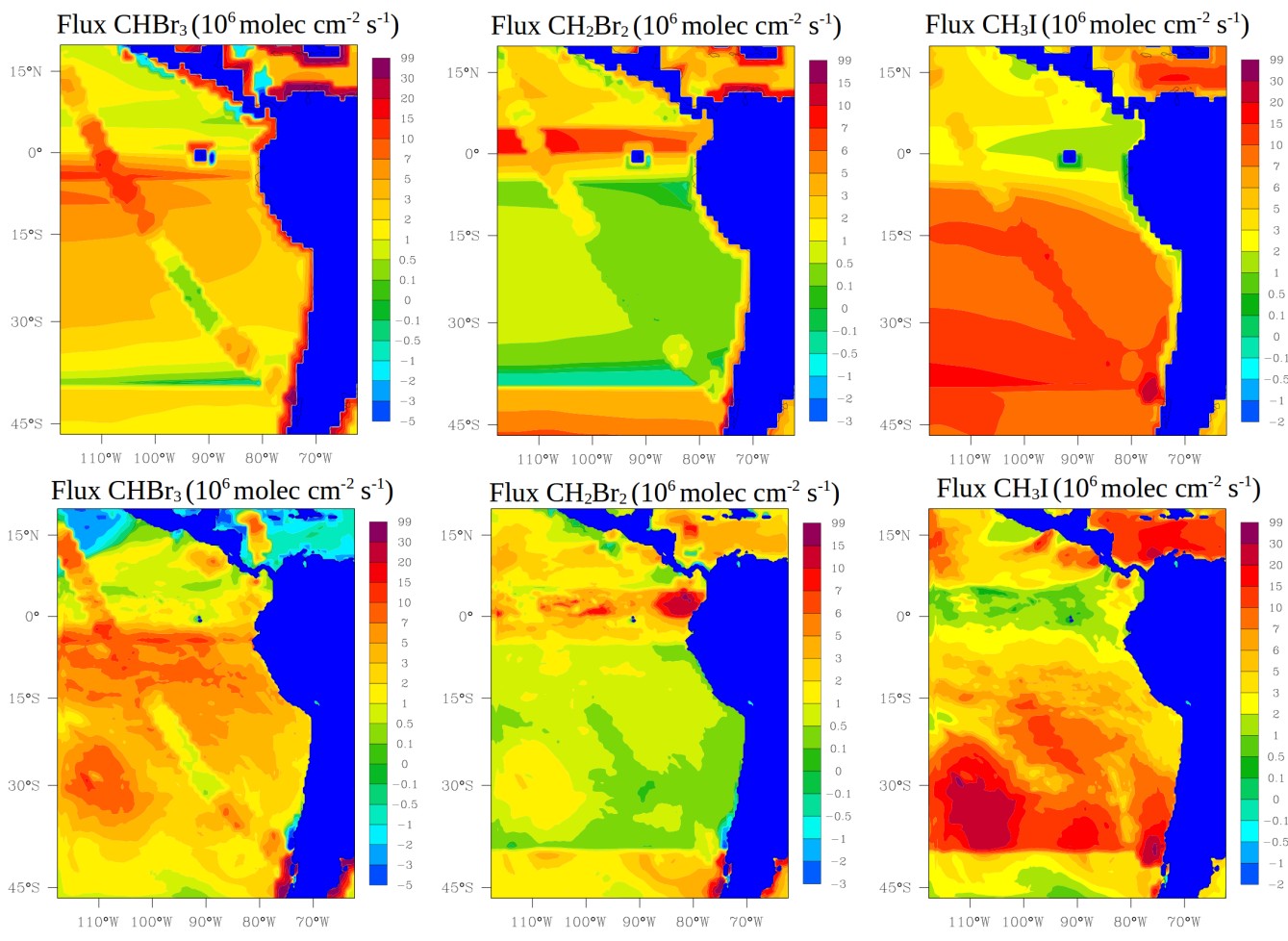

**Figure 3.** Mean oceanic fluxes for halocarbons ($CHBr_3$, $CH_2Br_2$ and $CH_3I$) during January and February 2012. Prescribed fluxes are shown on the top and online fluxes on the bottom. Values are given in $10^6$ molec cm$^{-2}$ s$^{-1}$.

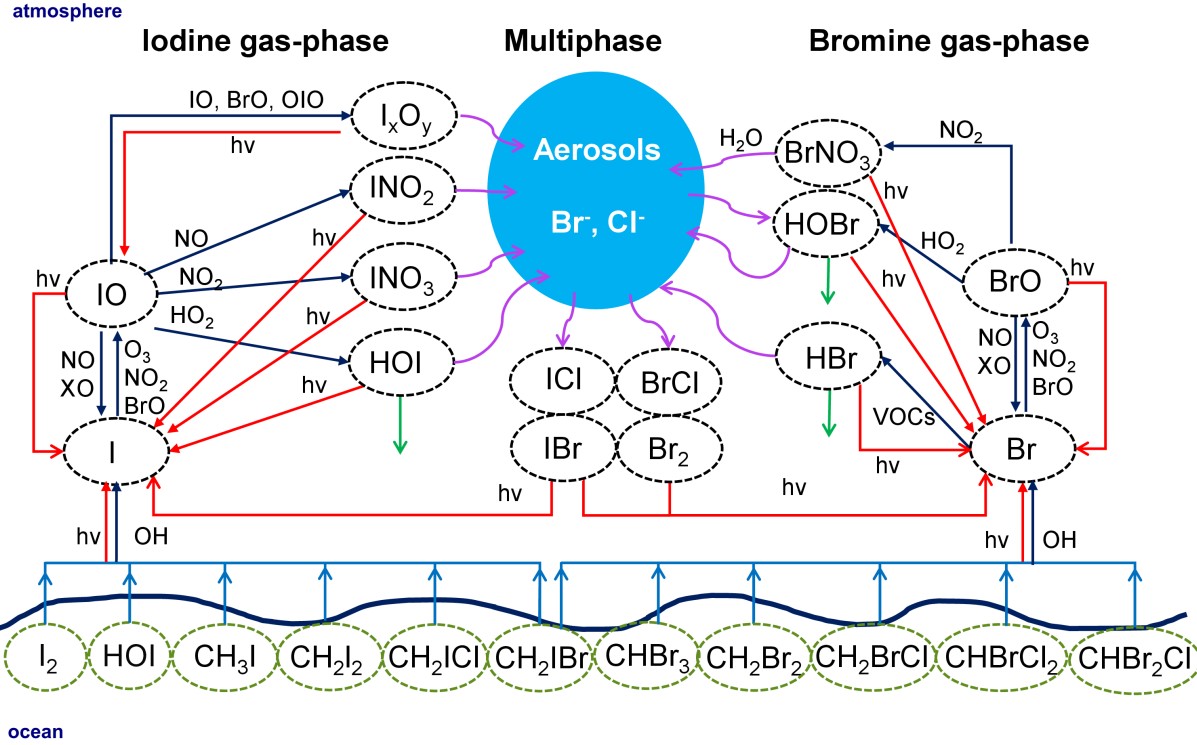

**Figure 4.** Schematic representation of the implemented iodine and bromine chemistry in WRF-Chem. Chlorine chemistry have been included into the model, since our results are mainly focused on reactive bromine and iodine, we decide not to include chlorine chemistry in this figure. Red lines represent photolytic reactions, dark blue lines gas-phase pathways, light blue lines fluxes, green lines deposition and purple curved lines heterogeneous pathways.

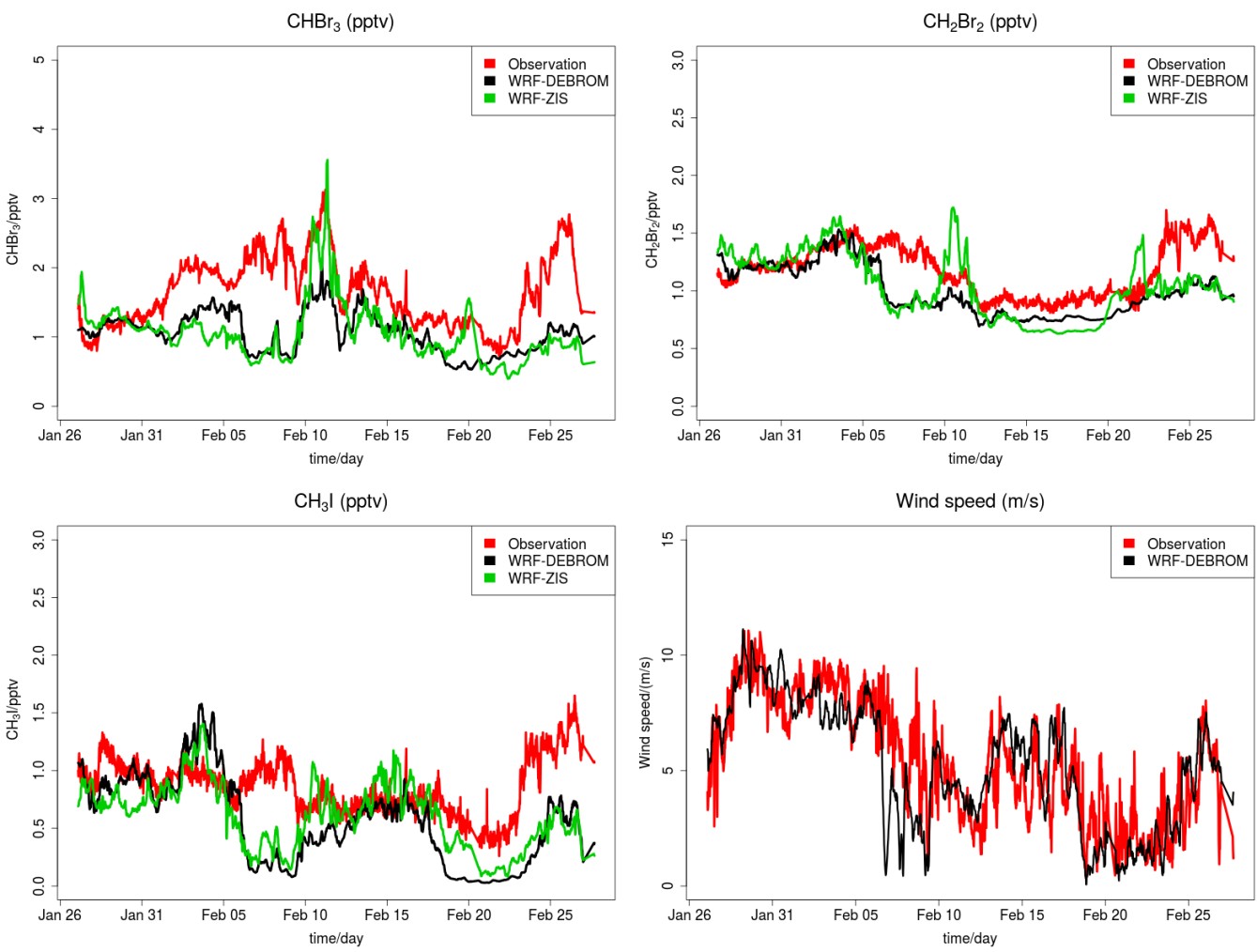

**Figure 5.** Time series of CHBr$_3$ (top left), CH$_2$Br$_2$ (top right) and CH$_3$I (bottom left) mixing rations (in pptv) for the WRF-ZIS (green line) and WRF-DEBROM (black line) runs during the period of the TORERO campaign in 2012. On the bottom-right, the wind speed (m s$^{-1}$) of the model is shown with a black line. Measurements during the TORERO campaign are depicted with red lines.

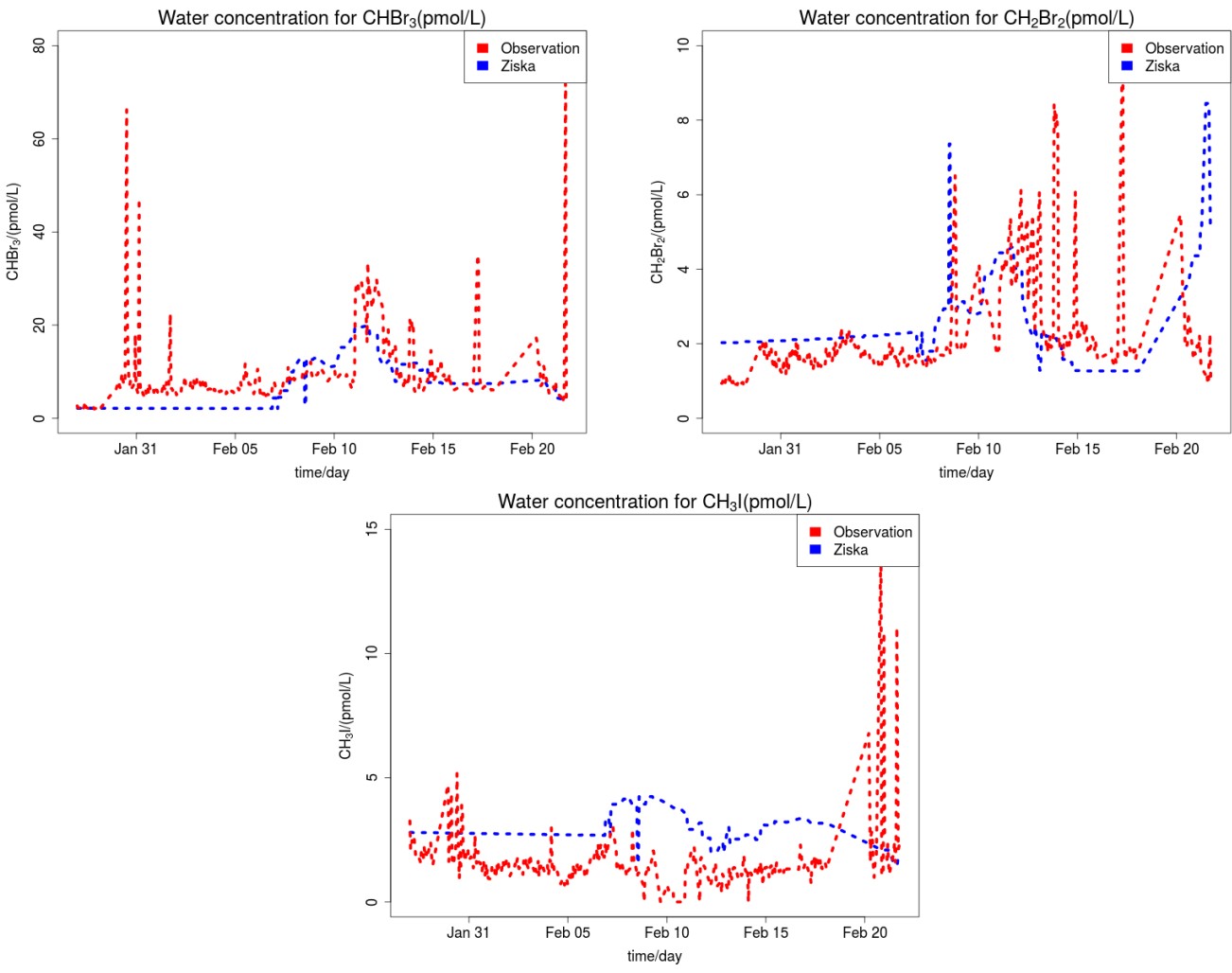

**Figure 6.** Time series of measured CHBr$_3$ (top left), CH$_2$Br$_2$ (top right) and CH$_3$I (bottom) water concentration (in pmol/L) during the TORERO campaign (red dashed line) and from the Ziska et al. (2013) climatology (blue dashed line).

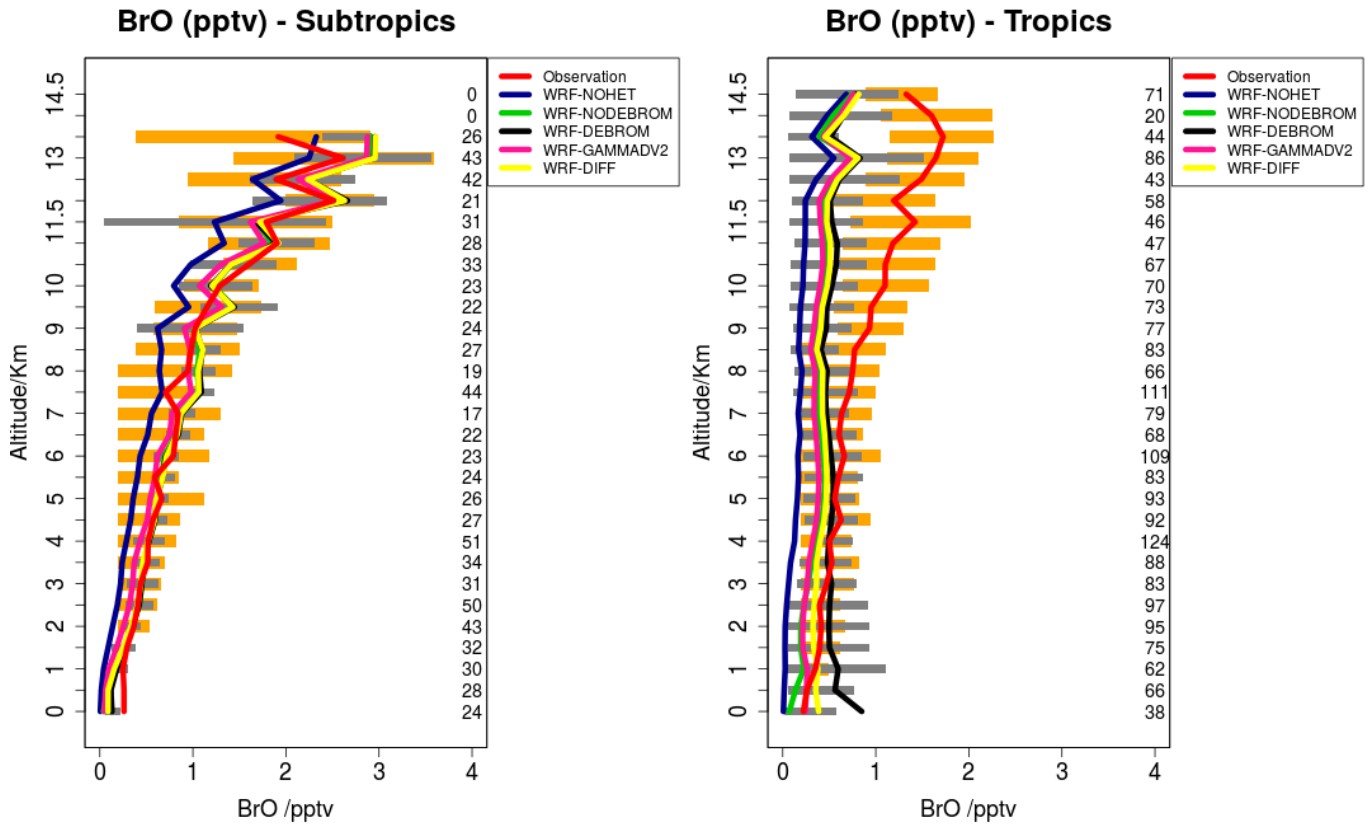

**Figure 7.** Mean vertical profile of BrO (pptv) over the subtropics (left) and tropics (right). An average over 16 flights of the TORERO campaign (red line) are compared to the 5 different WRF-Chem simulations: WRF-NOHET (blue line), WRF-NODEBROM (green line), WRF-DEBROM (black line), WRF-GAMMADV2 (pink line) and WRF-DIFF (yellow line). Orange and grey horizontal bars indicate the 25th-75th quartile interval for the observations of the TORERO campaign and WRF-DEBROM simulation, respectively. Values are considered in 0.5 km bins and the number of aircraft measurement points for each altitude are given on the right side of each plot.

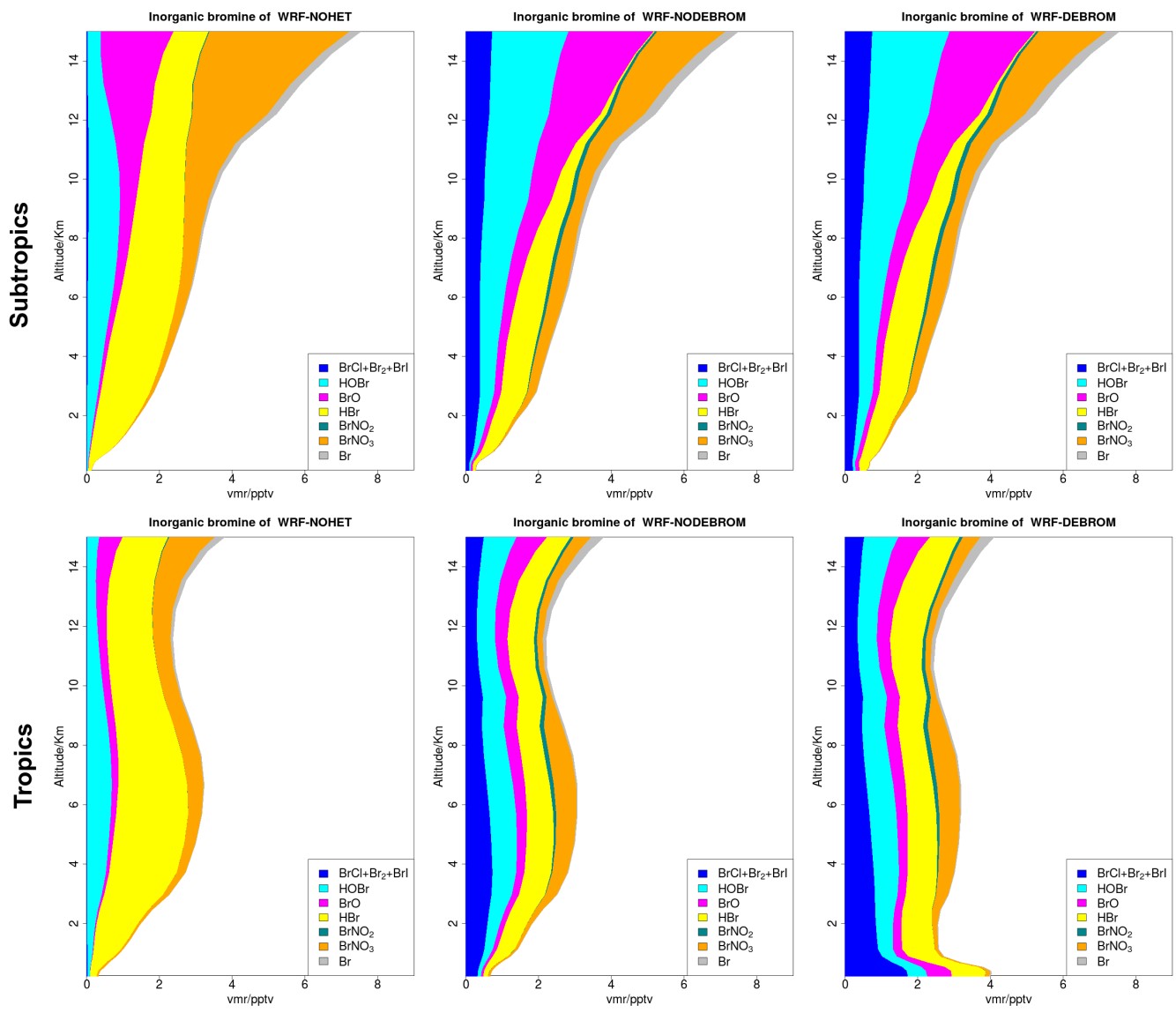

**Figure 8.** Regional average vertical partitioning of inorganic bromine ($Br_y$) for the three different simulations WRF-NOHET (left panel), WRF-NODEBROM (middle panel) and WRF-DEBROM (right panel) during January and February 2012. Top panels are over the subtropical area and bottom panels over the tropical. Units are in pptv.

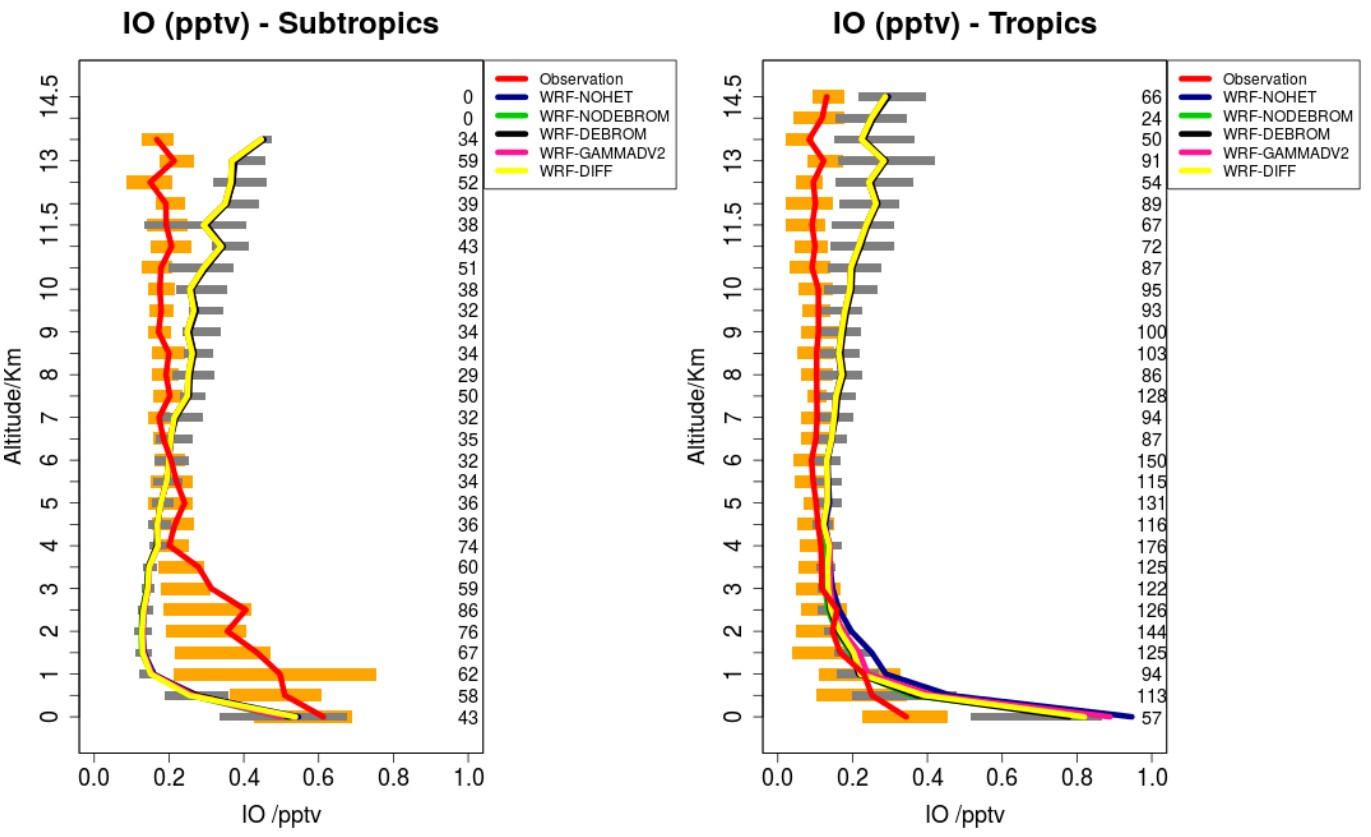

**Figure 9.** Mean vertical profile of IO (pptv) over the subtropics (left) and tropics (right). An average over 16 flights of the TORERO campaign (red line) are compared to the 5 different WRF-Chem simulations: WRF-NOHET (blue line), WRF-NODEBROM (green line), WRF-DEBROM (black line), WRF-GAMMADV2 (pink line) and WRF-DIFF (yellow line). Orange and grey horizontal bars indicate the 25th-75th quartile interval for the observations of the TORERO campaign and WRF-DEBROM simulation, respectively. Values are considered in 0.5 km bins and the number of aircraft measurement points for each altitude are given on the right side of each plot.

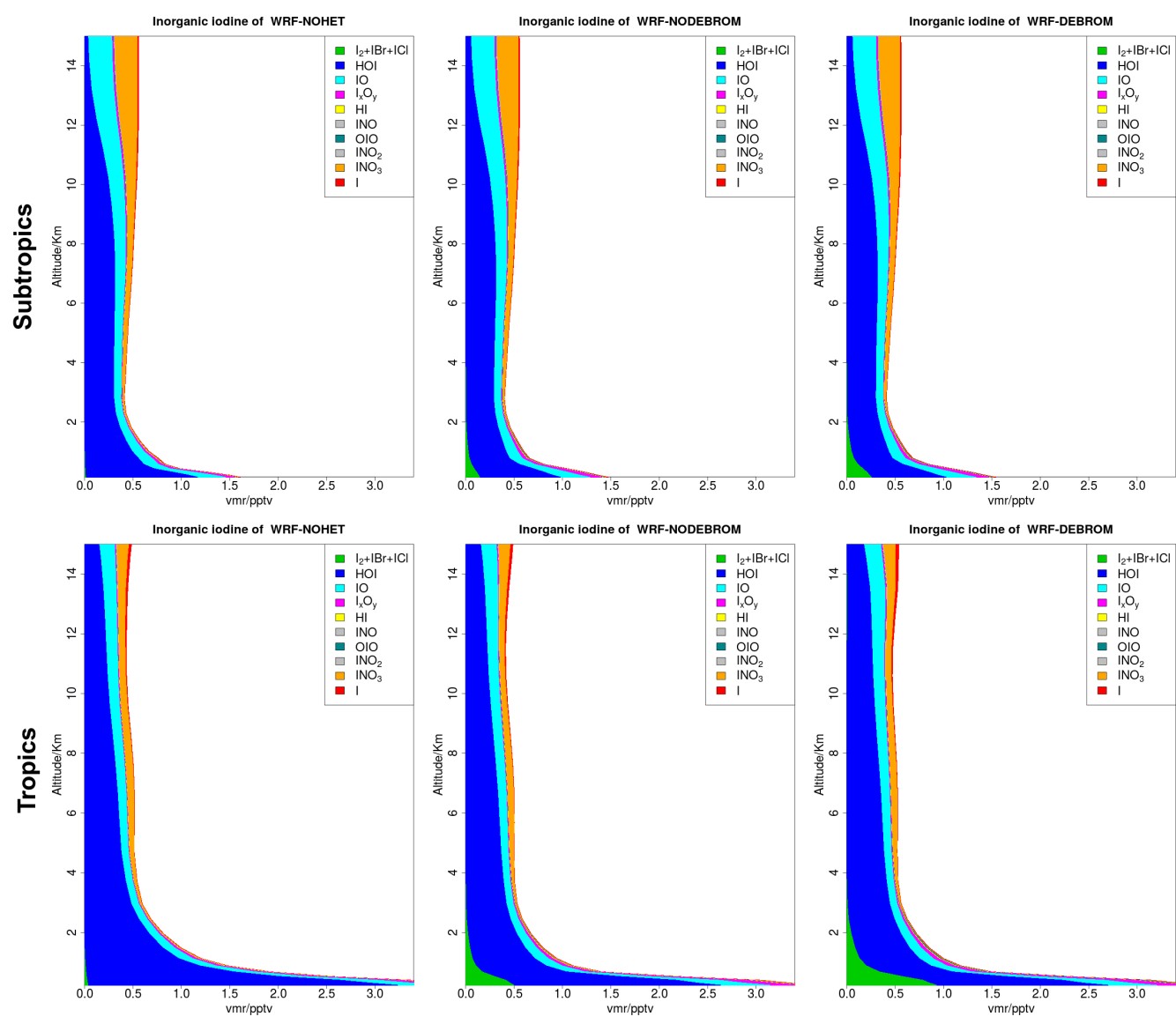

**Figure 10.** Regional average vertical partitioning of inorganic iodine ($I_y$) for the three different simulations WRF-NOHET (left panel), WRF-NODEBROM (middle panel) and WRF-DEBROM (right panel) during January and February 2012. Top panels are over the subtropical area and bottom panels over the tropical. Units are in pptv.

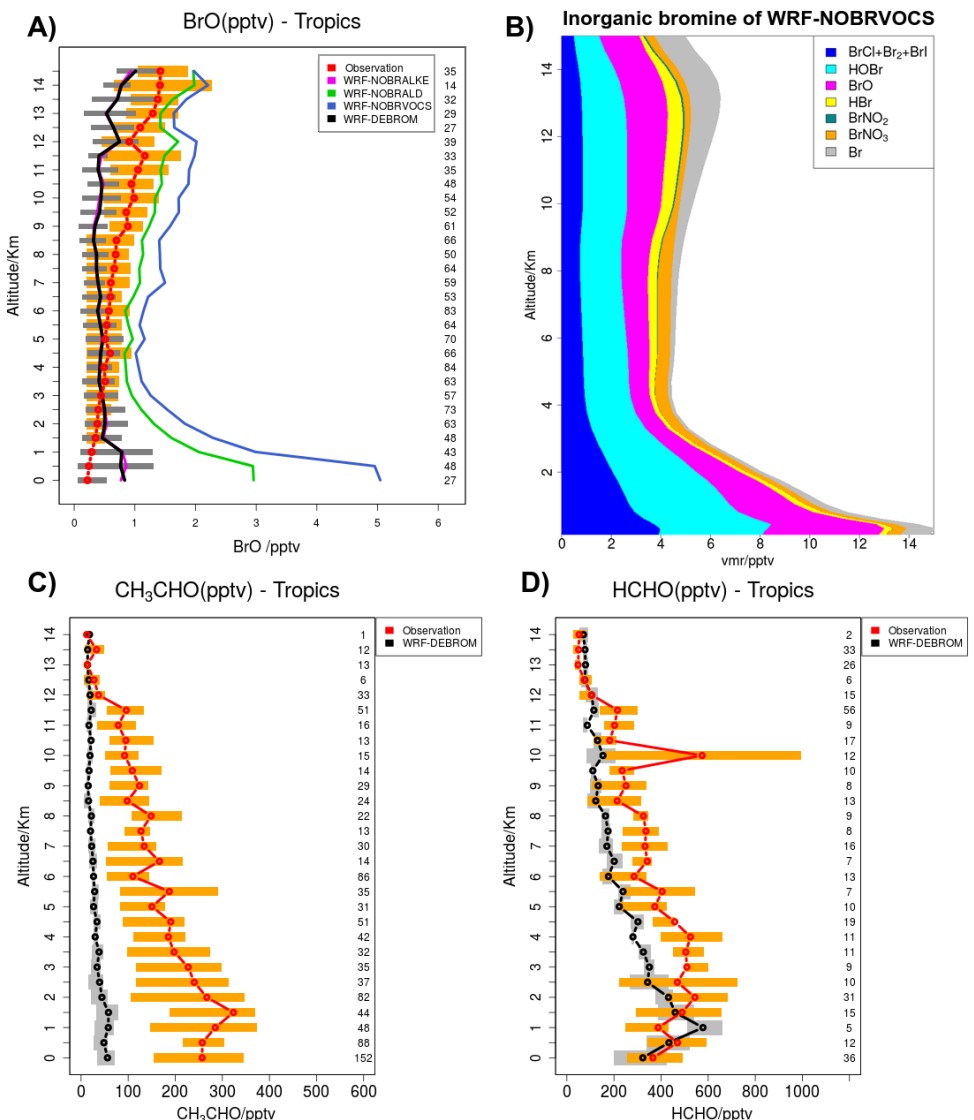

**Figure 11.** A) Mean vertical profile of BrO (pptv) over the tropics in the left panel. A sub-set of 9 flights from the TORERO campaign (red line) are compared to the 4 different WRF-Chem simulations: WRF-NOBRALKE (blue line), WRF-NOBRALD (green line), WRF-NOBRVOCS (blue line) and WRF-DEBROM (black line). B) Regional average vertical partitioning of inorganic bromine (Br$_y$) for the the WRF-NOBRVOCS run over the tropical area during January and February 2012. C) and D) the WRF-DEBROM (black line) simulation is compared with acetaldehyde and formaldehyde TORERO observations for the same 9 flights (red line). Orange and grey horizontal bars indicate the 25th-75th quartile interval for the observations of the TORERO campaign and WRF-DEBROM simulation, respectively. Values are considered in 0.5 km bin and the number of points for each altitude is given on the right side of each plot. Units are in pptv.

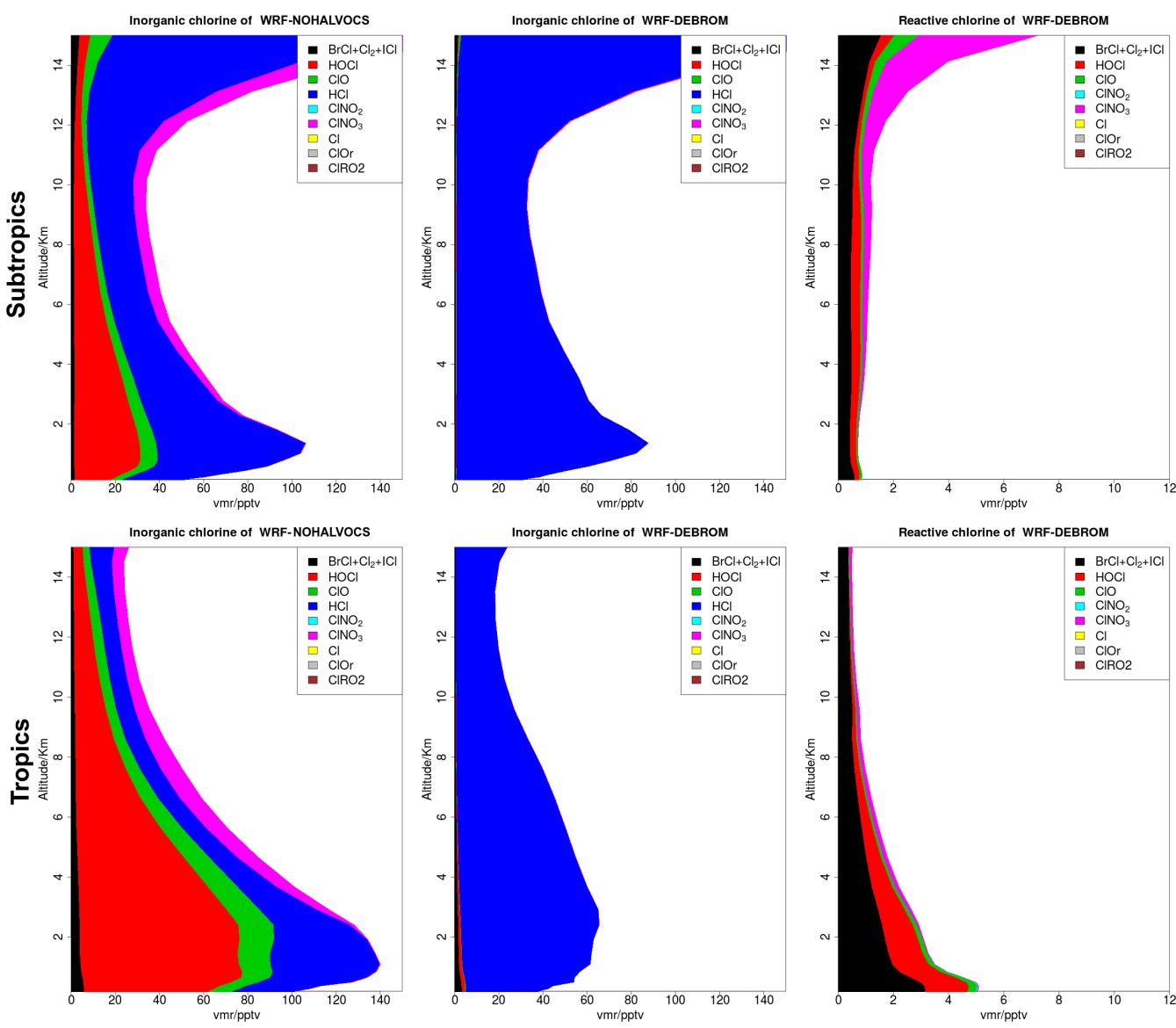

**Figure 12.** Regional average vertical partitioning of inorganic chlorine ($Cl_y$) for the two different simulations WRF-NOHALVOCS (left panel) and WRF-DEBROM (middle panel) during January and February 2012. Regional average vertical partitioning of reactive chlorine species ($Cl^*$) is also showed (right panel). $Cl^*$ is defined as $Cl_y$ gases other than HCl. Top panels are over the subtropical area and bottom panels over the tropical. Units are in pptv.

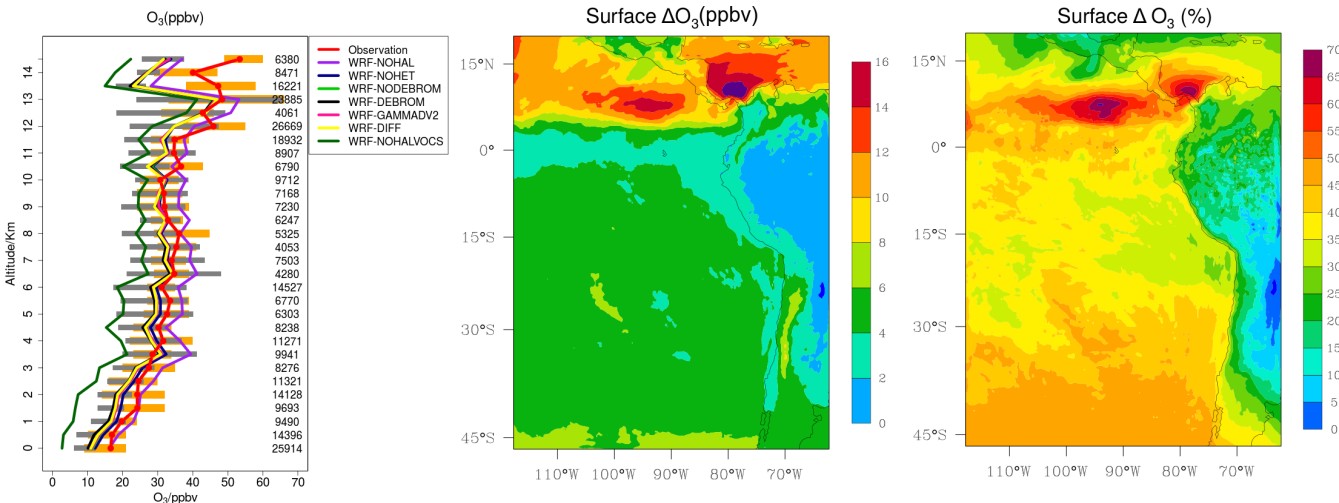

**Figure 13.** On the left, mean vertical profile of O$_3$ (ppbv) over the domain area using 13 flights from the TORERO campaign (red line) are compared to the 7 different WRF-Chem simulations: WRF-NOHAL (purple line), WRF-NOHET (blue line), WRF-NODEBROM (light green line), WRF-DEBROM (black line), WRF-GAMMADV2 (pink line), WRF-DIFF (yellow line) and WRF-NOHALVOCS (dark green line). Orange and grey horizontal bars indicate the 25th-75th quartile interval for the observations of the TORERO campaign and WRF-DEBROM simulation, respectively. Values are considered in 0.5 km bins and the numbers of aircraft measurement points for each altitude are given on the right side of each plot. On the middle and right, mean O$_3$ difference between the simulation with no halogen chemistry (WRF-NOHAL) and with halogen chemistry (WRF-DEBROM) for January and February 2012. Surface mean bias (ppbv) is shown in the left panel and surface relative mean bias (%) in the right panel. Relative mean bias (%) is calculated as (WRF-NOHAL - WRF-DEBROM)/WRF-NOHAL $\times$ 100.

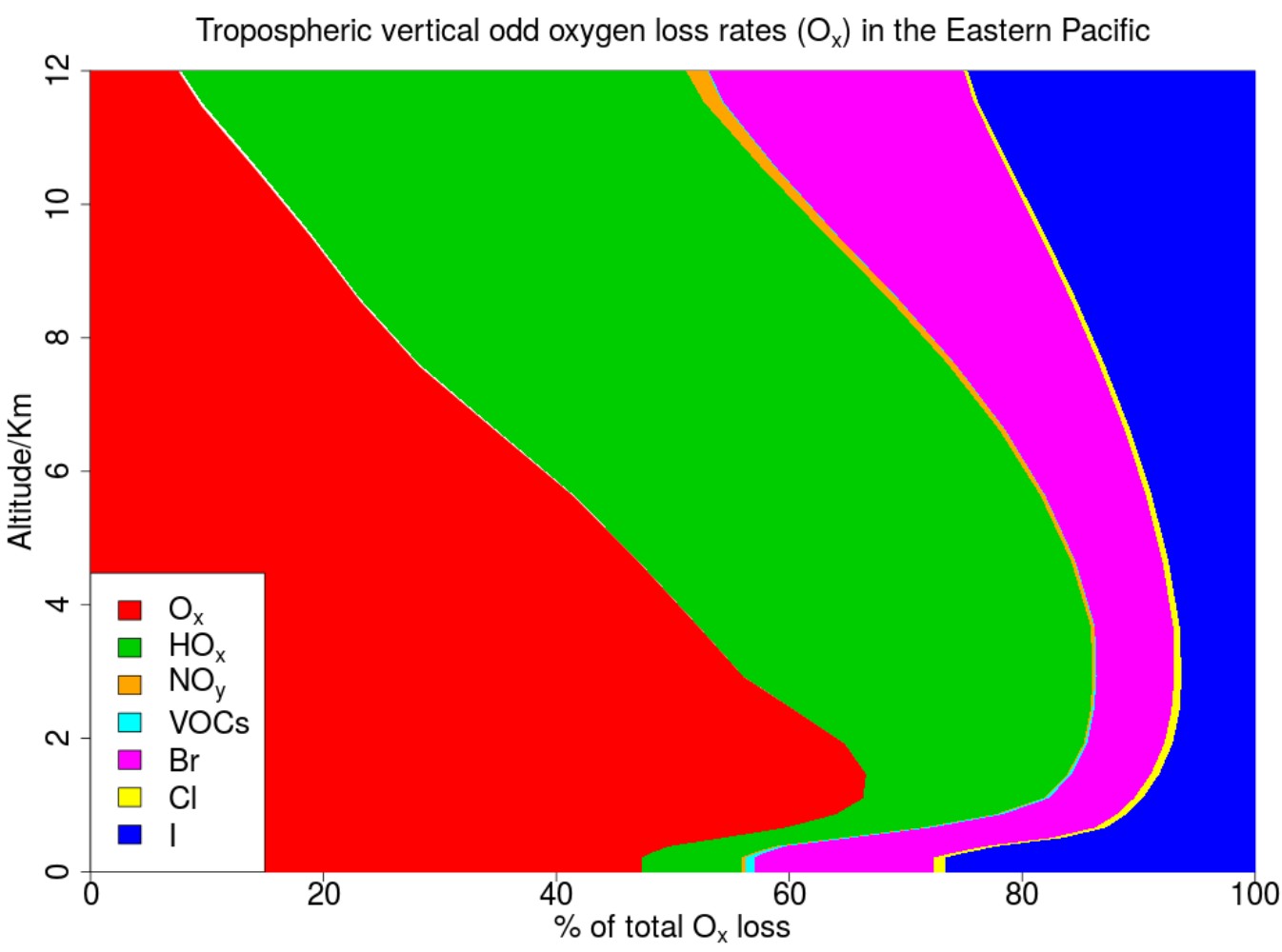

**Figure 14.** Regional average percentage contribution of each ozone depleting family to the total tropospheric vertical odd oxygen loss ($O_x$) for the WRF-DEBROM simulation.

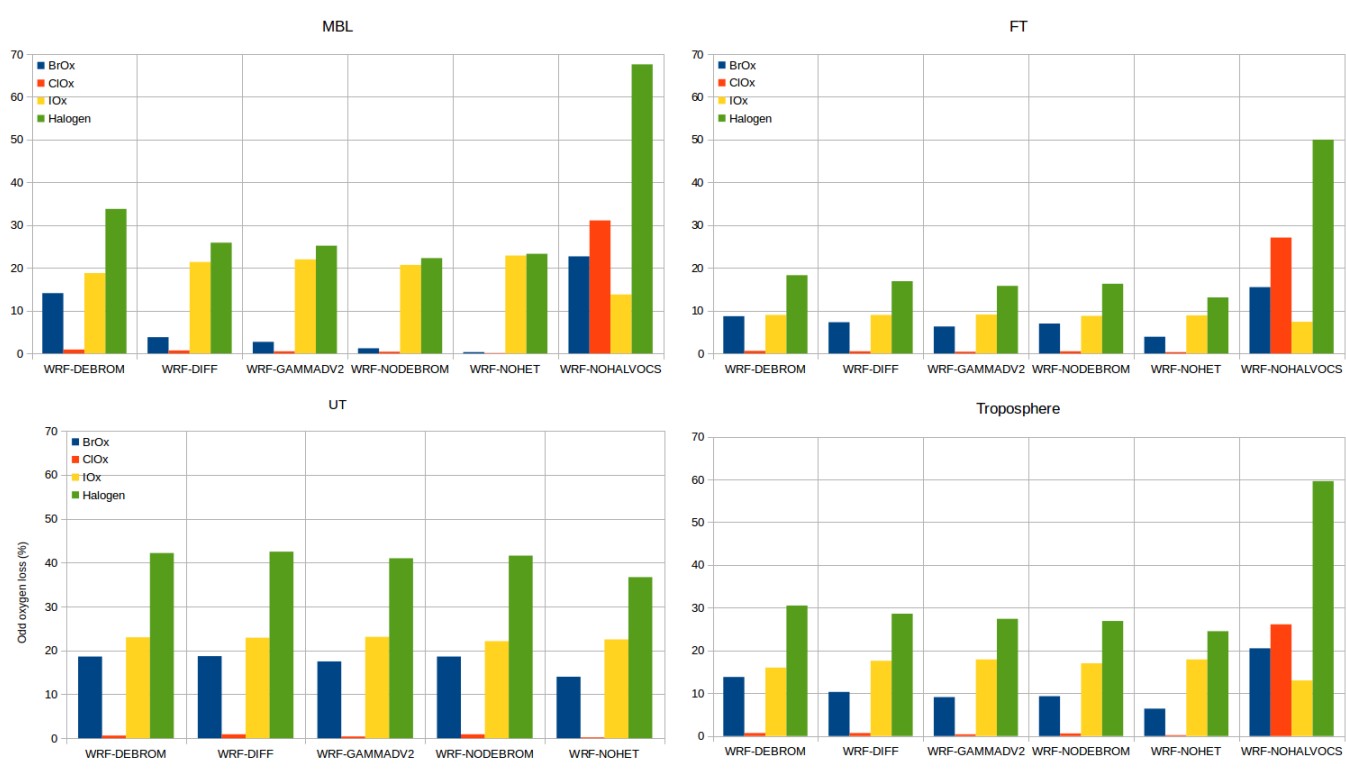

**Figure 15.** Integrated odd oxygen loss rates for each O₃ depleting halogen family within the troposphere at different altitude levels: MBL (surface - 900 hPa), FT (900 hPa- 350 hPa), UT (350 hPa- Tropopause) and troposphere (surface-Tropopause) for the WRF-DEBROM, WRF-GAMMADV2, WRF-DIFF, WRF-NODEBROM, WRF-NOHET and WRF-NOHALVOCS simulation.

**Table 1.** Henry's Law constant for relevant halogen species implemented in WRF-Chem. $INO_2$ Henry's law constant is assumed equal to that of $BrNO_2$. Iodine oxides ( $I_2O_X$ ) Henry's law constants are assumed to be infinity by analogy with $INO_3$. Virtually infinity solubility is represented by using a very large number ($2.69 \times 10^{15}$).

| Species | Henry's Law Constant (H) at 298K (M atm$^{-1}$) | $\frac{d(lnH)}{d(1/T)}$ (K) | Reference |
|---|---|---|---|
| $ClNO_3$ | $\infty$ | - | Sander (2015) |
| $BrNO_3$ | $\infty$ | - | Sander (2015) |
| $INO_3$ | $\infty$ | - | Sander et al. (2006) |
| $HOCl$ | $6.5 \times 10^2$ | 5900 | Sander (2015) |
| $HOBr$ | $1.9 \times 10^3$ | - | Sander (2015) |
| $HOI$ | $4.5 \times 10^2$ | - | Sander et al. (2011a) |
| $HCl^a$ | $7.1 \times 10^{15}$ | 5900 | Sander (2015) |
| $HBr^a$ | $7.5 \times 10^{13}$ | 10200 | Frenzel et al. (1998); Schweitzer et al. (2000) |
| $HI^a$ | $7.43 \times 10^{13}$ | 3190 | Sander (2015); Sander et al. (2006) |
| $BrCl$ | 0.9 | 5600 | Sander (2015) |
| $IBr$ | $2.4 \times 10^1$ | - | Sander (2015) |
| $ICl$ | $1.1 \times 10^2$ | - | Sander (2015) |
| $BrNO_2$ | $3. \times 10^{-1}$ | - | Sander (2015) |
| $ClNO_2$ | $4 \times 10^{-2}$ | - | Sander (2015) |
| $INO_2$ | $3. \times 10^{-1}$ | - | see caption text |
| $I_2$ | 2.63 | 4600 | Sander (2015) |
| $Br_2$ | 0.8 | 4000 | Sander (2015) |
| $I_2O_2$ | $\infty$ | - | see caption text |
| $I_2O_3$ | $\infty$ | - | see caption text |
| $I_2O_4$ | $\infty$ | - | see caption text |

[a] Effective Henry's law of HX is calculated for acid conditions (pH = 4.5): $K_H^*(T) = K_H(T) \times (1 + \frac{K_a}{[H^+]})$, where X= Cl, I or Br and $K_a = 1 \times 10^9$M is the acid dissociation constant (Bell, 1973).

**Table 2.** Bimolecular and thermal decomposition halogen reactions included in WRF-Chem. These reactions are given in the Arrhenius form with the rate equal to $A \times e^{\frac{-Ea}{RT}}$.

| Reactions | A (cm$^3$ molecules$^{-1}$ s$^{-1}$) | $\frac{Ea}{R}$ (K) | Reference |
|---|---|---|---|
| Cl + O$_3$ → ClO + O$_2$ | 2.8×10$^{-11}$ | 250 | Atkinson et al. (2007) |
| Cl + HO$_2$ → HCl + O$_2$ | 7.8×10$^{-11}$ | 620 | Atkinson et al. (2007) |
| Cl + HO$_2$ → ClO + OH | 7.8×10$^{-11}$ | 620 | Atkinson et al. (2007) |
| Cl + H$_2$O$_2$ → HCl + HO$_2$ | 1.1×10$^{-11}$ | 980 | Atkinson et al. (2007) |
| Cl + H$_2$ + O$_2$ → HCl + HO$_2$ | 3.9×10$^{-11}$ | 2310 | Atkinson et al. (2007) |
| ClO + OH → Cl + HO$_2$ | 6.8×10$^{-12}$ | -300 | Atkinson et al. (2007) |
| ClO + OH → HCl | 4.38×10$^{-13}$ | -300 | Atkinson et al. (2007) |
| ClO + HO$_2$ → HOCl | 2.2×10$^{-12}$ | -340 | Atkinson et al. (2007) |
| ClO + O$_3$ → Cl + 2O$_2$ | 1.5×10$^{-17}$ | - | Atkinson et al. (2007) |
| ClO + NO → Cl + NO$_2$ | 6.2×10$^{-12}$ | -295 | Atkinson et al. (2007) |
| HCl + OH → Cl + H$_2$O | 1.7×10$^{-12}$ | 230 | Atkinson et al. (2007) |
| HOCl + OH → ClO + H$_2$O | 3.0×10$^{-12}$ | 500 | Burkholder et al. (2015) |
| Cl + ClNO$_3$ → Cl$_2$ + NO$_3$ | 6.2×10$^{-12}$ | -145 | Atkinson et al. (2007) |
| ClNO$_3$ + OH → 0.5ClO + 0.5HNO$_3$ + 0.5HOCl + 0.5NO$_3$ | 1.2×10$^{-12}$ | 330 | Atkinson et al. (2007) |
| ClNO$_2$ + OH → HOCl + NO$_2$ | 2.4×10$^{-12}$ | 1250 | Sander et al. (2011b) |
| ClO + ClO → Cl$_2$ + O$_2$ | 1.0×10$^{-12}$ | 1590 | Sander et al. (2011b) |
| ClO + ClO → OClO + Cl | 3.5×10$^{-13}$ | 1370 | Sander et al. (2011b) |
| ClO + ClO → 2Cl | 3.0×10$^{-11}$ | 2450 | Sander et al. (2011b) |
| Cl + CH$_4$ → HCl + CH$_3$O$_2$ | 6.6×10$^{-12}$ | 1240 | Atkinson et al. (2006) |
| Cl + CH$_2$O → HCl + HO$_2$ + CO | 8.1×10$^{-11}$ | 34 | Atkinson et al. (2006) |
| Cl + CH$_3$CHO → HCl + CH$_3$CO$_3$ | 8.0×10$^{-11}$ | - | Atkinson et al. (2006) |
| Cl + CH$_3$OH → HCl + HO$_2$ + CH$_2$O | 5.5×10$^{-11}$ | - | Atkinson et al. (2006) |
| Cl + CH$_3$OOH → HCl + CH$_3$O$_2$ + OH | 5.7×10$^{-11}$ | - | Atkinson et al. (2006) |
| Cl + CH$_3$O$_2$ → 0.5CH$_2$O + 0.5CO + 0.5H$_2$O + 0.5HO$_2$ + 0.5HCl+ 0.5ClO | 1.6×10$^{-10}$ | - | Burkholder et al. (2015) |
| ClO + CH$_3$O$_2$ → Cl + CH$_2$O + HO$_2$ | 3.3×10$^{-12}$ | 115 | Atkinson et al. (2008) |
| Cl + C$_2$H$_6$ (+ O$_2$) → HCl + C$_2$H$_5$O$_2$ | 7.2×10$^{-11}$ | 70 | Sander et al. (2011b) |
| Cl + C$_3$H$_8$ (+ O$_2$) → HCl + C$_3$H$_7$O$_2$ | 7.85×10$^{-11}$ | 80 | Sander et al. (2011b) |
| Cl + C$_3$H$_6$ (+ O$_2$) → HCl + PO2 | 3.6×10$^{-12}$ | | Sander et al. (2011b) |
| CH$_3$Cl + Cl → HO$_2$ + CO + 2HCl | 3.20×10$^{-11}$ | 1250 | Sander et al. (2011b) |
| CH$_3$Cl + OH → Cl + H$_2$O + HO$_2$ | 2.40×10$^{-12}$ | 12509 | Sander et al. (2011b) |

| Reactions | A (cm$^3$ molecules$^{-1}$ s$^{-1}$) | $\frac{Ea}{R}$ (K) | Reference |
|---|---|---|---|
| Br + O$_3$ → BrO + O$_2$ | 1.7×10$^{-11}$ | 800 | Atkinson et al. (2007) |
| Br + HO$_2$ → HBr + O$_2$ | 7.7×10$^{-12}$ | 450 | Atkinson et al. (2007) |
| BrO + OH → Br + HO$_2$ | 1.8×10$^{-11}$ | -250 | Atkinson et al. (2007) |
| BrO + HO$_2$ → HOBr + O$_2$ | 4.5×10$^{-12}$ | -500 | Atkinson et al. (2007) |
| BrO + NO → Br + NO$_2$ | 8.7×10$^{-12}$ | -260 | Atkinson et al. (2007) |
| BrO + BrO → 2Br + O$_2$ | 2.4×10$^{-12}$ | -40 | Sander et al. (2011b) |
| BrO + BrO → Br$_2$ + O$_2$ | 2.8×10$^{-14}$ | -840 | Sander et al. (2011b) |
| HBr + OH → Br + H$_2$O | 6.7×10$^{-12}$ | -155 | Atkinson et al. (2007) |
| BrNO$_3$ + Br → Br$_2$ + NO$_3$ | 4.9×10$^{-11}$ | - | Orlando and Tyndall (1996) |
| Br + NO$_3$ → BrO + NO$_2$ | 1.6×10$^{-11}$ | - | Sander et al. (2011b) |
| Br$_2$ + OH → HOBr + Br | 2.1×10$^{-11}$ | -240 | Sander et al. (2011b) |
| Br + CH$_2$O → HBr + CO + HO$_2$ | 1.7×10$^{-11}$ | 800 | Sander et al. (2011b)) |
| Br + CH$_3$CHO → HBr + CH$_3$CO$_3$ | 1.8×10$^{-11}$ | 460 | Atkinson et al. (2006) |
| Br + C$_2$H$_4$ (+ O$_2$) → BrRO$_2$ | 1.3×10$^{-13}$ | - | Atkinson et al. (2006) |
| Br + C$_3$H$_6$ (+ O$_2$) → BrRO$_2$ | 3.6×10$^{-12}$ | - | Atkinson et al. (2006) |
| BrRO$_2$ + NO → 0.2 HBr + 0.8 Br + CH$_3$CO$_3$ + NO$_2$ + 0.5 CH$_2$O + HO$_2$ | 4.06×10$^{-12}$ | -360 | Toyota et al. (2004) |
| BrRO$_2$ + CH$_3$O$_2$ → 0.2 HBr + 0.8 Br + CH$_3$CO$_3$ + HO$_2$ + CH$_2$O | 1.48×10$^{-12}$ | - | Toyota et al. (2004) |
| BrRO$_2$ + HO$_2$ → BrOR + H$_2$O | 7.5×10$^{-12}$ | - | Toyota et al. (2004) |
| CH$_3$Br + OH → Br + H$_2$O+ HO$_2$ | 2.35×10$^{-12}$ | 1300 | Sander et al. (2006) |
| CH$_2$Br$_2$ + OH → 2Br | 2.0×10$^{-12}$ | 840 | Sander et al. (2006) |
| CH$_2$BrCl + OH → Br + Cl | 2.4×10$^{-12}$ | 920 | Sander et al. (2006) |
| CHBrCl$_2$ + OH → Br + 2Cl | 9.0×10$^{-13}$ | 600 | Sander et al. (2006) |
| CHBr$_2$Cl + OH → 2Br + Cl | 9.0×10$^{-13}$ | 600 | Sander et al. (2006) |
| CHBr$_3$ + OH → 3Br | 1.35×10$^{-12}$ | 600 | Sander et al. (2006) |
| I + O$_3$ → IO (+ O$_2$) | 2.1×10$^{-11}$ | 830 | Atkinson et al. (2007) |
| I + HO$_2$ → HI (+ O$_2$) | 1.5×10$^{-11}$ | 1090 | Atkinson et al. (2007) |
| I + NO$_3$ → IO + NO$_2$ | 1.0×10$^{-10}$ | - | Atkinson et al. (2007) |
| I$_2$ + OH → HOI + I | 1.8×10$^{-10}$ | - | Burkholder et al., (2015) |
| IO + HO$_2$ → HOI (+ O$_2$) | 1.4×10$^{-11}$ | -540 | Atkinson et al. (2007) |
| IO + NO → I + NO$_2$ | 7.15×10$^{-12}$ | -300 | Atkinson et al. (2007) |
| IO + IO → I + OIO | 2.16×10$^{-11}$ | -180 | Atkinson et al. (2007) |
| IO + IO → I$_2$O$_2$ | 3.24×10$^{-11}$ | -180 | Atkinson et al. (2007) |
| OIO + NO → NO$_2$ + IO | 1.1×10$^{-12}$ | -542 | Atkinson et al. (2007) |

| Reactions | A (cm$^3$ molecules$^{-1}$ s$^{-1}$) | $\frac{Ea}{R}$ (K) | Reference |
|---|---|---|---|
| OIO + OIO → I$_2$O$_4$ | 1.5×10$^{-10}$ | - | Gómez Martín et al. (2007) |
| IO + OIO → I$_2$O$_3$ | 1.5×10$^{-10}$ | - | Gómez Martín et al. (2007) |
| I$_2$O$_2$ → IO + IO | 1×10$^{-12}$ | 9770 | Ordóñez et al. (2012) |
| I$_2$O$_2$ → OIO + I | 2.5×10$^{-14}$ | 9770 | Ordóñez et al. (2012) |
| I$_2$O$_4$ → 2OIO | 3.8×10$^{-2}$ | - | Kaltsoyannis and Plane 2008 |
| HI + OH → I + H$_2$O | 1.6×10$^{-11}$ | -440 | Atkinson et al. (2007) |
| HOI + OH → IO + H$_2$O | 5.0×10$^{-12}$ | - | Riffault et al., 2005 |
| INO$_2$ (+M) → I + NO$_2$ | 9.94×10$^{17}$ | 11859 | McFiggans et al. (2000) |
| INO$_3$ → IO + NO$_2$ | 1.1×10$^{15}$ | 12060 | Atkinson et al. (2007) |
| INO + INO → I$_2$ + 2NO | 8.4×10$^{-11}$ | 2620 | Atkinson et al. (2007) |
| INO$_2$ + INO$_2$ → I$_2$ + 2NO$_2$ | 4.7×10$^{-12}$ | 1670 | Atkinson et al. (2007) |
| I$_2$ + NO$_3$ → I + INO$_3$ | 1.5×10$^{-12}$ | | Atkinson et al. (2007) |
| INO$_3$ + I → I$_2$ + NO$_3$ | 9.1×10$^{-11}$ | 146 | Kaltsoyannis and Plane 2008 |
| IO + CH$_3$O$_2$ + O$_2$ → CH$_2$O + HO$_2$ + I + 0.5O$_2$ | 2.0×10$^{-12}$ | - | Dillon et al., 2006 |
| I + BrO → IO + Br | 1.2×10$^{-11}$ | - | Sander et al. (2011b) |
| IO + Br → I + BrO | 2.7×10$^{-11}$ | - | Bedjanina et al. 1997 |
| BrO + ClO → Br + OClO | 1.6×10$^{-12}$ | -430 | Atkinson et al. (2007) |
| BrO + ClO → Br + Cl + O$_2$ | 2.9×10$^{-12}$ | -220 | Atkinson et al. (2007) |
| BrO + ClO → BrCl + O$_2$ | 5.8×10$^{-13}$ | -170 | Atkinson et al. (2007) |
| IO + ClO → 0.33ICl + 0.67I + 0.33Cl + 0.33OClO + 0.67O$_2$ | 9.4×10$^{-13}$ | -280 | Atkinson et al. (2007) |
| IO + BrO → Br + I + 0.5O$_2$ | 3.0×10$^{-12}$ | -510 | Atkinson et al. (2007) |
| IO + BrO → Br + OIO | 1.2×10$^{-11}$ | -510 | Atkinson et al. (2007) |
| CH$_3$I + OH → I + H$_2$O + HO$_2$ | 2.9×10$^{-12}$ | 1100 | Sander et al. (2011b) |

**Table 3.** Termolecular reactions for halogens species included in WRF-Chem. The lower pressure limit rate ($K_0$) is given by $A_0 \times (\frac{T}{300})^a$. The high pressure limit ($K_\infty$) is given by $B_0 \times (\frac{300}{T})^b$. $F_c$ describes the fall of curve of the reaction described by Atkinson et al. (2007). Then the reaction rate (k) is defined as $K_0[M]/(1+\frac{K_0[M]}{K_\infty}) \times F_c^n$ and n as $(1+ (\log_{10} \frac{K_0[M]}{K_\infty})^2)^{-1}$.

| Termolecular reactions | $A_0$ (cm$^6$ molecules$^{-2}$ s$^{-1}$) | a | $B_0$ (cm$^3$ molecules$^{-1}$ s$^{-1}$) | b | $F_c$ | Reference |
|---|---|---|---|---|---|---|
| Cl + NO$_2$ $\overset{M}{\to}$ ClNO$_2$ | $1.8 \times 10^{-31}$ | -2 | $1.0 \times 10^{-10}$ | -1 | 0.6 | Sander et al. (2011b) |
| ClO + NO$_2$ $\overset{M}{\to}$ ClNO$_3$ | $1.8 \times 10^{-31}$ | -3.4 | $1.5 \times 10^{-11}$ | -1.9 | 0.4 | Sander et al. (2011b) |
| Br + NO$_2$ $\overset{M}{\to}$ BrNO$_2$ | $4.2 \times 10^{-31}$ | -2.4 | $2.7 \times 10^{-11}$ | 0.0 | 0.55 | Sander et al. (2011b) |
| BrO + NO$_2$ $\overset{M}{\to}$ BrNO$_3$ | $5.2 \times 10^{-31}$ | -3.2 | $6.9 \times 10^{-12}$ | -2.9 | 0.6 | Sander et al. (2011b) |
| I + NO $\overset{M}{\to}$ INO | $1.8 \times 10^{-32}$ | -1 | $1.7 \times 10^{-11}$ | 0.0 | 0.6 | Atkinson et al. (2007) |
| I + NO$_2$ $\overset{M}{\to}$ INO$_2$ | $3.0 \times 10^{-31}$ | -1 | $6.6 \times 10^{-11}$ | 0.0 | 0.63 | Atkinson et al. (2007) |
| IO + NO$_2$ $\overset{M}{\to}$ INO$_3$ | $7.7 \times 10^{-31}$ | -5 | $1.6 \times 10^{-11}$ | 0.0 | 0.4 | Atkinson et al. (2007) |

**Table 4.** Photolytic reactions of halogens included in WRF-Chem.

| Photolysis reactions |
| --- |
| $Cl_2 \xrightarrow{h\nu} 2\ Cl$ |
| $OClO\ (+O_2) \xrightarrow{h\nu} O_3 + ClO$ |
| $HOCl \xrightarrow{h\nu} Cl + OH$ |
| $ClNO_2 \xrightarrow{h\nu} Cl + NO_2$ |
| $ClNO_3 \xrightarrow{h\nu} Cl + NO_3$ |
| $ClNO_3 \xrightarrow{h\nu} ClO + NO_2$ |
| $Br_2 \xrightarrow{h\nu} 2\ Br$ |
| $BrO \xrightarrow{h\nu} Br\ (+O_3)$ |
| $HOBr \xrightarrow{h\nu} Br + OH$ |
| $BrNO_2 \xrightarrow{h\nu} Br + NO_2$ |
| $BrNO_3 \xrightarrow{h\nu} Br + NO_3$ |
| $BrNO_3 \xrightarrow{h\nu} BrO + NO_2$ |
| $I_2 \xrightarrow{h\nu} 2\ I$ |
| $IO\ (+O_2) \xrightarrow{h\nu} I\ (+O_3)$ |
| $I_2O_4 \xrightarrow{h\nu} OIO + OIO$ |
| $OIO \xrightarrow{h\nu} I\ (+O2)$ |
| $I_2O_2 \xrightarrow{h\nu} I + OIO$ |
| $HOI \xrightarrow{h\nu} I + OH$ |
| $INO \xrightarrow{h\nu} I + NO$ |
| $INO_2 \xrightarrow{h\nu} I + NO_2$ |
| $INO_3 \xrightarrow{h\nu} I + NO_3$ |
| $I_2O_3 \xrightarrow{h\nu} OIO + IO$ |
| $IBr \xrightarrow{h\nu} I + Br$ |
| $ICl \xrightarrow{h\nu} I + Cl$ |
| $BrCl \xrightarrow{h\nu} Br + Cl$ |
| $CHBr_3\ (+O_2) \xrightarrow{h\nu} 3\ Br$ |
| $CH_3Br \xrightarrow{h\nu} Br + CH_3O_2$ |
| $CH_2Br_2 \xrightarrow{h\nu} 2Br$ |
| $CH_2BrCl \xrightarrow{h\nu} Br + Cl$ |
| $CHBrCl_2 \xrightarrow{h\nu} Br + 2\ Cl$ |
| $CHBr_2Cl \xrightarrow{h\nu} 2Br + Cl$ |
| $CH_2I_2 + (O_2) \xrightarrow{h\nu} 2\ I$ |
| $CH_3I \xrightarrow{h\nu} I + CH_3O_2$ |
| $CH_2ClI \xrightarrow{h\nu} I + Cl + 2\ HO_2 + CO$ |
| $CH_2IBr \xrightarrow{h\nu} Br + I$ |

**Table 5.** Halogen heterogeneous reactions added to WRF-Chem in this study.

| Heterogeneous reactions | Note | Uptake coefficient |
|---|---|---|
| $INO_3 \rightarrow 0.5\ IBr + 0.5\ ICl + HNO_3$ | Sea salt only if pH < 5.5 | 0.01 |
| $INO_3 \rightarrow 0.5\ I_2 + HNO_3$ | Sea salt only if pH > 5.5 | 0.01 |
| $INO_2 \rightarrow 0.5\ IBr + 0.5\ ICl + HNO_3$ | Sea salt only if pH < 5.5 | 0.02 |
| $INO_2 \rightarrow 0.5\ I_2 + HNO_3$ | Sea salt only if pH > 5.5 | 0.02 |
| $HOI \rightarrow 0.5\ IBr + 0.5\ ICl$ | Sea salt only if pH < 5.5 | 0.06 |
| $HOI \rightarrow 0.5\ I_2$ | Sea salt only if pH > 5.5 | 0.06 |
| $BrNO_3 \rightarrow 0.6\ Br_2 + HNO_3$ | Sea salt only if pH < 5.5 | 0.08 |
| $BrNO_2 \rightarrow 0.6\ Br_2 + HNO_3$ | Sea salt only if pH < 5.5 | 0.04 |
| $HOBr \rightarrow 0.6\ Br_2$ | Sea salt only if pH < 5.5 | 0.1 |
| $I_2O_2 \rightarrow I(aerosol)$ | | 0.02 |
| $I_2O_3 \rightarrow I(aerosol)$ | | 0.02 |
| $I_2O_4 \rightarrow I(aerosol)$ | | 0.02 |
| $ClNO_3 \rightarrow HOCl + HNO_3$ | Hydrolysis | $0.001^a/0.01^b$ |
| $BrNO_3 \rightarrow HOBr + HNO_3$ | Hydrolysis | $0.03^a/0.8^b$ |
| $ClNO_3 + HCl \rightarrow Cl_2 + HNO_3$ | | 0.1 |
| $ClNO_3 + HBr \rightarrow BrCl + HNO_3$ | | 0.1 |
| $HOBr + HBr \rightarrow Br_2 + H_2O$ | | 0.1 |
| $HOBr + HCl \rightarrow BrCl + H_2O$ | | 0.1 |

[a] Uptake coefficient for moderate temperature

[b] Uptake coefficient for cold temperatures

**Table 6.** Model details and experiment configuration

| Chemistry | |
|---|---|
| Chemical mechanism | MOZART-4 (Emmons et al., 2010; Knote et al., 2014) |
| Halogen chemical mechanism | MISTRA (Sommariva and von Glasow, 2012) |
| Photolysis scheme | FTUV (Tie et al., 2003) |
| Dry deposition | Wesely (1989) |
| Wet deposition | Grell and Dévényi (2002) |
| Biogenic emissions | MEGAN (Guenther et al., 2006) |
| Halocarbons and OVOCs air-sea fluxes | Online calculation (Liss and Slater, 1974; Johnson, 2010) |
| Alkenes and alkanes oceanic emissions | POET (Granier et al., 2005) |
| Sea-salt emissions | $seas\_opt = 4$, Archer-Nicholls et al. (2014) |
| $N_2O_5$ heterogeneous chemistry | $n2o5\_hetchem = 2$, Lowe et al. (2015) |
| Resolution and Initial conditions | |
| Horizontal resolution | 30 km x 30 km |
| Vertical layers | 30 or 52 |
| Top of the atmosphere | 50 hPa |
| Chemical initial condition | GEOS-Chem (Sherwen et al., 2016b) |
| Meteorological initial condition | Era-Interim (Dee et al., 2011) |
| Chemistry spin-up | 20 days |

**Table 7.** Summary of all the simulations to investigate the main processes involving reactions between halogen chemistry.

| Simulation name | Oceanic fluxes | Debromination | Heterogeneous | Br-Alkenes | Br-Aldehydes | Cl-VOCs | Halogens |
|---|---|---|---|---|---|---|---|
| WRF-DEBROM | Online | √ | √ | √ | √ | √ | √ |
| WRF-GAMMADV2 | Online | √ ($\gamma$ divided by 2) | √ ($\gamma$ divided by 2) | √ | √ | √ | √ |
| WRF-DIFF | Online | √ (uses Eq. 3) | √ (uses Eq. 3) | √ | √ | √ | √ |
| WRF-ZIS | Prescribed | √ | √ | √ | √ | √ | √ |
| WRF-NODEBROM | Online | | √ | √ | √ | √ | √ |
| WRF-NOHET | Online | | | √ | √ | √ | √ |
| WRF-NOBRALKE | Online | √ | √ | | √ | √ | √ |
| WRF-NOBRALD | Online | √ | √ | √ | | √ | √ |
| WRF-NOBRVOCS | Online | √ | √ | | | √ | √ |
| WRF-NOHALVOCS | Online | √ | √ | | | | √ |
| WRF-NOHAL | - | | | | | | |

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
