# Peer review of "Importance of reactive halogens in the tropical marine atmosphere: A regional modelling study using WRF-Chem"

_Atmospheric Chemistry and Physics, 2017_

## Referee Comment (RC1) · Anonymous Referee #2 · 12 Dec 2017

The manuscript attempts to compare observations of reactive halogen species (containing Cl, Br, I) in the Pacific with model results. Comparison of model results with real observations are always important and n particular provide a test of the degree of our understanding of the modelled part of the reality. As such the authors have to be commended for their undertaking. The outcome of the inter comparison is the (not entirely new) finding that our knowledge of the relevant halogen chemistry and the source mechanisms of halogen species is very limited. Probably a meaningful modelling of abundance and distribution of reactive halogen species in the troposphere is not even possible at present. Nevertheless the modelling effort appears to be a useful exercise, however it is important to more clearly state what can be accomplished with models

at present and what not. In particular it would be very helpful if the following could be stated:

1) What are the real uncertainties in the model results taking into account our knowledge gaps. And what is the uncertainty of stated numbers (e.g. the tropospheric O3 loss due to halogen reactions and its vertical distribution as shown in Fig. 13). The real uncertainties are clearly much larger than the differences between the various scenarios (given in Table 7).

2) Where are the most important gaps in our knowledge, both in the area of input data (e.g. precursor fluxes) and regarding mechanistic data.

3) What could be done in the future to improve the situation

In detail there are many open questions and points to improve as shown in the (incomplete) numbered list below. For instance what is the Definitions of 'halogens' in the text (e.g. in the 1st line of the abstract certainly not just halogen molecules or halogen atoms are meant) and of 'reactive halogen' and 'Bry'.

Overall I feel that this manuscript has the potential to become an important contribution to our understanding of tropospheric halogen chemistry once the authors had a chance to make the above additions and to answer the many unclear points listed below. Unfortunately due to the simplistic implementation of aerosol reactions (Equation (2), see comment 22) below) the model runs must be repeated with a state of the art parameterisation of heterogeneous aerosol reactions. This is also likely to change many results of the manuscript. In view of the many required changes and the many unclear points a very major revision of the manuscript is required.

Detailed comments:

1) Page 2, line 4: Define the term 'reactive halogens'

2) Page 2, line 5: Only Cl-atoms 'reduce the lifetime of CH4'

3) Page 2, lines 6, 7: Define the term 'halogen species', also the statement '. . . play an important role in the oxidation capacity . . .' appears to be a repetition of the statements made above.

4) Page 2, line 10: what should be the effect on 'oxidation capacity' beyond the already mentioned CH4 and particles?

5) Page 2, lines 12-15: What about fluorine? Are there not other cycles, e.g. XO self reactions and cross reactions like BrO + IO, . . .?

6) Page 2, lines 16-29: this paragraph should be edited for better structure. At present the authors talk about 'numerical models' in general then progress to global and regional models, then progress to specific models (e.g. WRF chem)

7) Page 2, line 30: Give definition of VSLH.

8) Page 3, line 1: What about the direct transport of XO?

9) Page 3, line 11: Which OVOC species do the authors have in mind?

10) Page 3, lines 15-19: Which heterogeneous reactions do the authors have in mind? Why is Cl not mentioned? How does de-bromination work? Is it different from heterogeneous Br-release through HOBr and BrNO3? Is there not also de-iodination and de-chlorination? Actually it would be good to have a list of sources of halogen atoms.

11) Page 3, lines 19-22: What does 'This source' refer to? What exactly is poorly understood? What do the authors mean with 'inconsistent levels' too high or too low?

12) Page 3, line 23: Do the authors really mean to state that all atmospheric models are largely untested or does this statement rather pertain to halogen chemistry in models?

13) Page 3, line 33: 'heterogeneous recycling reactions' of what?

14) Page 4, lines 7-9: Here it would be good to have some information on the measurement techniques used.

15) Page 4, first sentence of section 3.1: See comment 10, above.

16) Page 4, line 29: How is Ka parameterised?

17) Page 5, lines 6-8: Give examples for OVOC's emitted from the ocean.

18) Page 5, line 8: Which VOC's were included?

19) Page 5, lines 16-17: Give some details on sea-salt emission parameterisation.

20) Page 5, lines 21-25: This description is not quite clear. When 'Most of the reactions come from ... MISTRA', which ones? How is the high uncertainty of I2Ox reactions dealt with?

21) Page 5, lines 31-33: Explain 17 bins and give a few details on FTUV. It would also be good to give e.g. noontime values of the photolysis frequencies (e.g. in Table 4).

22) Page 6, Section 3.3, Equation (2) is simplistic in that it neglects diffusion limitations, which can be important at uptake coefficients higher than about 0.01, see e.g. Schwartz and Freiberg 1967. The authors should check likely changes in model results when more realistic parameterisations of heterogeneous reactions are used.

Schwartz S.E. and Freiberg J.E. (1967), Mass-transport limitation to the rate of reaction of gases in liquid droplets: Application to oxidation of SO2 in aqueous solutions, Atmospheric Environment 15, (7),1129-1144

23) Page 7, Lines 13-28: Are reactions R11 and R12 not occurring at pH > 5.5? What about mass conservation in R13? Numbers or reactions are missing in Table 5

24) Page 14, lines 15, 16: The model overestimates heterogeneous reactions on aerosol (see comment 22, above), thus the increase in the concentration of reactive species like BrO could be just a model effect.

25) Page 14, lines 19, 20: The underestimation of aldehydes could be due to neglect of Cl reactions in the model. In fact heterogeneous oxidation of halogenides (by O3) is

discussed in the text, but does not appear to be implemented in the model, since no reactions to that effect can be found in Table 5.

26) Page 14, line 22: What could be the reason for inorganic iodine emissions being lower than modelled in the tropics?

27) Page 14, lines 23, 24: What is the reason for the minor importance of heterogeneous chemistry in the model?

28) Page 14, line 25: For which conditions was the 31% contribution to O3 loss calculated?

29) Page 14, line 26: How can the Ox loss be attributed to Br, I, Cl reactions when cross reactions play a major role (e.g. ClO + BrO)?

31) Page 14, lines 27-29: What is the meaning of 'negative bias', is it an overestimation of O3 loss?

32) Page 14, lines 30-35: the reviewer can only agree with the statement in Line 30. What is the effect of these uncertainties for the conclusions drawn like 'Halogens constitute 31% of the overall ... O3 loss'? Should the conclusion not rather be 'Halogens reactions constitute a considerable fraction (10-60%) of the overall tropospheric O3 loss'?

33) Caption of Figure 5: It would be more clear if "obs" was changed to "observation". In the time series of CH3I there is a clear anti-correlation between observation and model results during Feb. 5 – 10, can this be explained?

34) Table 6: The N2O5 heterogeneous chemistry is not listed in Table 5.

---

## Referee Comment (RC2) · Anonymous Referee #1 · 19 Jan 2018

The authors have completed an enormous amount of work, but unfortunately the paper does not currently meet the requirements for publication in ACP. This paper can be published after addressing the following major comments:

- The authors must provide a justification as to why they have not included chlorine cycling in the model. Even if there are no measurements of chlorine, the authors cannot focus on bromine and iodine cycling without some description of chlorine chemistry. The authors cannot simply ignore chlorine especially in the remote marine boundary layer. See for example Schmidt et al. (2016).

- A more complete description of heterogeneous chemistry already included in the

model is needed. If the authors used the model as described, ClNO2 is formed on aerosols via reactions of N2O5. However, once formed ClNO2 is treated as an inert species. This should be updated prior to publication.

- As already pointed out by the other reviewer, Equation 2 is not an acceptable treatment for reactions on aerosols. This equation ignores the fact that reactions are limited by the rate at which gases can diffuse towards the aerosol surface (diffusion limitation). There are clear descriptions of how to treat this correctly in the literature, for example in Schwartz (1986).

- While this paper was not submitted to Geoscientific Model Development (GMD), this is the first paper that describes a new model development. The paper should be held to the GMD standard for publication. For example, the subroutines that have been updated to perform the study should be included in an electronic supplement and/or the code used in the study should be provided as an electronic supplement.

The authors have attempted to study the influence of bromine and chlorine cycling in the tropical marine atmosphere. This can be an important study and hopefully the authors will make the effort to address these major comments so that the paper can be published in ACP.

References:

Schwartz S.E. (1986) Mass-Transport Considerations Pertinent to Aqueous Phase Reactions of Gases in Liquid-Water Clouds. In: Jaeschke W. (eds) Chemistry of Multiphase Atmospheric Systems. NATO ASI Series (Series G: Ecological Sciences), vol 6. Springer, Berlin, Heidelberg.

Schmidt, J. A., et al. (2016), Modeling the observed tropospheric BrO background: Importance of multiphase chemistry and implications for ozone, OH, and mercury, J. Geophys. Res. Atmos., 121, 11,819–11,835, doi:10.1002/2015JD024229.

---

## Author Comment (AC1) · 1 Nov 2018

Review of ""Importance of reactive halogens in the tropical marine atmosphere: A regional modelling study using WRF-Chem" by Alba Badia et al.

The manuscript attempts to compare observations of reactive halogen species (containing Cl, Br, I) in the Pacific with model results. Comparison of model results with real observations are always important and n particular provide a test of the degree of our understanding of the modelled part of the reality. As such the authors have to be commended for their undertaking. The outcome of the inter comparison is the (not entirely new) finding that our knowledge of the relevant halogen chemistry and the source

mechanisms of halogen species is very limited. Probably a meaningful modelling of abundance and distribution of reactive halogen species in the troposphere is not even possible at present. Nevertheless the modelling effort appears to be a useful exercise, however it is important to more clearly state what can be accomplished with models at present and what not.

In particular it would be very helpful if the following could be stated: 1) What are the real uncertainties in the model results taking into account our knowledge gaps. And what is the uncertainty of stated numbers (e.g. the tropospheric O3 loss due to halogen reactions and its vertical distribution as shown in Fig. 13). The real uncertainties are clearly much larger than the differences between the various scenarios (given in Table 7). 2) Where are the most important gaps in our knowledge, both in the area of input data (e.g. precursor fluxes) and regarding mechanistic data. 3) What could be done in the future to improve the situation

Response: We address each of the individual points below.

In detail there are many open questions and points to improve as shown in the (incomplete) numbered list below. For instance what is the Definitions of 'halogens' in the text (e.g. in the 1st line of the abstract certainly not just halogen molecules or halogen atoms are meant) and of 'reactive halogen' and 'Bry'.

Response: "Reactive halogens species, containing Cl, Br and I" from Saiz-Lopez and von Glasow (2012). BrY is defined as inorganic bromine. These definitions are now included in the updated manuscript.

Overall I feel that this manuscript has the potential to become an important contribution to our understanding of tropospheric halogen chemistry once the authors had a chance to make the above additions and to answer the many unclear points listed below. Unfortunately due to the simplistic implementation of aerosol reactions (Equation (2), see comment 22) below) the model runs must be repeated with a state of the art parameterisation of heterogeneous aerosol reactions. This is also likely to change

many results of the manuscript. In view of the many required changes and the many unclear points a very major revision of the manuscript is required.

Response: The authors wish to thank anonymous reviewer #2 for his/her valuable comments and suggestions.

We agree with the reviewer that current understanding of the sources of halogen species and their chemistry is limited. Much of this is due to a lack of field observations with which to constrain model input and to evaluate models with. The new observations from TORERO provide a great opportunity to advance our understanding of the importance of reactive halogens in the tropical marine atmosphere and to highlight gaps in knowledge and this paper is part of that process. This contribution describes an important step in the understanding the tropospheric halogen chemistry because it is the first time that halogen chemistry (including bromine, chlorine and iodine) has been implemented into a regional online chemistry transport model. Of course there are limitations and the suggestions made by the reviewer in points 1 - 3 above are very valuable and we have made improvements to the manuscript to address them - see below.

Heterogeneous chemistry (uptake coefficients and surface aerosol area), photolysis rates (especially iodine photochemistry (Sommariva et al., 2012)), the chemical mechanism for the repartitioning of Bry resulting from the reactions of Br with VOCs, interactions between halogens and the sulfur cycle (Chen et al., 2017, Chen et al., 2018), physical constants (such as Henry's Law value for HOI), deposition of halogenated species and inputs to calculate the oceanic fluxes (e.g. sea-water concentration, including iodide concentrations) are the most important gaps in our knowledge. We have discussed these more in the revised manuscript, in particular emphasizing how the results of our study contribute to this.

We are aware that our treatment of the aerosol chemistry is a simplification, but it is an approach in-line with many other studies published since the Schwartz and Freiberg

paper in 1967. Due to computational resource limitations, large-scale models must be parameterized representations of reality. Furthermore, we are limited by the input data with which to constrain these parametrizations.

One important limitation is information with which to constrain a more detailed treatment of the heterogeneous chemistry. We have taken an approach similar to that used with other models such as TOMCAT (Yang et al., 2008), CAM-Chem (Ordoñez et al., 2012) and GU–WRF/Chem (Karamchandani et al., 2012). In this approach the heterogeneous chemistry is assumed to take place between a gas-phase species and an adsorbed species with an uptake coefficient ($\gamma$) used to calculate first-order rate constants for heterogeneous loss of the gas-phase species. There are large uncertainties in these uptake coefficients. Moreover, there are uncertainties in the ability of the model to simulate the aerosol size distribution (and thus aerosol surface area) and the mixing state and surface composition of the aerosols. In addition, there are a limited number of measurements with which to evaluate the model and these measurements have their own uncertainties. We agree that ignoring the diffusion limitation will artificially increase the heterogeneous reactions in the model, however we feel this is one of several uncertainties that will affect the rate of heterogeneous chemistry. Instead of focusing on one factor, we prefer to address the wider uncertainties in the heterogeneous chemistry through sensitivity studies in which $\gamma$ is varied.

To illustrate the sensitivity of our results to this we have made a new run assuming that $\gamma$ is reduced by a factor of 2 (WRF-GAMMADV2). Moreover, simulations WRF-NOALD, WRF-NOALK and WRF-NOVOCS have been renamed as WRF-NOBRALD, WRF-NOBRALK and WRF-NOBRVOCS. A new simulation where Br and Cl reactions with VOCS are not considered have been made and named WRF-NOHALVOCS. Some of the new results have been added in the revised manuscript (Sec. 5) and Supplement (Fig S4) and we also present them below:

Figure 1. Mean vertical profile of BrO (pptv) over the subtropics (left) and tropics (right). An average over 16 flights of the TORERO campaign (red line) are compared to

the 4 different WRF-Chem simulations: WRF-NOHET (blue line), WRF-NODEBROM (green line), WRF-DEBROM (black line) and WRF-GAMMADV2 (pink line). Orange and grey horizontal bars indicate the 25th-75th quartile interval for the observations of the TORERO campaign and WRF-DEBROM simulation, respectively. Values are considered in 0.5 km bins and the number of aircraft measurement points for each altitude are given on the right side of each plot.

Figure 2. Mean vertical profile of IO (pptv) over the subtropics (left) and tropics (right). An average over 16 flights of the TORERO campaign (red line) are compared to the 3 different WRF-Chem simulations: WRF-NOHET (blue line), WRF-NODEBROM (green line), WRF-DEBROM (black line) and WRF-GAMMADV2 (pink line). Orange and grey horizontal bars indicate the 25th-75th quartile interval for the observations of the TORERO campaign and WRF-DEBROM simulation, respectively. Values are considered in 0.5 km bins and the number of aircraft measurement points for each altitude are given on the right side of each plot.

Figure 3: Regional average vertical partitioning of inorganic chlorine (Cly) for the two different simulations WRF-NOHALVOCS (left panel) and WRF-DEBROM (middle panel) during January and February 2012. Regional average vertical partitioning of reactive chlorine species (Cl∗) for WRF-DEBROM is also showed (right panel). Cl∗ is defined as Cly gases other than HCl. Top panels are over the subtropical area and bottom panels over the tropical. Units are in pptv.

Figure 4: A) Mean vertical profile of BrO (pptv) over the tropics in the left panel. A subset of 9 flights from the TORERO campaign (red line) are compared to the 4 different WRF-Chem simulations: WRF-NOBRALKE (blue line), WRF-NOBRALD (green line), WRF-NOBRVOCS (blue line) and WRF-DEBROM (black line). B) Regional average vertical partitioning of inorganic bromine (Bry ) for the the WRF-NOBRVOCS run over the subtropical area (left panel) and over the tropical area (right panel) during January and February 2012. C) and D) the WRF-DEBROM (black line) simulation is compared with acetaldehyde and formaldehyde TORERO observations for the same flights (red

[Figure]

line). Orange and grey horizontal bars indicate the 25th-75th quartile interval for the observations of the TORERO campaign and WRF-DEBROM simulation, respectively. Values are considered in 0.5 km bin and the number of points for each altitude is given on the right side of each plot. Units are in pptv.

Figure 5. On the left, mean vertical profile of O3 (ppbv) over the domain area using 13 flights from the TORERO campaign (red line) are compared to the 6 different WRF-Chem simulations: WRF-NOHAL (purple line), WRF-NOHET (blue line), WRF-NODEBROM (light green line), WRF-DEBROM (black line), WRF-GAMMADV2 (pink line) and WRF-NOHALVOCS(dark green line). Orange and grey horizontal bars indicate the 25th-75th quartile interval for the observations of the TORERO campaign and WRF-DEBROM simulation, respectively. Values are considered in 0.5 km bins and the number of aircraft measurement points for each altitude are given on the right side of each plot.

Figure 6. Integrated odd oxygen loss rates for each O3 depleting halogen family within the troposphere at different altitude levels: MBL (surface - 900 hPa), FT (900 hPa- 350 hPa), UT (350 hPa-tropopause) and troposphere (surface-tropopause) for the WRF-DEBROM, WRF-GAMMADV2, WRF-NODEBROM, WRF-NOHET and WRF-NOHALVOCS simulation.

New text has been added to the revised manuscript that discuss the results and the uncertainty of the stated numbers, such as the tropospheric O3 loss due to halogen reactions and its vertical distribution. The main new points made are:

- In the MBL the impact of halving gamma (WRF-GAMMADV2 run) on BrO has approximately half the the impact of including heterogeneous chemistry (i.e. half the difference between the WRF-DEBROM run and the WRF-NOHET run). For areas where debromination dominates, such as the tropical MBL, the difference is greater. However, smaller effects are seen in the free troposphere with very little impact in the UT, where there is only a small decrease in BrO when gamma is halved (WRF-GAMMADV2).

[Figure]

- Changing GAMMA (difference between the WRF-DEBROM and WRF-GAMMADV2 runs) has very little impact on IO.

- O3 mixing ratios are sensitive to the inclusion of halogen chemistry (∼7 ppbv) with heterogeneous chemistry contributing up to 3 ppbv of this change in the MBL and even less throughout the free and upper troposphere. The uncertainty in the heterogeneous chemistry introduced by neglecting the diffusion limitation in equation 2 therefore has a relatively small impact in the overall role of the halogen chemistry on O3. The biggest difference (7-12 ppbv) in O3 values is seen when reactions between Br and Cl and VOCS are not considering (WRF-NOHALVOCS run) demonstrating that a large uncertainty in our understanding of the impact of halogen chemistry on O3 is related to the treatment of the VOC emissions and gas phase chemistry.

- Our model results show that the tropospheric Ox loss due to halogens ranges between 25-60%. Uncertainties in the heterogeneous chemistry accounted for a small proportion of this range (25% to 31%). The upper bound (60%) is found when reactions between Br and Cl with VOCs are not included and, consequently, Ox lost by BrOx and ClOx cycles is high (60%).

Future steps should focus on providing more data from laboratory studies along with field observations to better constrain the modelled representation of halogen species. Only a few Bry and Iy species are provided from field campaigns making it difficult to evaluate the model against observations and a very limited amount of data exist to compute the sea-water concentration for the halocarbons, giving uncertainty in this input data when the halocarbon fluxes are computed. Furthermore, it is clear that we need to understand better the emissions and distributions of VOCs in the remote atmosphere, and the impact of their chemistry on halogen repartitioning.

Yang, X.,  ăJ. A. Pyle, and ăR. A. Cox ă(2008),  ăSea salt aerosol production and bromine release: Role of snow on sea ice,  ăGeophys. Res. Lett.,  ă35, L16815, doi:10.1029/2008GL034536.

[Figure]

Ordóñez, C., Lamarque, J.-F., Tilmes, S., Kinnison, D. E., Atlas, E. L., Blake, D. R., Sousa Santos, G., Brasseur, G., and Saiz-Lopez, A.: Bromine and iodine chemistry in a global chemistry-climate model: description and evaluation of very short-lived oceanic sources, Atmos. Chem. Phys., 12, 1423-1447, https://doi.org/10.5194/acp-12-1423-2012, 2012.

Karamchandani et al., 2012: Development of an extended chemical mechanism for global-through-urban applications, Atmospheric Pollution Research 3 (2012), doi:10.5094/APR.2011.047

Chen,ÂăQ.,ÂăJ. A. Schmidt,ÂăV. Shah,ÂăL. Jaeglé,ÂăT. Sherwen, andÂăB. Alexander (2017),ÂăSulfate production by reactive bromine: Implications for the global sulfur and reactive bromine budgets,ÂăGeophys. Res. Lett.,Âă44, 7069–7078, doi:Âă. Chen, Q., Sherwen, T., Evans, M., and Alexander, B.: DMS oxidation and sulfur aerosol formation in the marine troposphere: a focus on reactive halogen and multiphase chemistry, Atmos. Chem. Phys., 18, 13617-13637, https://doi.org/10.5194/acp-18-13617-2018, 2018.

R. Sommariva, W.J. Bloss, R. von Glasow, Uncertainties in gas-phase atmospheric iodine chemistry, Atmospheric Environment, Volume 57, 2012, 219-232, 1352-2310,https://doi.org/10.1016/j.atmosenv.2012.04.032.

Detailed comments: 1) Page 2, line 4: Define the term 'reactive halogens'

Response: "Reactive halogens species, containing Cl, Br and I" from Saiz-Lopez and von Glasow (2012). This definition is now included in the revised manuscript (1st line of the abstract).

A. Saiz-Lopez and R. von Glasow. Reactive halogen chemistry in the troposphere. Chem. Soc. Rev., 41:6448–6472, 2012. doi:10.1039/C2CS35208G. URL http://dx.doi.org/10.1039/C2CS35208G.

2) Page 2, line 5: Only Cl-atoms 'reduce the lifetime of $CH_4$'

Response: Amended. The manuscript has been changed.

3) Page 2, lines 6, 7: Define the term 'halogen species', also the statement '. . . play an important role in the oxidation capacity . . .' appears to be a repetition of the statements made above.

Response: Yes it is a repetition, so this sentence has been removed from the revised manuscript.

4) Page 2, line 10: what should be the effect on 'oxidation capacity' beyond the already mentioned CH4 and particles?

Response: We do not understand the reviewer's question. In the paragraph that the reviewer points to we already briefly explain how halogen chemistry affects the budgets of key oxidants in the atmosphere in order to provide relevance and justification for this study. The original paragraph with minor modifications in response to other comments is copied below and we cannot see what additional information the reviewer is asking for.

"Reactive halogens cause ozone ($O_3$) destruction, change the HOX ($HO_2$ + OH) and NOX ($NO_2$ + NO) partitioning, affect the oxidation of VOCs and mercury, and take part in new particle formation (Chameides and Davis, 1980; von Glasow et al., 2004; Saiz-Lopez and von Glasow, 2012). Moreover, reactive chlorine reduces the lifetime of methane ($CH_4$). Halogen species are known to play an important role in the oxidising capacity of the troposphere. The atmospheric oxidation capacity is to a large extent determined by the budgets of the hydroxyl radical (OH) and $O_3$; globally most tropospheric OH is found in the tropics (Bloss et al., 2005). Therefore a quantitative understanding of the composition and chemistry of the tropical marine atmosphere is essential to examine the atmospheric oxidative capacity and climate forcing."

5) Page 2, lines 12-15: What about fluorine? Are there not other cycles, e.g. XO self reactions and cross reactions like BrO + IO, . . .?

[Figure]

Response: We don't consider fluorine chemistry in the model.

"Fluorine atoms are so reactive that they form HF very rapidly and play no role in tropospheric chemistry." from Saiz-Lopez and von Glasow (2012).

A. Saiz-Lopez and R. von Glasow. Reactive halogen chemistry in the troposphere. Chem. Soc. Rev., 41:6448–6472, 2012. doi:10.1039/C2CS35208G. URL http://dx.doi.org/10.1039/C2CS35208G.

6) Page 2, lines 16-29: this paragraph should be edited for better structure. At present the authors talk about 'numerical models' in general then progress to global and regional models, then progress to specific models (e.g. WRF chem)

Response: This paragraph has been rewritten for better structure.

7) Page 2, line 30: Give definition of VSLH.

Response: Very Short Lived Halocarbons (VSLH) are defined as trace gases whose chemical lifetimes are generally under six months. This information is now included in the revised manuscript.

8) Page 3, line 1: What about the direct transport of XO?

Response: The lifetime of XO radicals is sufficiently short, ranging from minutes to one hour. Thus, direct transport of XO is not considered.

9) Page 3, line 11: Which OVOC species do the authors have in mind?

Response: Basically aldehydes. Aldehydes species that have been included in the model are formaldehydes and acetaldehydes. This is described in section 3.2.1.

10) Page 3, lines 15-19: Which heterogeneous reactions do the authors have in mind? Why is Cl not mentioned?

Response: Heterogeneous reactions involving reactive halogen species are included in the model. This is now clarified in the revised manuscript. Cl is not mention here

because we try to explain observations of BrO and IO and there are no reported direct observations of Cl.

How does debromination work? Is it different from heterogeneous Br-release through HOBr and BrNO3? Is there not also de-iodination and de-chlorination? Actually it would be good to have a list of sources of halogen atoms.

Response: Debromination is the process by which sea-salt aerosols are depleted in bromide (Br-) through the release of Br- into the gas-phase. It occurs through the uptake of a gaseous species into the sea-salt and the subsequent reaction with Br- (Yang et al, 2005). We represent this as release of Br through heterogeneous reactions that involve uptake of gases species including HOBr and BrNO3. We described this in more detail on page 7. We changed the text on pages 3 and 7 to make it clearer what we mean by debromination:

Page 3: "Another source of reactive inorganic bromine in the troposphere is the release of bromide (Br-) from sea-salt aerosols into the gas-phase. This is known as debromination and occurs through the uptake of a gaseous species in the sea-salt and the subsequent reaction with Br-. Page 7. "It is known that the chemistry involving the release of bromine from the sea-salt aerosol (debromination) is strongly pH dependent, being more efficient for acidified aerosol especially with a pH < 5.5 (Keene et al., 1998). Therefore, the pH value of the aerosol particles is calculated in the model for each size bin (see Zaveri et al. (2008) for further description of the pH calculation). We then apply a pH dependence to the heterogeneous reactions that occur on the surface of the sea-salt. When the pH < 5.5 debromination reactions occur with the release of Br2 and IBr resulting from the uptake of BrNO3, BrNO2, HOBr, INO3, INO2 and HOI (R11-R16). When the pH > 5.5 no debromination reactions occur, although uptake of INO3, INO2 and HOI on the sea-salt still occurs (R17-R19)".

Whilst the processes of deiodination and dechlorination do occur, they are not thought to be important sources of atmospheric I- and Cl- so we do not include them in the

model.

List of sources of halogen atoms considered in this study: oceanic source of organic halogens (CHBr3, CH2Br2, CH3I, CH2BrCl, CHBrCl2, CHBr2Cl, CH2I2, CH2IBr and CH2ICl), debromination (Br2, IBr) and inorganic I from the ocean (HOI and I2).

11) Page 3, lines 19-22: What does 'This source' refer to?

Response: This 'source' refers to the 'source' mention in the previous sentence i.e. 'debromination', the last word in the previous sentence. However, with the rewording made to address the previous point "This source has been included" has been replaced with "Debromination has been included as a source of gas-phase bromine".

What exactly is poorly understood?

Response: With the rewording of the previous sentence "this process" now clearly refers to debromination.

What do the authors mean with 'inconsistent levels' too high or too low?

Response: The authors mean inconsistent high levels of bromine species. This is now clarified in the revised manuscript.

12) Page 3, line 23: Do the authors really mean to state that all atmospheric models are largely untested or does this statement rather pertain to halogen chemistry in models?

Response: This statement pertains to halogen chemistry in models. We have rewritten the start of this sentence in the revised manuscript: "Halogen chemistry in atmospheric models...".

13) Page 3, line 33: 'heterogeneous recycling reactions' of what?

Response: The authors mean "Heterogeneous recycling reactions involving halogens". This is now clarified in the revised manuscript.

14) Page 4, lines 7-9: Here it would be good to have some information on the measurement techniques used.

Response: Air samples from the TORERO ship cruise were taken from a 10m bow mast and surface water samples were taken from the underway supply. Halocarbons in air and water phases were measured using two automated on- line GC-MS systems (Andrews et al., 2015) and calibrated using NOAA standard SX-3570. Ozone was measured by UV absorption (Coburn et al., 2014), OVOCs by the Trace Organic Gas Analyzer (TOGA) (Apel and UCAR/NCAR, 2016), and bromine oxide (BrO) and iodine oxide (IO) radicals were measured by the University of Colorado Airborne Multi-Axis Differential Optical Absorption Spectroscopy (CU AMAX-DOAS) instrument with typical detection limits of 0.5 pptv for BrO, and 0.05 pptv for IO (Volkamer et al., 2015; Dix et al., 2016). This information is now included in the revised manuscript.

S. Coburn, I. Ortega, R. Thalman, B. Blomquist, C. W. Fairall, and R. Volkamer. Measurements of diurnal variations and eddy covariance (EC) fluxes of glyoxal in the tropical marine boundary layer: description of the Fast LED-CE-DOAS instrument. Atmospheric Measurement Techniques, 7(10):3579–3595, 2014. doi:10.5194/amt-7-3579-2014. URL http://www.atmos-meas-tech.net/7/3579/2014/.

Apel, E., & UCAR/NCAR - Earth Observing Laboratory. (2011). Trace Organic Gas Analyzer (TOGA) for HIAPER. UCAR/NCAR - Earth Observing Laboratory. https://doi.org/10.5065/D6DF6P9Q Retrieved December 20, 2016

R. Volkamer, S. Baidar, T. L. Campos, S. Coburn, J. P. DiGangi, B. Dix, E. W. Eloranta, T. K. Koenig, B. Morley, I. Ortega, B. R. Pierce, M. Reeves, R. Sinreich, S. Wang, M. A. Zondlo, and P. A. Romashkin. Aircraft measurements of BrO, IO, glyoxal, NO 2 , H 2 O, O 2 –O 2 and aerosol extinction profiles in the tropics: comparison with aircraft-/ship-based in situ and lidar measurements. Atmospheric Measurement Techniques, 8(5):2121–2148, 2015. doi:10.5194/amt-8-2121-2015. URL http://www.atmos-meas-tech.net/8/2121/2015/.

S. J. Andrews, S. C. Hackenberg, and L. J. Carpenter. Technical Note: A fully automated purge and trap GC-MS system for quantification of volatile organic compound (VOC) fluxes between the ocean and atmosphere. Ocean Science, 11(2):313–321, 2015. doi:10.5194/os-11-313-2015. URL https://www.ocean-sci.net/11/313/2015/.

15) Page 4, first sentence of section 3.1: See comment 10, above.

Response: See answer from comment 10 above.

16) Page 4, line 29: How is Ka parameterised?

Response: Ka is parameterized following Johnson (2010) which is mainly a function of wind speed and sea surface temperature (SST) taken from the model at each time-step. Following Johnson (2010), Ka = ( 1/ka+KH/kw)-1, where ka is the rate of air-side transfer, kw is the rate of water-side transfer, and KH is the dimensionless gas over-liquid form of Henry's law constant. ka is based on Jeffrey et al. (2010) and kw is based on Nightingale et al 2000. We do not feel it necessary to explain this is the text of the manuscript beyond the first sentence where we cite Johnson (2010).

Johnson, M. T.: A numerical scheme to calculate temperature and salinity dependent air-water transfer velocities for any gas. Ocean Science, 6(4):913–932, 2010. doi:10.5194/os-6-913-2010. URL http://www.ocean-sci.net/6/913/2010/.

Jeffery, C. D., Robinson, I. S., and Woolf, D. K.: Tuning a physically-based model of the air–sea gas transfer velocity, Ocean Model., 31, 28–35, doi:10.1016/j.ocemod.2009.09.001, 2010.

Nightingale, P. D., G. Malin, C. S. Law, A. J. Watson, P. S. Liss, M. I. Liddicoat, J. Boutin, and R. C. Upstill-Goddard (2000), In situ evaluation of air-sea gas exchange parameterizations using novel conservative and volatile tracers, Global Biogeochem. Cycles, 14(1), 373–387, doi:10.1029/1999GB900091.

17) Page 5, lines 6-8: Give examples for OVOC's emitted from the ocean.

Response: OVOCs emitted from the ocean are acetaldehyde, ethanol and methanol.

Those three are included into the model. This is now clarified in the revised manuscript.

18) Page 5, line 8: Which VOC's were included? "the model as follows, the same... (2014); emissions for alkenes.. "

Response: Acetaldehyde, ethanol, methanol, propane, propene, ethane and ethene are included into the model. This is now clarified in the revised manuscript.

19) Page 5, lines 16-17: Give some details on sea-salt emission parameterisation.

Response: The sea-salt aerosol emission parameterization from Archer-Nicholls et al. (2014) is used in this study. This parameterization is mainly a function of wind speed from the model and uses the emissions scheme from Gong et al. (1997) for particles with dry diameters of 0.45nm or more and for smaller particles uses Fuentes et al. (2010). This information is now included in the revised manuscript.

Archer-Nicholls, S., Lowe, D., Utembe, S., Allan, J., Zaveri, R. A., Fast, J. D., Hodne-brog, Ø., Denier van der Gon, H., and McFiggans, G.: Gaseous chemistry and aerosol mechanism developments for version 3.5.1 of the online regional model, WRF-Chem, Geosci. Model Dev., 7, 2557-2579, https://doi.org/10.5194/gmd-7-2557-2014, 2014.

Gong, S. L., Bartie, L. A., and Blanchet, J.-P.: Modeling sea-salt aerosols in the atmo-sphere 1. Model development, J. Geophys. Res., 102, 3805–3818, 1997.

Fuentes, E., Coe, H., Green, D., de Leeuw, G., and McFiggans, G.: On the impacts of phytoplankton-derived organic matter on the properties of the primary marine aerosol – Part 1: Source fluxes, Atmos. Chem. Phys., 10, 9295–9317, doi:10.5194/acp-10-9295-2010, 2010.

20) Page 5, lines 21-25: This description is not quite clear. When 'Most of the reactions come from ... MISTRA', which ones?

Response: Inorganic, organic and inter-halogens reactions from Tables 2, 3 and 4 come from MISTRA. Moreover, the chemical loss of VSLH through oxidation by the

hydroxyl radical (OH) and by photolysis was included using data from Sander et al. (2011b). This information is now clarified in the revised manuscript.

How is the high uncertainty of I2Ox reactions dealt with?

Response: We have not tested the sensitivity of the results to the uncertainty of these reactions. We have included the state-of-the-art knowledge of these reactions (Saiz-Lopez et al., 2014).

A. Saiz-Lopez, R. P. Fernandez, C. Ordóñez, D. E. Kinnison, J. C. Gómez Martín, J.-F. Lamarque, and S. Tilmes. Iodine chemistry in the troposphere and its effect on ozone. Atmospheric Chemistry and Physics, 14(23):13119–13143, 2014. doi:10.5194/acp-14-13119-2014. URL http://www.atmos-chem-phys.net/14/13119/2014/.

21) Page 5, lines 31-33: Explain 17 bins and give a few details on FTUV. It would also be good to give e.g. noontime values of the photolysis frequencies (e.g. in Table 4).

Response: The Fast Tropospheric Ultraviolet-Visible model (FTUV) photolysis scheme is based on the Tropospheric Ultraviolet-Visible model (TUV) model developed by Madronich (1987) and uses a modified wavelength grid between 121–850 nm, where the number of spectral bins is reduced from 140 to 17 (Tie et al., 2004).

FTUV is a well-known and often used photolysis scheme. It is included as a standard option within WRF-Chem. We believe it is sufficient to refer to the paper Tie et al (2003) which describes the scheme in detail.

Below are the photolysis frequencies for a specific latitude and altitude at solar noon and under clear skies in response to the reviewer. However, these will vary throughout the model domain as a function of solar zenith angle, altitude and cloud cover so we feel that it would be misleading to include them in Table 4 of the paper.

Table 1. Photolysis reactions of halogens included in WRF-Chem.

Photolysis reactions Photolysis rates (s-1)

[Figure]

Cl2 -> 2 Cl 2.5 x 10-3

OClO (+O2) -> O3 + ClO 8.2 x 10-2

HOCl -> Cl + OH 3.0 x 10-4

ClNO2 -> Cl + NO2 5.7 x 10-4

ClNO3 -> Cl + NO3 4.3 x 10-5

ClNO3 -> ClO + NO2 4.7 x 10-4

Br2 -> 2 Br 3.1 x 10-2

BrO -> Br (+O3) 4.2 x 10-2

HOBr -> Br +OH 2.2 x 10-3

BrNO2 -> Br + NO2 2.6 x 10-2

BrNO3 -> Br + NO3 3.8 x 10-4

BrNO3 -> BrO + NO2 9.5 x 10-4

I2 -> 2 I 1.6 x 10-1

IO (+O2) -> I (+ O3) 1.5 x 10-3

I2O4 -> OIO + OIO 7.3 x 10-3

OIO -> I (+ O2) 2.0 x 10-1

I2O2 -> I + OIO 2.8 x 10-2

HOI -> I + OH 8.6 x 10-3

INO -> I + NO 3.0 x 10-2

INO2 -> I + NO2 3.3 x 10-3

INO3 -> I + NO3 1.0 x 10-2

I2O3 -> OIO + IO 3.0 x 10-2

IBr -> I + Br 7.3 x 10-2

ICl -> I + Cl 2.3 x 10-2

BrCl -> Br + Cl 1.0 x 10-2

CHBr3 (+O2) -> 3 Br 1.4 x 10-6

CH3Br -> Br + CH3O2 3.2 x 10-22

CH2Br2 -> 2Br 1.6 x 10-9

CH2BrCl -> Br + Cl 7.3 x 10-12

CHBrCl2 -> Br + 2 Cl 4.5 x 10-7

CHBr2Cl -> 2Br + Cl 2.7 x 10-7

CH2I2 + (O2) -> 2 I 1.0 x 10-2

CH3I -> I + CH3O2 8.7 x 10-6

CH2ClI -> I + Cl + 2 HO2+CO 1.8 x 10-4

CH2IBr -> Br + I 5.3 x 10-4

Tie,ÂăX.,ÂăS. Madronich,ÂăS. Walters,ÂăR. Zhang,ÂăP. Racsh, andÂăW. Collins (2003),ÂăEffect of clouds on photolysis and oxidants in the troposphere,ÂăJ. Geophys. Res.,Âă108, 4642, doi:Âă, D20.

22) Page 6, Section 3.3, Equation (2) is simplistic in that it neglects diffusion limitations, which can be important at uptake coefficients higher than about 0.01, see e.g. Schwartz and Freiberg 1967. The authors should check likely changes in model results when more realistic parameterisations of heterogeneous reactions are used. Schwartz S.E. and Freiberg J.E. (1967), Mass-transport limitation to the rate of reac- tion of gases in liquid droplets: Application to oxidation of SO2 in aqueous solutions, Atmospheric

Environment 15, (7),1129-1144

Response: See response above.

23) Page 7, Lines 13-28: Are reactions R11 and R12 not occurring at pH > 5.5?

Response: Reactions R11-R16 occurs only when pH < 5.5 and reactions R17-R19 only when pH> 5.5.

This is clarified in the document "We then apply a pH dependence to the heterogeneous reactions that occur on the surface of the sea-salt. When the pH < 5.5 debromination reactions occur with the release of $Br_2$ and $IBr$ resulting from the uptake of $BrNO_3$, $BrNO_2$, $HOBr$, $INO_3$, $INO_2$ and $HOI$ (R11-R16). When the pH > 5.5 no debromination reactions occur, although uptake of $INO_3$, $INO_2$ and $HOI$ on the sea-salt still occurs (R17-R19)"

What about mass conservation in R13?

Response: In this simplification we do not model the chemistry within the sea-salt explicitly so reactions R11-R19 represent just the changes to the gas phase species. As such mass may not be conserved and in particular for reaction R11-R16 they are intended to add an extra source of Br. The same approach is used in the global models TOMCAT (Yang et al., 2008) and CAM-Chem (Ordoñez et al., 2012).

Yang, X., J. A. Pyle, and R. A. Cox (2008), Sea salt aerosol production and bromine release: Role of snow on sea ice, Geophys. Res. Lett., 35, L16815, doi:10.1029/2008GL034536.

Ordóñez, C., Lamarque, J.-F., Tilmes, S., Kinnison, D. E., Atlas, E. L., Blake, D. R., Sousa Santos, G., Brasseur, G., and Saiz-Lopez, A.: Bromine and iodine chemistry in a global chemistry-climate model: description and evaluation of very short-lived oceanic sources, Atmos. Chem. Phys., 12, 1423-1447, https://doi.org/10.5194/acp-12-1423-2012, 2012.

Numbers or reactions are missing in Table 5

Response: No reactions are missing in Table 5.

24) Page 14, lines 15, 16: The model overestimates heterogeneous reactions on aerosol (see comment 22, above), thus the increase in the concentration of reactive species like BrO could be just a model effect.

Response: It is true that neglecting diffusion limitations will overestimate the impact of the heterogeneous reactions. As discussed above in response to the reviewer's earlier comment, there are many other uncertainties in the heterogeneous reactions. We have addressed this through sensitivity runs and revisions to the text described above.

25) Page 14, lines 19, 20: The underestimation of aldehydes could be due to neglect of Cl reactions in the model.

Response: The model chemical mechanism is of course a reduction of an explicit scheme, but it does include several reactions of Cl with VOCs that could lead to the formation of aldehydes including reactions of Cl with CH4, C2H6, C3H8 and C3H6 (See Table 2 and Section 3.2.1). The problem may be a lack of VOCs in the model rather than specifically reactions of VOCs with Cl.

In fact heterogeneous oxidation of halogenides (by O3) is discussed in the text, but does not appear to be implemented in the model, since no reactions to that effect can be found in Table 5.

Response: We do not understand what the reviewer is referring to here. The heterogeneous oxidation of halogenides (by O3) is not included. All the heterogeneous reactions discussed in the text are included in Table 5.

26) Page 14, line 22: What could be the reason for inorganic iodine emissions being lower than modelled in the tropics?

Response: The biggest uncertainty in the inorganic iodine emissions parameterization

is the calculation of the iodide concentration in the sea water. This is now added in the revised manuscript. Other uncertainties in the flux calculation are in the Henry's law of HOI and suppression of flux from DOC (Shaw et al., 2013).

Shaw, M. and Carpenter, L. (2013). Modification of Ozone Deposition and I2 Emissions at the Air-Aqueous Interface by Dissolved Organic Carbon of Marine Origin. Environmental science & technology. 47. 10.1021/es4011459.

27) Page 14, lines 23, 24: What is the reason for the minor importance of heterogeneous chemistry in the model?

Response: Unlike BrY, there is only a small change in the IY partitioning with the inclusion of the heterogeneous chemistry: The main change in IY with the inclusion of the heterogeneous chemistry occurs near the surface, due to the removal of the iodine oxides, and the production of more di-halogens in the MBL, especially when debromination is included. Heterogenous iodine reactions (reactions R11-R19 from the manuscript) compete with the photolysis. Iodine species are more readily photolyzed (see Table 1), so less is taken up into the aerosol and the impact of heterogeneous chemistry is less.

28) Page 14, line 25: For which conditions was the 31% contribution to O3 loss calculated?

Response: In the first version of the manuscript, OX loss calculations presented in the manuscript were from our base simulation, the WRF-DEBROM simulation, where its conditions were described in Sec. 4.1 and Table 7: "Our base simulation, WRF-DEBROM, considered all main processes involving halogen chemistry (sea-salt debromination, heterogeneous chemistry and reactions between halogens and VOCs) and computes the oceanic halocarbons fluxes online."

However, in order to quantify the uncertainty of OX loss calculations due to halogens, Fig. 12 have been updated and Fig. 14 has been included with calculations from other

simulations. These results are discussed above in the general comment.

29) Page 14, line 26: How can the Ox loss be attributed to Br, I, Cl reactions when cross reactions play a major role (e.g. ClO + BrO)?

Response: Any cross reaction between XO species (where X is Br, Cl and I) will be counted as a loss of OX and will be attributed to each family involved. i.e. BrO + IO means that the Br family is responsible for the loss of one OX in the form of IO and at the same time the I family is responsible for the loss of one OX in the form of BrO. This is clarified in the document "The tropospheric OX loss due to the BrOX , IOX and ClOX cycles" .

31) Page 14, lines 27-29: What is the meaning of 'negative bias', is it an overestimation of O3 loss?

Response: 'Negative bias' means that the simulation with halogens (DEBROM) is underestimating the observed O3 values in the MBL where the oceanic emissions of the halogenated species are higher (see Fig. 12 of the manuscript). The manuscript has been changed to "The simulation with halogens (WRF-DEBROM) is underestimating the observed O3 values in the MBL, where the oceanic emissions of the halogenated species are higher."

32) Page 14, lines 30-35: the reviewer can only agree with the statement in Line 30. What is the effect of these uncertainties for the conclusions drawn like 'Halogens constitute 31% of the overall ... O3 loss'? Should the conclusion not rather be 'Halogens reactions constitute a considerable fraction (10-60%) of the overall tropospheric O3 loss'? Response: See comment 28.

33) Caption of Figure 5: It would be more clear if "obs" was changed to "observation". In the time series of CH3I there is a clear anti-correlation between observation and model results during Feb. 5 – 10, can this be explained? Response: Amended. The word "obs" has been changed to "observation" in all Figures.

Results for low CH3I during this period are already discussed in the manuscript in Sec. 5.1 "A possible explanation for the underestimation in halocarbon atmospheric concentrations might be due to the input data (e.g. wind speed, SST, sea-water concentration) that we used to compute these fluxes. In the case of the online fluxes, between days 6-8 of February the model underestimates wind speed and this is directly accompanied by an underestimation for all three halocarbons atmospheric concentrations. "

F. Ziska, B. Quack, K. Abrahamsson, S. D. Archer, E. Atlas, T. Bell, J. H. Butler, L. J. Carpenter, C. E. Jones, N. R. P. Harris, H. Hepach, K. G. Heumann, C. Hughes, J. Kuss, K. Krüger, P. Liss, R. M. Moore, A. Orlikowska, S. Raimund, C. E. Reeves, W. Reifenhäuser, A. D. Robinson, C. Schall, T. Tanhua, S. Tegtmeier, S. Turner, L. Wang, D. Wallace, J. Williams, H. Yamamoto, S. Yvon-Lewis, and Y. Yokouchi. Global sea-to-air flux climatology for bromoform, dibromomethane and methyl iodide. Atmospheric Chemistry and Physics, 13(17):8915–8934, 2013. doi:10.5194/acp-13-8915-2013. URL http://www.atmos-chem-phys.net/13/8915/2013/.

34) Table 6: The N2O5 heterogeneous chemistry is not listed in Table 5. Response: N2O5 heterogeneous chemistry was already included in the model version we were using, which is why it was not included in Table 5 which lists the chemistry added as part of the current study. We have clarified this in the Table 5 heading. The N2O5 chemistry is described in Archer-Nicholls et al., 2014.

Archer-Nicholls, S., Lowe, D., Utembe, S., Allan, J., Zaveri, R. A., Fast, J. D., Hodnebrog, Ø., Denier van der Gon, H., and McFiggans, G.: Gaseous chemistry and aerosol mechanism developments for version 3.5.1 of the online regional model, WRF-Chem, Geosci. Model Dev., 7, 2557-2579, https://doi.org/10.5194/gmd-7-2557-2014, 2014.
* * *

[Figure]

Fig. 1.

**IO (pptv) - Subtropics**

**IO (pptv) - Tropics**

Fig. 2.

[Figure]

Fig. 3.

**A)** BrO(pptv) - Tropics

Observation
WRF-NOBRALKE
WRF-NOBRALD
WRF-NOBRVOCS
WRF-DEBROM

**B)** Inorganic bromine of WRF-NOBRVOCS

BrCl+Br$_2$+BrI
HOBr
BrO
HBr
BrNO$_2$
BrNO$_3$
Br

**C)** CH$_3$CHO(pptv) - Tropics

Observation
WRF-DEBROM

**D)** HCHO(pptv) - Tropics

Observation
WRF-DEBROM

**Fig. 4.**

[Figure]

**Fig. 5.**

[Figure]

**Fig. 6.**

---

## Author Response (AR1)

**RC1: 'Review of Badia et al. 2016', Anonymous Referee #1, 19 January 2018**

Review of ""Importance of reactive halogens in the tropical marine atmosphere: A regional modelling study using WRF-Chem" by Alba Badia et al.

**The authors have completed an enormous amount of work, but unfortunately the paper**
**does not currently meet the requirements for publication in ACP.**

**This paper can be published after addressing the following major comments:**
**- The authors must provide a justification as to why they have not included chlorine cycling in the model. Even if there are no measurements of chlorine, the authors cannot focus on bromine and iodine cycling without some description of chlorine chemistry. The authors cannot simply ignore chlorine especially in the remote marine boundary layer. See for example Schmidt et al. (2016).**

*Response: Chlorine cycling is included in the model. This is mentioned in the Abstract "To do this the regional chemistry transport model WRF-Chem has been extended, for the first time, to include halogen chemistry (bromine, chlorine and iodine chemistry), Section 3.1 "This mechanism has been extended to include bromine, chlorine and iodine chemistry" and Section 3.2 "Chlorine chemistry is also included into the model, however, since our results are mainly focused on reactive bromine and iodine for which we have observed data, we have not included the chlorine chemistry in Fig 4." Moreover, Tables 1-5 describe all the reactions (including chlorine) that have been added to the model.*

*To make it clearer that chlorine is implemented in the model, we have added the following in the caption of Fig.4 "Chlorine chemistry has been added into the model, but since our results mainly focus on reactive bromine and iodine, chlorine chemistry is omitted from this figure (See Tables 1-5 for full list of additional reactions)."*

*In the 2 sections where we consider the impact of the halogen chemistry on VOCs (5.3) and on $O_3$ and $O_X$ (5.4), we include a consideration of the Cl chemistry which was referred to several times in these two sections. We have added more on the Cl chemistry in Section 5.4 , a figure of the $Cl_Y$ partitioning (Fig. 12) and the integrated odd oxygen loss rates for $ClO_x$ cycles in Fig. 15. This new information has been included in the revised manuscript, as follows:*
*"When the VOCs react with Cl (WRF-DEBROM), almost all the inorganic Cl is in the form of HCl (see Fig. 12). When these reactions are not considered (WRF-NOHALVOCS), $Cl_y$ increases and there is a shift in the partitioning to more reactive chlorine increases, in particular HOCl, but also ClO and the di-halogens."*
*"Moreover, the big change though is for the $ClO_x$ which increases from < 1% to 26%. Cl is very important in the oxidation of the alkanes. When this chemistry is not included the concentrations of $Cl_y$ increases and there is an impact on the partitioning increasing reactive species (see Fig. 12), hence, the $ClO_x$ cycles play an important role in $O_x$ loss. It should be noted that very little is known about the abundance and distribution of $Cl_y$ so this is a large uncertainty."*

*We agree that not including the Cl chemistry would have been a significant weakness, but this simply was not the case. We hope this is now completely clear to readers and that the additions address the reviewer's concerns.*

**- A more complete description of heterogeneous chemistry already included in the model is needed. If the authors used the model as described, ClNO2 is formed on aerosols via reactions of N2O5. However, once formed ClNO2 is treated as an inert species. This should be updated prior to publication.**

*Response: This heterogeneous chemistry is included in this work. Moreover, $ClNO_2$ is not treated as an inert species but is broken down via photolysis and reaction with OH (see Table 2 and 4). Further information on this chemistry has now been added in the revised manuscript in which we cite another WRF-Chem paper (Archer-Nicholls et al., 2014) that provides more details of this chemistry as implemented in the model. This new information has been included in the revised manuscript, as follows:*
*"After uptake $N_2O_5$ is taken up onto the particle, it reacts reversibly with liquid water to form protonated nitric acid intermediate ($H2ONO^{+2}$). This then reacts with either liquid water, to form aqueous nitric acid ($HNO_3$), or with chloride ions to form $ClNO_2$. See Archer-Nicholls et al. (2014) for further description of this chemistry. In Archer-Nicholls et al. (2014) $ClNO_2$ was considered as an inert specie, however in our study $ClNO_2$ is not treated as an inert specie but is broken down via photolysis and reaction with OH (see Tables 2 and 4)."*

*Some text on this heterogeneous chemistry in Archer-Nicholls et al. (2014) is reproduced below for the purpose of this response, but we do not feel it necessary to repeat these details in the revised manuscript as it is published elsewhere:*
*"The reaction mechanism that is used for the hydrolysis of $N_2O_5$ is that of Thornton et al. (2003). They suggest that, after uptake onto the aerosol particle, aqueous phase $N_2O_5$ reacts reversibly with liquid water to form an (as yet unobserved) protonated nitric acid intermediate ($H_2ONO^{+2}$). This then reacts with either liquid water, to form aqueous nitric acid ($HNO_3$), or with halide ions to form nitryl halide ($XNO_2$; where X=Cl, Br, or I):*

*$N_2O_5$ (gas) $N_2O_5$ (aq)*          *(R3)*
*$N_2O_5$ (aq) + $_{H2O}$(l) $H_2ONO^{+2}$ (aq) + $NO^{-3}$ (aq)*          *(R4)*
*$H_2ONO^{+2}$ (aq) + $H_2O$(l) $-\rightarrow$ $H_3O$ + (aq) + $HNO_3$ (aq)*      *(R5)*
*$H_2ONO^{+2}$ (aq) + $X^-$ (aq) $-\rightarrow$ $XNO_2$ + $H_2O$(l).*          *(R6)*

*In applying the parameterisation of Bertram and Thornton (2009) we assume that the limiting step is the uptake of $N_2O_5$ to the condensed-phase, and that it reacts in a near instantaneous manner with $H_2O$ and $Cl^-$ to give $NO^{-3}$ and $ClNO_2$ through Reactions (R4)–(R6). $ClNO_2$ is not added to the aerosol, but is instead assumed to out-gas in a near instantaneous manner, and has instead been added as an extra species to the gas-phase (currently as an inert tracer - no gas-phase reactions involving $ClNO_2$ have been added to the gas-phase chemistry scheme, although this could be added in the future, e.g. following Sarwar et al., 2012). In addition, for simplicity, we assume that the HNO3 molecules formed in Reaction (R5) undergoes ion dissociation to produce aqueous $NO^{-3}$ ."*

*Archer-Nicholls, S., Lowe, D., Utembe, S., Allan, J., Zaveri, R. A., Fast, J. D., Hodnebrog, Ø., Denier van der Gon, H., and McFiggans, G.: Gaseous chemistry and aerosol mechanism*

*developments for version 3.5.1 of the online regional model, WRF-Chem, Geosci. Model Dev., 7, 2557-2579, https://doi.org/10.5194/gmd-7-2557-2014, 2014.*

**- As already pointed out by the other reviewer, Equation 2 is not an acceptable treatment for reactions on aerosols. This equation ignores the fact that reactions are limited by the rate at which gases can diffuse towards the aerosol surface (diffusion limitation).**
**There are clear descriptions of how to treat this correctly in the literature, for example in Schwartz (1986).**
*Response: See response to the other reviewer.*

**- While this paper was not submitted to Geoscientific Model Development (GMD), this is the first paper that describes a new model development. The paper should be held to the GMD standard for publication. For example, the subroutines that have been updated to perform the study should be included in an electronic supplement and/or the code used in the study should be provided as an electronic supplement.**
*Response: Copies of the code and data used in this study are readily available upon request from the corresponding authors. To our knowledge it is not a requirement to publish new model code for papers published in ACP, however, we are happy to be guided on this one by the editorial office.*

**The authors have attempted to study the influence of bromine and chlorine cycling in the tropical marine atmosphere. This can be an important study and hopefully the authors will make the effort to address these major comments so that the paper can be published in ACP.**
*Response: The authors wish to thank anonymous reviewer #1 for his/her valuable comments and suggestions. We have added further comments to clarify that chlorine chemistry has been included in the model and hope that this will avoid any future misunderstanding. We're very happy to make new code available to further scientific research and look forward to guidance from the editorial office regarding any requirement regarding supplementary information. Whilst we acknowledge the limitation in our representation of the heterogeneous chemistry, we see no reason for this being a barrier to publication as this approach is one that has been adopted in other recent studies that have been acceptable for publication. We have added new material from additional runs to illustrate the sensitivity of the results to the approach we have taken.*

**References:**
**Schwartz S.E. (1986) Mass-Transport Considerations Pertinent to Aqueous Phase Reactions of Gases in Liquid-Water Clouds. In: Jaeschke W. (eds) Chemistry of Multiphase Atmospheric Systems. NATO ASI Series (Series G: Ecological Sciences), vol 6. Springer, Berlin, Heidelberg.**
**Schmidt, J. A., et al. (2016), Modeling the observed tropospheric BrO background: Importance of multiphase chemistry and implications for ozone, OH, and mercury, J. Geophys. Res. Atmos., 121, 11,819–11,835, doi:10.1002/2015JD024229.**

**RC2: 'Review of Badia et al. 2016', Anonymous Referee #2, 12 December 2017**

Review of ""Importance of reactive halogens in the tropical marine atmosphere: A regional modelling study using WRF-Chem" by Alba Badia et al.

**The manuscript attempts to compare observations of reactive halogen species (containing Cl, Br, I) in the Pacific with model results. Comparison of model results with real observations are always important and n particular provide a test of the degree of our understanding of the modelled part of the reality. As such the authors have to be commended for their undertaking. The outcome of the inter comparison is the (not entirely new) finding that our knowledge of the relevant halogen chemistry and the source mechanisms of halogen species is very limited. Probably a meaningful modelling of abundance and distribution of reactive halogen species in the troposphere is not even possible at present. Nevertheless the modelling effort appears to be a useful exercise, however it is important to more clearly state what can be accomplished with models at present and what not.**

**In particular it would be very helpful if the following could be stated:**
**1) What are the real uncertainties in the model results taking into account our knowledge gaps. And what is the uncertainty of stated numbers (e.g. the tropospheric O3 loss due to halogen reactions and its vertical distribution as shown in Fig. 13). The real uncertainties are clearly much larger than the differences between the various scenarios (given in Table 7).**
**2) Where are the most important gaps in our knowledge, both in the area of input data (e.g. precursor fluxes) and regarding mechanistic data.**
**3) What could be done in the future to improve the situation**
*Response: We address each of the individual points below.*

**In detail there are many open questions and points to improve as shown in the (incomplete) numbered list below. For instance what is the Definitions of 'halogens' in the text (e.g. in the 1st line of the abstract certainly not just halogen molecules or halogen atoms are meant) and of 'reactive halogen' and 'Bry'.**
*Response: "Reactive halogens species, containing Cl, Br and I" from Saiz-Lopez and von Glasow (2012). $Br_Y$ is defined as inorganic bromine. These definitions are now included in the updated manuscript as follows:*
*"This study investigates the impact of reactive halogen species (RHS, containing chlorine (Cl), bromine (Br) or iodine (I))".*

**Overall I feel that this manuscript has the potential to become an important contribution to our understanding of tropospheric halogen chemistry once the authors had a chance to make the above additions and to answer the many unclear points listed below. Unfortunately due to the simplistic implementation of aerosol reactions (Equation (2), see comment 22) below) the model runs must be repeated with a state of the art parameterisation of heterogeneous aerosol reactions. This is also likely to change many results of the manuscript. In view of the many required changes and the many unclear points a very major revision of the manuscript is required.**

*Response: The authors wish to thank anonymous reviewer #2 for his/her valuable comments and suggestions.*

*We agree with the reviewer that current understanding of the sources of halogen species and their chemistry is limited. Much of this is due to a lack of field observations with which to constrain model input and to evaluate models with. The new observations from TORERO provide a great opportunity to advance our understanding of the importance of reactive halogens in the tropical marine atmosphere and to highlight gaps in knowledge and this paper is part of that process. This contribution describes an important step in the understanding the tropospheric halogen chemistry because it is the first time that halogen chemistry (including bromine, chlorine and iodine) has been implemented into a regional online chemistry transport model. Of course there are limitations and the suggestions made by the reviewer in points 1 - 3 above are very valuable and we have made improvements to the manuscript to address them - see below.*

*Heterogeneous chemistry (uptake coefficients and surface aerosol area), photolysis rates (especially iodine photochemistry (Sommariva et al., 2012)), the chemical mechanism for the repartitioning of $Br_y$ resulting from the reactions of Br with VOCs, interactions between halogens and the sulfur cycle (Chen et al., 2017, Chen et al., 2018), physical constants (such as Henry's Law value for HOI), deposition of halogenated species and inputs to calculate the oceanic fluxes (e.g. sea-water concentration, including iodide concentrations) are the most important gaps in our knowledge. We have discussed these more in the revised manuscript, in particular emphasizing how the results of our study contribute to this.*

*We are aware that our treatment of the aerosol chemistry is a simplification, but it is an approach in-line with many other studies published since the Schwartz and Freiberg paper in 1967. Due to computational resource limitations, large-scale models must be parameterized representations of reality. Furthermore, we are limited by the input data with which to constrain these parametrizations.*

*One important limitation is information with which to constrain a more detailed treatment of the heterogeneous chemistry. We have taken an approach similar to that used with other models such as TOMCAT (Yang et al., 2008), CAM-Chem (Ordoñez et al., 2012) and GU–WRF/Chem (Karamchandani et al., 2012). In this approach the heterogeneous chemistry is assumed to take place between a gas-phase species and an adsorbed species with an uptake coefficient (γ) used to calculate first-order rate constants for heterogeneous loss of the gas-phase species. There are large uncertainties in these uptake coefficients. Moreover, there are uncertainties in the ability of the model to simulate the aerosol size distribution (and thus aerosol surface area) and the mixing state and surface composition of the aerosols. In addition, there are a limited number of measurements with which to evaluate the model and these measurements have their own uncertainties. We agree that ignoring the diffusion limitation will artificially increase the heterogeneous reactions in the model, however we feel this is one of several uncertainties that will affect the rate of heterogeneous chemistry. Instead of focusing on one factor, we prefer to address the wider uncertainties in the heterogeneous chemistry through sensitivity studies in which γ is varied.*

*To illustrate the sensitivity of our results to this we have made a new run assuming that γ is reduced by a factor of 2 (WRF-GAMMADV2). Moreover, simulations WRF-NOALD, WRF-*

*NOALK and WRF-NOVOCS have been renamed as WRF-NOBRALD, WRF-NOBRALK and WRF-NOBRVOCS. A new simulation where Br and Cl reactions with VOCS are not considered have been made and named WRF-NOHALVOCS. Some of the new results have been added in the revised manuscript (Sec. 5) and Supplement (Fig S4) and we also present them below:*

[Figure]

*Figure 1. Mean vertical profile of BrO (pptv) over the subtropics (left) and tropics (right). An average over 16 flights of the TORERO campaign (red line) are compared to the 4 different WRF-Chem simulations: WRF-NOHET (blue line), WRF-NODEBROM (green line), WRF-DEBROM (black line) and WRF-GAMMADV2 (pink line). Orange and grey horizontal bars indicate the 25th-75th quartile interval for the observations of the TORERO campaign and WRF-DEBROM simulation, respectively. Values are considered in 0.5 km bins and the number of aircraft measurement points for each altitude are given on the right side of each plot.*

[Figure]

*Figure 2. Mean vertical profile of IO (pptv) over the subtropics (left) and tropics (right). An average over 16 flights of the TORERO campaign (red line) are compared to the 3 different WRF-Chem simulations: WRF-NOHET (blue line), WRF-NODEBROM (green line), WRF-DEBROM (black line) and WRF-GAMMADV2 (pink line). Orange and grey horizontal bars indicate the 25th-75th quartile interval for the observations of the TORERO campaign and WRF-DEBROM simulation, respectively. Values are considered in 0.5 km bins and the number of aircraft measurement points for each altitude are given on the right side of each plot.*

[Figure]

*Figure 3: Regional average vertical partitioning of inorganic chlorine ($Cl_y$) for the two different simulations WRF-NOHALVOCS (left panel) and WRF-DEBROM (middle panel) during January and February 2012. Regional average vertical partitioning of reactive chlorine species ($Cl^*$) for WRF-DEBROM is also showed (right panel). $Cl^*$ is defined as $Cl_y$ gases other than HCl. Top panels are over the subtropical area and bottom panels over the tropical. Units are in pptv.*

[Figure]

*Figure 4: A) Mean vertical profile of BrO (pptv) over the tropics in the left panel. A sub-set of 9 flights from the TORERO campaign (red line) are compared to the 4 different WRF-Chem simulations: WRF-NOBRALKE (blue line), WRF-NOBRALD (green line), WRF-NOBRVOCS (blue line) and WRF-DEBROM (black line). B) Regional average vertical partitioning of inorganic bromine (Bry ) for the the WRF-NOBRVOCS run over the subtropical area (left panel) and over the tropical area (right panel) during January and February 2012. C) and D) the WRF-DEBROM (black line) simulation is compared with acetaldehyde and formaldehyde TORERO observations for the same flights (red line). Orange and grey horizontal bars indicate the 25th-75th quartile interval for the observations of the TORERO campaign and WRF-DEBROM simulation, respectively. Values are considered in 0.5 km bin and the number of points for each altitude is given on the right side of each plot. Units are in pptv.*

[Figure]

*Figure 5. On the left, mean vertical profile of O₃ (ppbv) over the domain area using 13 flights from the TORERO campaign (red line) are compared to the 6 different WRF-Chem simulations: WRF-NOHAL (purple line), WRF-NOHET (blue line), WRF-NODEBROM (light green line), WRF-DEBROM (black line), WRF-GAMMADV2 (pink line) and WRF-NOHALVOCS(dark green line). Orange and grey horizontal bars indicate the 25th-75th quartile interval for the observations of the TORERO campaign and WRF-DEBROM simulation, respectively. Values are considered in 0.5 km bins and the number of aircraft measurement points for each altitude are given on the right side of each plot.*

[Figure]

Figure 6. Integrated odd oxygen loss rates for each O3 depleting halogen family within the troposphere at different altitude levels: MBL (surface - 900 hPa), FT (900 hPa- 350 hPa), UT (350 hPa-tropopause) and troposphere (surface-tropopause) for the WRF-DEBROM, WRF-GAMMADV2, WRF-NODEBROM, WRF-NOHET and WRF-NOHALVOCS simulation.

*New text has been added to the revised manuscript that discuss the results and the uncertainty of the stated numbers, such as the tropospheric $O_3$ loss due to halogen reactions and its vertical distribution. The main new points that have been added are:*

*"Areas, such as the tropics, where debromination dominates, the impact of halving gamma (WRF-GAMMADV2 run) is approximately half of the impact of including heterogeneous chemistry (i.e. the difference between the WRF-DEBROM run and the WRF-NOHET run) at least for the lower troposphere. Very little impact is seen in the UT, a slight decrease in BrO, when gamma is halved (WRF-GAMMADV2)."*

*"Changing GAMMA (difference between the WRF-DEBROM and WRF-GAMMADV2 runs) has very little impact on IO."*

*"The heterogeneous halogen chemistry has an impact on $O_3$ concentrations where a difference of up to 3 ppbv of $O_3$ is seen between the simulation with and without heterogeneous chemistry (WRF-DEBROM run WRF-NOHET run, respectively) mainly in the MBL. The modelled $O_3$ is highly sensitive to the inclusion of the reactions of the halogens with the VOCs (WRF-NOHALVOCS) where $O_3$ concentrations are much lower (between 12-7 ppbv) than in the WRF-DEBROM run."*

*"When comparing different simulations with the WRF-DEBROM run, the biggest difference is seen with WRF-NOHALVOCS simulation, where around 60% of $O_x$ is removed by halogens. BrO is much higher when the VOC reactions are not included (see Fig. 11), which explains why the amount of $O_x$ lost by $BrO_x$ reactions is much larger (20.5%). Moreover, the big change though is for the $ClO_x$ which increases from < 1% to 26%. Cl is very important in the oxidation of the alkanes. When this chemistry is not included the concentrations of Cly increases and there is an impact on the partitioning increasing reactive species (see Fig. 12), hence, the $ClO_x$ cycles play an important role in $O_x$ loss. It should be noted that very little is known about the abundance and distribution of Cly so this is a large uncertainty. Therefore, a large uncertainty in the impact of halogen cycling on the $O_3$ budget are the reactions of halogens with VOCs. In the model runs performed, excluding these reactions doubled the percentage contribution of halogens to $O_x$ loss (i.e. increase it from 31% to 60%) in the troposphere. Heterogeneous chemistry (including debromination) has the effect of increasing the $O_x$ loss by halogen cycling from 25 to 31% for the whole troposphere (i.e. comparision between WRF-NOHET and WRF-DEBROM runs). For the UT the equivalent values are 37% to 40%, for the FT 13% to 18% and for the MBL 23% to 34%. Hence, heterogeneous chemistry increases the percentage of the $O_x$ loss that is attributable to the halogens by about 6% for the troposphere ranging from 3% to 11% depending on the region of the troposphere. Note that the gas phase halogen chemistry makes a bigger contribution of around 25% (WRF-NOHET run) to the $O_x$ loss for the troposphere ranging from 13% to 37% depending on the region of the troposphere. Therefore, the overall impact of the halogen chemistry on $O_x$ loss appears not to be very sensitive to the treatment of the heterogeneous chemistry."*

*Future steps should focus on providing more data from laboratory studies along with field observations to better constrain the modelled representation of halogen species. Only a few $Br_y$ and $I_y$ species are provided from field campaigns making it difficult to evaluate the model against observations and a very limited amount of data exist to compute the sea-water*

*concentration for the halocarbons, giving uncertainty in this input data when the halocarbon fluxes are computed. Furthermore, it is clear that we need to understand better the emissions and distributions of VOCs in the remote atmosphere, and the impact of their chemistry on halogen repartitioning.*

*Yang, X., J. A. Pyle, and R. A. Cox (2008), Sea salt aerosol production and bromine release: Role of snow on sea ice, Geophys. Res. Lett., 35, L16815, doi:10.1029/2008GL034536.*

*Ordóñez, C., Lamarque, J.-F., Tilmes, S., Kinnison, D. E., Atlas, E. L., Blake, D. R., Sousa Santos, G., Brasseur, G., and Saiz-Lopez, A.: Bromine and iodine chemistry in a global chemistry-climate model: description and evaluation of very short-lived oceanic sources, Atmos. Chem. Phys., 12, 1423-1447, https://doi.org/10.5194/acp-12-1423-2012, 2012.*

*Karamchandani et al., 2012: Development of an extended chemical mechanism for global-through-urban applications, Atmospheric Pollution Research 3 (2012), doi:10.5094/APR.2011.047*

*Chen, Q., J. A. Schmidt, V. Shah, L. Jaeglé, T. Sherwen, and B. Alexander (2017), Sulfate production by reactive bromine: Implications for the global sulfur and reactive bromine budgets, Geophys. Res. Lett., 44, 7069–7078, https://doi.org/10.1002/2017GL073812*

*Chen, Q., Sherwen, T., Evans, M., and Alexander, B.: DMS oxidation and sulfur aerosol formation in the marine troposphere: a focus on reactive halogen and multiphase chemistry, Atmos. Chem. Phys., 18, 13617-13637, https://doi.org/10.5194/acp-18-13617-2018, 2018.*

*R. Sommariva, W.J. Bloss, R. von Glasow, Uncertainties in gas-phase atmospheric iodine chemistry, Atmospheric Environment, Volume 57, 2012, 219-232, 1352-2310,https://doi.org/10.1016/j.atmosenv.2012.04.032.*

**Detailed comments:**
**1) Page 2, line 4: Define the term 'reactive halogens'**
*Response: "Reactive halogens species, containing Cl, Br and I" from Saiz-Lopez and von Glasow (2012). This definition is now included in the revised manuscript (1st line of the abstract).*

*A. Saiz-Lopez and R. von Glasow. Reactive halogen chemistry in the troposphere. Chem. Soc. Rev., 41:6448–6472, 2012. doi:10.1039/C2CS35208G. URL http://dx.doi.org/10.1039/C2CS35208G.*

**2) Page 2, line 5: Only Cl-atoms 'reduce the lifetime of CH4'**
*Response: Amended. The manuscript has been changed.*

**3) Page 2, lines 6, 7: Define the term 'halogen species', also the statement '. . . play an important role in the oxidation capacity . . .' appears to be a repetition of the statements made above.**

*Response:* Yes it is a repetition, so this sentence has been removed from the revised manuscript.

**4) Page 2, line 10: what should be the effect on 'oxidation capacity' beyond the already mentioned CH4 and particles?**
*Response:* We do not understand the reviewer's question. In the paragraph that the reviewer points to we already briefly explain how halogen chemistry affects the budgets of key oxidants in the atmosphere in order to provide relevance and justification for this study. The original paragraph with minor modifications in response to other comments is copied below and we cannot see what additional information the reviewer is asking for.

*"Reactive halogens cause ozone ($O_3$) destruction, change the $HO_X$ ($HO_2$ + OH) and $NO_X$ ($NO_2$ + NO) partitioning, affect the oxidation of VOCs and mercury, and take part in new particle formation (Chameides and Davis, 1980; von Glasow et al., 2004; Saiz-Lopez and von Glasow, 2012). Moreover, reactive chlorine reduces the lifetime of methane ($CH_4$). Halogen species are known to play an important role in the oxidising capacity of the troposphere. The atmospheric oxidation capacity is to a large extent determined by the budgets of the hydroxyl radical (OH) and $O_3$; globally most tropospheric OH is found in the tropics (Bloss et al., 2005). Therefore a quantitative understanding of the composition and chemistry of the tropical marine atmosphere is essential to examine the atmospheric oxidative capacity and climate forcing."*

**5) Page 2, lines 12-15: What about fluorine? Are there not other cycles, e.g. XO self reactions and cross reactions like BrO + IO, . . .?**
*Response:* We don't consider fluorine chemistry in the model.

*"Fluorine atoms are so reactive that they form HF very rapidly and play no role in tropospheric chemistry." from Saiz-Lopez and von Glasow (2012).*

*A. Saiz-Lopez and R. von Glasow. Reactive halogen chemistry in the troposphere. Chem. Soc. Rev., 41:6448–6472, 2012. doi:10.1039/C2CS35208G. URL http://dx.doi.org/10.1039/C2CS35208G.*

**6) Page 2, lines 16-29: this paragraph should be edited for better structure. At present the authors talk about 'numerical models' in general then progress to global and regional models, then progress to specific models (e.g. WRF chem)**
*Response:* This paragraph has been rewritten for better structure as follows:

*"In the past, tropospheric halogen chemistry has been studied using a number of box models and 1D models (Sander and Crutzen, 1996; von Glasow et al., 2002a; Saiz-Lopez et al., 2006; Simpson et al., 2015; Lowe et al., 2009; Sommariva and von Glasow, 2012). Currently, there are several global models that have been used to study tropospheric halogens (Hossaini et al., 2010; Ordóñez et al., 2012; Saiz-Lopez et al., 2012a; Fernandez et al., 2014; Saiz-Lopez et al., 2015; Sherwen et al., 2016b; Schmidt et al., 2016). Numerical models predict that reactive halogen compounds account for 30% of $O_3$ destruction in the MBL (von Glasow et al., 2002b, 2004; Saiz-Lopez et al., 2015; Sherwen et al., 2016b) and 5-20% globally (Yang et al., 2005; Saiz-Lopez et al., 2015, 2012a; Sherwen et al., 2016b). Up to 34% of $O_3$ loss is calculated to be due to I and Br combined the tropical East Pacific*

*(Wang et al., 2015). However, there are only a few regional models that have studied tropospheric halogens. Chlorine chemistry was implemented into the WRF-Chem model (Lowe et al., 2015; Li et al., 2016) and into the CMAQ model (Sarwar et al., 2014) to study the formation of nitryl chloride ($ClNO_2$) from the uptake of dinitrogen pentoxide ($N_2O_5$) on aerosols containing chloride. Moreover, bromine and iodine chemistry was implemented in CMAQ in Gantt et al. (2017) and Sarwar et al. (2015), where the impact of iodide-mediated O3 deposition on surface ozone concentrations was studied, and in the recent work of Muñiz-Unamunzaga et al. (2018), that concluded that oceanic halogens and dimethyl sulfide (DMS) emissions need to be included into the regional models to accurately reproduce the air quality in coastal cities."*

**7) Page 2, line 30: Give definition of VSLH.**
*Response: Very Short Lived Halocarbons (VSLH) are defined as trace gases whose chemical lifetimes are generally under six months. This information is now included in the revised manuscript.*

**8) Page 3, line 1: What about the direct transport of XO?**
*Response: The lifetime of XO radicals is sufficiently short, ranging from minutes to one hour. Thus, direct transport of XO is not considered.*

**9) Page 3, line 11: Which OVOC species do the authors have in mind?**
*Response: Basically aldehydes. Aldehydes species that have been included in the model are formaldehydes and acetaldehydes. This is described in section 3.2.1.*

**10) Page 3, lines 15-19: Which heterogeneous reactions do the authors have in mind? Why is Cl not mentioned?**
*Response: Heterogeneous reactions involving reactive halogen species are included in the model. This is now clarified in the revised manuscript.*
*Cl is not mention here because we try to explain observations of BrO and IO and there are no reported direct observations of Cl.*

**How does debromination work? Is it different from heterogeneous Br-release through HOBr and BrNO3? Is there not also de-iodination and de-chlorination? Actually it would be good to have a list of sources of halogen atoms.**
*Response: Debromination is the process by which sea-salt aerosols are depleted in bromide (Br$^-$) through the release of Br into the gas-phase. It occurs through the uptake of a gaseous species into the sea-salt and the subsequent reaction with Br$^-$ (Yang et al, 2005). We represent this as release of Br through heterogeneous reactions that involve uptake of gases species including HOBr and $BrNO_3$. We described this in more detail on page 7. We changed the text on pages 3 and 7 to make it clearer what we mean by debromination:*

*Page 3: "Another source of reactive inorganic bromine in the troposphere is the release of bromide (Br$^-$) from sea-salt aerosols into the gas-phase. This is known as debromination and occurs through the uptake of a gaseous species in the sea-salt and the subsequent reaction with Br$^-$.*
*Page 7. "It is known that the chemistry involving the release of bromine from the sea-salt aerosol (debromination) is strongly pH dependent, being more efficient for acidified aerosol especially with a pH < 5.5 (Keene et al., 1998). Therefore, the pH value of the aerosol*

*particles is calculated in the model for each size bin (see Zaveri et al. (2008) for further description of the pH calculation). We then apply a pH dependence to the heterogeneous reactions that occur on the surface of the sea-salt. When the pH < 5.5 debromination reactions occur with the release of $Br_2$ and IBr resulting from the uptake of $BrNO_3$, $BrNO_2$, HOBr, $INO_3$, $INO_2$ and HOI (R11-R16). When the pH > 5.5 no debromination reactions occur, although uptake of $INO_3$, $INO_2$ and HOI on the sea-salt still occurs (R17-R19)".*

*Whilst the processes of deiodination and dechlorination do occur, they are not thought to be important sources of atmospheric $I^-$ and $Cl^-$ so we do not include them in the model.*

*List of sources of halogen atoms considered in this study: oceanic source of organic halogens ($CHBr_3$, $CH_2Br_2$, $CH_3I$, $CH_2BrCl$, $CHBrCl_2$, $CHBr_2Cl$, $CH_2I_2$, $CH_2IBr$ and $CH_2ICl$), debromination ($Br_2$, IBr) and inorganic I from the ocean (HOI and $I_2$). This list has been added in the revised manuscript in Sec. 3.*

**11) Page 3, lines 19-22: What does 'This source' refer to?**
*Response: This 'source' refers to the 'source' mention in the previous sentence i.e. 'debromination', the last word in the previous sentence. However, with the rewording made to address the previous point "This source has been included" has been replaced with "Debromination has been included as a source of gas-phase bromine".*

**What exactly is poorly understood?**
*Response: With the rewording of the previous sentence "this process" now clearly refers to debromination.*

**What do the authors mean with 'inconsistent levels' too high or too low?**
*Response: The authors mean inconsistent high levels of bromine species. This is now clarified in the revised manuscript.*

**12) Page 3, line 23: Do the authors really mean to state that all atmospheric models are largely untested or does this statement rather pertain to halogen chemistry in models?**
*Response: This statement pertains to halogen chemistry in models. We have rewritten the start of this sentence in the revised manuscript: "Halogen chemistry in atmospheric models...".*

**13) Page 3, line 33: 'heterogeneous recycling reactions' of what?**
*Response: The authors mean "Heterogeneous recycling reactions involving halogens". This is now clarified in the revised manuscript.*

**14) Page 4, lines 7-9: Here it would be good to have some information on the measurement techniques used.**
*Response: Air samples from the TORERO ship cruise were taken from a 10m bow mast and surface water samples were taken from the underway supply. Halocarbons in air and water phases were measured using two automated on- line GC-MS systems (Andrews et al., 2015) and calibrated using NOAA standard SX-3570. Ozone was measured by UV absorption (Coburn et al., 2014), OVOCs by the Trace Organic Gas Analyzer (TOGA) (Apel and UCAR/NCAR, 2016), and bromine oxide (BrO) and iodine oxide (IO) radicals were*

*measured by the University of Colorado Airborne Multi-Axis Differential Optical Absorption Spectroscopy (CU AMAX-DOAS) instrument with typical detection limits of 0.5 pptv for BrO, and 0.05 pptv for IO (Volkamer et al., 2015; Dix et al., 2016).*
*This information is now included in the revised manuscript in Sec. 2.*

*S. Coburn, I. Ortega, R. Thalman, B. Blomquist, C. W. Fairall, and R. Volkamer. Measurements of diurnal variations and eddy covariance (EC) fluxes of glyoxal in the tropical marine boundary layer: description of the Fast LED-CE-DOAS instrument. Atmospheric Measurement Techniques, 7(10):3579–3595, 2014. doi:10.5194/amt-7-3579-2014. URL http://www.atmos-meas-tech.net/7/3579/2014/.*

*Apel, E., & UCAR/NCAR - Earth Observing Laboratory. (2011). Trace Organic Gas Analyzer (TOGA) for HIAPER. UCAR/NCAR - Earth Observing Laboratory. https://doi.org/10.5065/D6DF6P9Q Retrieved December 20, 2016*

*R. Volkamer, S. Baidar, T. L. Campos, S. Coburn, J. P. DiGangi, B. Dix, E. W. Eloranta, T. K. Koenig, B. Morley, I. Ortega, B. R. Pierce, M. Reeves, R. Sinreich, S. Wang, M. A. Zondlo, and P. A. Romashkin. Aircraft measurements of BrO, IO, glyoxal, NO 2 , H 2 O, O 2 –O 2 and aerosol extinction profiles in the tropics: comparison with aircraft-/ship-based in situ and lidar measurements. Atmospheric Measurement Techniques, 8(5):2121–2148, 2015. doi:10.5194/amt-8-2121-2015. URL http://www.atmos-meas-tech.net/8/2121/2015/.*

*S. J. Andrews, S. C. Hackenberg, and L. J. Carpenter. Technical Note: A fully automated purge and trap GC-MS system for quantification of volatile organic compound (VOC) fluxes between the ocean and atmosphere. Ocean Science, 11(2):313–321, 2015. doi:10.5194/os-11-313-2015. URL https://www.ocean-sci.net/11/313/2015/.*

**15) Page 4, first sentence of section 3.1: See comment 10, above.**
***Response****: See answer from comment 10 above.*

**16) Page 4, line 29: How is Ka parameterised?**
***Response:*** *Ka is parameterized following Johnson (2010) which is mainly a function of wind speed and sea surface temperature (SST) taken from the model at each time-step. Following Johnson (2010), Ka = ( 1/ka+KH/kw)$^{-1}$, where ka is the rate of air-side transfer, kw is the rate of water-side transfer, and KH is the dimensionless gas over-liquid form of Henry's law constant. ka is based on Jeffrey et al. (2010) and kw is based on Nightingale et al 2000. We do not feel it necessary to explain this is the text of the manuscript beyond the first sentence where we cite Johnson (2010).*

*Johnson, M. T.: A numerical scheme to calculate temperature and salinity dependent air-water transfer velocities for any gas. Ocean Science, 6(4):913–932, 2010. doi:10.5194/os-6-913-2010. URL http://www.ocean-sci.net/6/913/2010/.*

*Jeffery, C. D., Robinson, I. S., and Woolf, D. K.: Tuning a physically-based model of the air–sea gas transfer velocity, Ocean Model., 31, 28–35, doi:10.1016/j.ocemod.2009.09.001, 2010.*

*Nightingale, P. D., G. Malin, C. S. Law, A. J. Watson, P. S. Liss, M. I. Liddicoat, J. Boutin, and R. C. Upstill-Goddard (2000), In situ evaluation of air-sea gas exchange parameterizations using novel conservative and volatile tracers, Global Biogeochem. Cycles, 14(1), 373–387, doi:10.1029/1999GB900091.*

**17) Page 5, lines 6-8: Give examples for OVOC's emitted from the ocean.**
*Response:* *OVOCs emitted from the ocean are acetaldehyde, ethanol and methanol. Those three are included into the model. This is now clarified in the revised manuscript:*

*"For the three OVOCs (acetaldehyde (CH$_3$CHO), ethanol (C$_2$H$_6$O) and methanol (CH$_3$OH)), the same online approach for the VSLH is used to calculate the marine fluxes where their sea-water concentrations are taken from Yang et al. (2014)."*

**18) Page 5, line 8: Which VOC's were included?**
**"the model as follows, the same... (2014); emissions for alkenes.. "**
*Response:* *Acetaldehyde, ethanol, methanol, propane, propene, ethane and ethene are included into the model. This is now clarified in the revised manuscript:*

*"Oceanic fluxes of several VOCs have been included into the WRF-Chem as part if this study. For the three OVOCs (acetaldehyde (CH$_3$CHO), ethanol (C$_2$H$_6$O) and methanol (CH$_3$OH)), the same online approach for the VSLH is used to calculate the marine fluxes where their sea-water concentrations are taken from Yang et al. (2014).*
*Emissions for alkenes and alkanes (C$_2$H$_4$, C$_3$H$_6$, C$_2$H$_6$, C$_3$H$_8$) are prescribed and based on the POET (Granier et al., 2005) global inventory."*

**19) Page 5, lines 16-17: Give some details on sea-salt emission parameterisation.**
*Response:* *The sea-salt aerosol emission parameterization from Archer-Nicholls et al. (2014) is used in this study. This parameterization is mainly a function of wind speed from the model and uses the emissions scheme from Gong et al. (1997) for particles with dry diameters of 0.45nm or more and for smaller particles uses Fuentes et al. (2010). This information is now included in the revised manuscript as follows:*

*"The sea-salt aerosol emissions parameterization used in this study is described in Archer-Nicholls et al. (2014). This parameterization is mainly a function of wind speed from the model and uses the emissions scheme from Gong et al. (1997) for particles with dry diameters of 0.45nm or more and for smaller particles uses Fuentes et al. (2010)."*

*Archer-Nicholls, S., Lowe, D., Utembe, S., Allan, J., Zaveri, R. A., Fast, J. D., Hodnebrog, Ø., Denier van der Gon, H., and McFiggans, G.: Gaseous chemistry and aerosol mechanism developments for version 3.5.1 of the online regional model, WRF-Chem, Geosci. Model Dev., 7, 2557-2579, https://doi.org/10.5194/gmd-7-2557-2014, 2014.*

*Gong, S. L., Bartie, L. A., and Blanchet, J.-P.: Modeling sea-salt aerosols in the atmosphere 1. Model development, J. Geophys. Res., 102, 3805–3818, 1997.*

*Fuentes, E., Coe, H., Green, D., de Leeuw, G., and McFiggans, G.: On the impacts of phytoplankton-derived organic matter on the properties of the primary marine aerosol – Part*

*1: Source fluxes, Atmos. Chem. Phys., 10, 9295–9317, doi:10.5194/acp-10-9295- 2010, 2010.*

**20) Page 5, lines 21-25: This description is not quite clear. When 'Most of the reactions come from ... MISTRA', which ones?**
*Response: Inorganic, organic and inter-halogens reactions from Tables 2, 3 and 4 come from MISTRA. Moreover, the chemical loss of VSLH through oxidation by the hydroxyl radical (OH) and by photolysis was included using data from Sander et al. (2011b).*
*This information is now clarified in the revised manuscript.*

**How is the high uncertainty of I2Ox reactions dealt with?**
*Response: We have not tested the sensitivity of the results to the uncertainty of these reactions. We have included the state-of-the-art knowledge of these reactions (Saiz-Lopez et al., 2014).*

*A. Saiz-Lopez, R. P. Fernandez, C. Ordóñez, D. E. Kinnison, J. C. Gómez Martín, J.-F. Lamarque, and S. Tilmes. Iodine chemistry in the troposphere and its effect on ozone. Atmospheric Chemistry and Physics, 14(23):13119–13143, 2014. doi:10.5194/acp-14-13119-2014. URL http://www.atmos-chem-phys.net/14/13119/2014/.*

**21) Page 5, lines 31-33: Explain 17 bins and give a few details on FTUV. It would also be good to give e.g. noontime values of the photolysis frequencies (e.g. in Table 4).**
*Response: The Fast Tropospheric Ultraviolet-Visible model (FTUV) photolysis scheme is based on the Tropospheric Ultraviolet-Visible model (TUV) model developed by Madronich (1987) and uses a modified wavelength grid between 121–850 nm, where the number of spectral bins is reduced from 140 to 17 (Tie et al., 2004).*

*FTUV is a well-known and often used photolysis scheme. It is included as a standard option within WRF-Chem. We believe it is sufficient to refer to the paper Tie et al (2003) which describes the scheme in detail.*

*Below are the photolysis frequencies for a specific latitude and altitude at solar noon and under clear skies in response to the reviewer. However, these will vary throughout the model domain as a function of solar zenith angle, altitude and cloud cover so we feel that it would be misleading to include them in Table 4 of the paper.*

*Table 1. Photolysis reactions of halogens included in WRF-Chem.*

| Photolysis reactions | Photolysis rates ($s^{-1}$) |
|---|---|
| $Cl_2 \xrightarrow{h\nu} 2\ Cl$ | $2.5 \times 10^{-3}$ |
| $OClO\ (+O_2) \xrightarrow{h\nu} O_3 + ClO$ | $8.2 \times 10^{-2}$ |
| $HOCl \xrightarrow{h\nu} Cl + OH$ | $3.0 \times 10^{-4}$ |
| $ClNO_2 \xrightarrow{h\nu} Cl + NO_2$ | $5.7 \times 10^{-4}$ |
| $ClNO_3 \xrightarrow{h\nu} Cl + NO_3$ | $4.3 \times 10^{-5}$ |

| Reaction | Rate |
|---|---|
| $ClNO_3 \xrightarrow{h\nu} ClO + NO_2$ | $4.7 \times 10^{-4}$ |
| $Br_2 \xrightarrow{h\nu} 2\ Br$ | $3.1 \times 10^{-2}$ |
| $BrO \xrightarrow{h\nu} Br\ (+O_3)$ | $4.2 \times 10^{-2}$ |
| $HOBr \xrightarrow{h\nu} Br + OH$ | $2.2 \times 10^{-3}$ |
| $BrNO_2 \xrightarrow{h\nu} Br + NO_2$ | $2.6 \times 10^{-2}$ |
| $BrNO_3 \xrightarrow{h\nu} Br + NO_3$ | $3.8 \times 10^{-4}$ |
| $BrNO_3 \xrightarrow{h\nu} BrO + NO_2$ | $9.5 \times 10^{-4}$ |
| $I_2 \xrightarrow{h\nu} 2\ I$ | $1.6 \times 10^{-1}$ |
| $IO\ (+O_2) \xrightarrow{h\nu} I\ (+ O_3)$ | $1.5 \times 10^{-3}$ |
| $I_2O_4 \xrightarrow{h\nu} OIO + OIO$ | $7.3 \times 10^{-3}$ |
| $OIO \xrightarrow{h\nu} I\ (+ O2)$ | $2.0 \times 10^{-1}$ |
| $I_2O_2 \xrightarrow{h\nu} I + OIO$ | $2.8 \times 10^{-2}$ |
| $HOI \xrightarrow{h\nu} I + OH$ | $8.6 \times 10^{-3}$ |
| $INO \xrightarrow{h\nu} I + NO$ | $3 \times 10^{-2}$ |
| $INO_2 \xrightarrow{h\nu} I + NO_2$ | $3.3 \times 10^{-3}$ |
| $INO_3 \xrightarrow{h\nu} I + NO_3$ | $1.0 \times 10^{-2}$ |
| $I_2O_3 \xrightarrow{h\nu} OIO + IO$ | $3.0 \times 10^{-2}$ |
| $IBr \xrightarrow{h\nu} I + Br$ | $7.3 \times 10^{-2}$ |
| $ICl \xrightarrow{h\nu} I + Cl$ | $2.3 \times 10^{-2}$ |
| $BrCl \xrightarrow{h\nu} Br + Cl$ | $1.0 \times 10^{-2}$ |
| $CHBr_3\ (+O_2) \xrightarrow{h\nu} 3\ Br$ | $1.4 \times 10^{-6}$ |
| $CH_3Br \xrightarrow{h\nu} Br + CH_3O_2$ | $3.2 \times 10^{-22}$ |
| $CH_2Br_2 \xrightarrow{h\nu} 2Br$ | $1.6 \times 10^{-9}$ |
| $CH_2BrCl \xrightarrow{h\nu} Br + Cl$ | $7.3 \times 10^{-12}$ |
| $CHBrCl_2 \xrightarrow{h\nu} Br + 2\ Cl$ | $4.5 \times 10^{-7}$ |
| $CHBr_2Cl \xrightarrow{h\nu} 2Br + Cl$ | $2.7 \times 10^{-7}$ |
| $CH_2I_2 + (O_2) \xrightarrow{h\nu} 2\ I$ | $1.0 \times 10^{-2}$ |
| $CH_3I \xrightarrow{h\nu} I + CH_3O_2$ | $8.7 \times 10^{-6}$ |
| $CH_2ClI \xrightarrow{h\nu} I + Cl + 2\ HO_2 + CO$ | $1.8 \times 10^{-4}$ |
| $CH_2IBr \xrightarrow{h\nu} Br + I$ | $5.3 \times 10^{-4}$ |

*Tie, X., S. Madronich, S. Walters, R. Zhang, P. Racsh, and W. Collins (2003), Effect of clouds on photolysis and oxidants in the troposphere, J. Geophys. Res., 108, 4642, doi: , D20.*

**22) Page 6, Section 3.3, Equation (2) is simplistic in that it neglects diffusion limitations, which can be important at uptake coefficients higher than about 0.01, see e.g. Schwartz and Freiberg 1967. The authors should check likely changes in model results when more realistic parameterisations of heterogeneous reactions are used. Schwartz S.E. and Freiberg J.E. (1967), Mass-transport limitation to the rate of reac-**

tion of gases in liquid droplets: Application to oxidation of SO2 in aqueous solutions, Atmospheric Environment 15, (7),1129-1144

*Response: See response above.*

**23) Page 7, Lines 13-28: Are reactions R11 and R12 not occurring at pH > 5.5?**

*Response: Reactions R11-R16 occurs only when pH < 5.5 and reactions R17-R19 only when pH> 5.5.*

*This is clarified in the document "We then apply a pH dependence to the heterogeneous reactions that occur on the surface of the sea-salt. When the pH < 5.5 debromination reactions occur with the release of $Br_2$ and IBr resulting from the uptake of $BrNO_3$, $BrNO_2$, HOBr, $INO_3$, $INO_2$ and HOI (R11-R16). When the pH > 5.5 no debromination reactions occur, although uptake of $INO_3$, $INO_2$ and HOI on the sea-salt still occurs (R17-R19)"*

*What about mass conservation in R13?*

*Response: In this simplification we do not model the chemistry within the sea-salt explicitly so reactions R11-R19 represent just the changes to the gas phase species. As such mass may not be conserved and in particular for reaction R11-R16 they are intended to add an extra source of Br. The same approach is used in the global models TOMCAT (Yang et al., 2008) and CAM-Chem (Ordoñez et al., 2012).*

*Yang, X., J. A. Pyle, and R. A. Cox (2008), Sea salt aerosol production and bromine release: Role of snow on sea ice, Geophys. Res. Lett., 35, L16815, doi:10.1029/2008GL034536.*

*Ordóñez, C., Lamarque, J.-F., Tilmes, S., Kinnison, D. E., Atlas, E. L., Blake, D. R., Sousa Santos, G., Brasseur, G., and Saiz-Lopez, A.: Bromine and iodine chemistry in a global chemistry-climate model: description and evaluation of very short-lived oceanic sources, Atmos. Chem. Phys., 12, 1423-1447, https://doi.org/10.5194/acp-12-1423-2012, 2012.*

**Numbers or reactions are missing in Table 5**

*Response: No reactions are missing in Table 5.*

**24) Page 14, lines 15, 16: The model overestimates heterogeneous reactions on aerosol (see comment 22, above), thus the increase in the concentration of reactive species like BrO could be just a model effect.**

*Response: It is true that neglecting diffusion limitations will overestimate the impact of the heterogeneous reactions. As discussed above in response to the reviewer's earlier comment, there are many other uncertainties in the heterogeneous reactions. We have addressed this through sensitivity runs and revisions to the text described above.*

**25) Page 14, lines 19, 20: The underestimation of aldehydes could be due to neglect of Cl reactions in the model.**

*Response: The model chemical mechanism is of course a reduction of an explicit scheme, but it does include several reactions of Cl with VOCs that could lead to the formation of aldehydes including reactions of Cl with $CH_4$, $C_2H_6$, $C_3H_8$ and $C_3H_6$ (See Table 2 and Section 3.2.1). The problem may be a lack of VOCs in the model rather than specifically reactions of VOCs with Cl.*

**In fact heterogeneous oxidation of halogenides (by O3) is discussed in the text, but does not appear to be implemented in the model, since no reactions to that effect can be found in Table 5.**

*Response: We do not understand what the reviewer is referring to here. The heterogeneous oxidation of halogenides (by $O_3$) is not included. All the heterogeneous reactions discussed in the text are included in Table 5.*

**26) Page 14, line 22: What could be the reason for inorganic iodine emissions being lower than modelled in the tropics?**

*Response: The biggest uncertainty in the inorganic iodine emissions parameterization is the calculation of the iodide concentration in the sea water. This is now added in the revised manuscript. Other uncertainties in the flux calculation are in the Henry's law of HOI and suppression of flux from DOC (Shaw et al., 2013).*

*Shaw, M. and Carpenter, L. (2013). Modification of Ozone Deposition and $I_2$ Emissions at the Air-Aqueous Interface by Dissolved Organic Carbon of Marine Origin. Environmental science & technology. 47. 10.1021/es4011459.*

**27) Page 14, lines 23, 24: What is the reason for the minor importance of heterogeneous chemistry in the model?**

*Response: Unlike $Br_Y$, there is only a small change in the $I_Y$ partitioning with the inclusion of the heterogeneous chemistry: The main change in $I_Y$ with the inclusion of the heterogeneous chemistry occurs near the surface, due to the removal of the iodine oxides, and the production of more di-halogens in the MBL, especially when debromination is included.*
*Heterogenous iodine reactions (reactions R11-R19 from the manuscript) compete with the photolysis. Iodine species are more readily photolyzed (see Table 1), so less is taken up into the aerosol and the impact of heterogeneous chemistry is less.*

**28) Page 14, line 25: For which conditions was the 31% contribution to O3 loss calculated?**

*Response: In the first version of the manuscript, $O_X$ loss calculations presented in the manuscript were from our base simulation, the WRF-DEBROM simulation, where its conditions were described in Sec. 4.1 and Table 7:*
*"Our base simulation, WRF-DEBROM, considered all main processes involving halogen chemistry (sea-salt debromination, heterogeneous chemistry and reactions between halogens and VOCs) and computes the oceanic halocarbons fluxes online."*

*However, in order to quantify the uncertainty of $O_X$ loss calculations due to halogens, Fig. 12 have been updated and Fig. 14 has been included with calculations from other simulations. These results are discussed above in the general comment.*

**29) Page 14, line 26: How can the Ox loss be attributed to Br, I, Cl reactions when cross reactions play a major role (e.g. ClO + BrO)?**

*Response: Any cross reaction between XO species (where X is Br, Cl and I) will be counted as a loss of $O_X$ and will be attributed to each family involved. i.e. BrO + IO means that the Br family is responsible for the loss of one $O_X$ in the form of IO and at the same time the I family is responsible for the loss of one $O_X$ in the form of BrO. This is clarified in the document "The tropospheric $O_X$ loss due to the $BrO_X$, $IO_X$ and $ClO_X$ cycles".*

**31) Page 14, lines 27-29: What is the meaning of 'negative bias', is it an overestimation of O3 loss?**

*Response: 'Negative bias' means that the simulation with halogens (WRF-DEBROM) is underestimating the observed $O_3$ values in the MBL where the oceanic emissions of the halogenated species are higher (see Fig. 12 of the manuscript). The manuscript has been changed to "The simulation with halogens (WRF-DEBROM) is underestimating the observed $O_3$ values in the MBL, where the oceanic emissions of the halogenated species are higher."*

**32) Page 14, lines 30-35: the reviewer can only agree with the statement in Line 30. What is the effect of these uncertainties for the conclusions drawn like 'Halogens constitute 31% of the overall ... O3 loss'? Should the conclusion not rather be 'Halogens reactions constitute a considerable fraction (10-60%) of the overall tropospheric O3 loss'?**

*Response: See comment 28.*

**33) Caption of Figure 5: It would be more clear if "obs" was changed to "observation". In the time series of CH3I there is a clear anti-correlation between observation and model results during Feb. 5 – 10, can this be explained?**

*Response: Amended. The word "obs" has been changed to "observation" in all Figures.*

*Results for low $CH_3I$ during this period are already discussed in the manuscript in Sec. 5.1 "A possible explanation for the underestimation in halocarbon atmospheric concentrations might be due to the input data (e.g. wind speed, SST, sea-water concentration) that we used to compute these fluxes. In the case of the online fluxes, between 6-8 of February the model underestimates wind speed and this is directly accompanied by an underestimation for all three halocarbons atmospheric concentrations. "*

*F. Ziska, B. Quack, K. Abrahamsson, S. D. Archer, E. Atlas, T. Bell, J. H. Butler, L. J. Carpenter, C. E. Jones, N. R. P. Harris, H. Hepach, K. G. Heumann, C. Hughes, J. Kuss, K. Krüger, P. Liss, R. M. Moore, A. Orlikowska, S. Raimund, C. E. Reeves, W. Reifenhäuser, A. D. Robinson, C. Schall, T. Tanhua, S. Tegtmeier, S. Turner, L. Wang, D. Wallace, J. Williams, H. Yamamoto, S. Yvon-Lewis, and Y. Yokouchi. Global sea-to-air flux climatology for bromoform, dibromomethane and methyl iodide. Atmospheric Chemistry and Physics, 13(17):8915–8934, 2013. doi:10.5194/acp-13-8915-2013. URL http://www.atmos-chem-phys.net/13/8915/2013/.*

**34) Table 6: The N2O5 heterogeneous chemistry is not listed in Table 5.**

*Response: $N_2O_5$ heterogeneous chemistry was already included in the model version we were using, which is why it was not included in Table 5 which lists the chemistry added as part of the current study. The $N_2O_5$ chemistry is described in Archer-Nicholls et al., 2014. We have clarified this in the Table 5 heading:*

[revised manuscript text omitted]

P. A. Wales, R. J. Salawitch, J. M. Nicely, D. C. Anderson, T. P. Canty, S. Baidar, B. Dix, T. K. Koenig, R. Volkamer, D. Chen, L. G. Huey, D. J. Tanner, C. A. Cuevas, R. P. Fernandez, D. E. Kinnison, J.-F. Lamarque, A. Saiz-Lopez, E. L. Atlas, S. R. Hall, M. A. Navarro, L. L. Pan, S. M. Schauffler, M. Stell, S. Tilmes, K. Ullmann, A. J. Weinheimer, H. Akiyoshi, M. P. Chipperfield, M. Deushi, S. S. Dhomse, W. Feng, P. Graf, R. Hossaini, P. Jöckel, E. Mancini, M. Michou, O. Morgenstern, L. D. Oman, G. Pitari, D. A. Plummer, L. E. Revell, E. Rozanov, D. Saint-Martin, R. Schofield, A. Stenke, K. A. Stone, D. Visioni, Y. Yamashita, and G. Zeng. Stratospheric injection of brominated very short-lived substances: Aircraft observations in the western pacific and representation in global models. *Journal of Geophysical Research: Atmospheres*, 123(10):5690–5719, 2018. doi:10.1029/2017JD027978. URL https://agupubs.onlinelibrary.wiley.com/doi/abs/10.1029/2017JD027978.

S. Wang, J. A. Schmidt, S. Baidar, S. Coburn, B. Dix, T. K. Koenig, E. Apel, D. Bowdalo, T. L. Campos, E. Eloranta, M. J. Evans, J. P. DiGangi, M. A. Zondlo, R.-S. Gao, J. A. Haggerty, S. R. Hall, R. S. Hornbrook, D. Jacob, B. Morley, B. Pierce, M. Reeves, P. Romashkin, A. ter Schure, and R. Volkamer. Active and widespread halogen chemistry in the tropical and subtropical free troposphere. *Proceedings of the National Academy of Sciences*, 112(30):9281–9286, 2015. doi:10.1073/pnas.1505142112. URL http://www.pnas.org/content/112/30/9281.abstract.

5   M. Wesely. Parameterization of surface resistances to gaseous dry deposition in regional-scale numerical models . *Atmospheric Environment (1967)*, 23(6):1293 – 1304, 1989. ISSN 0004-6981. doi:http://dx.doi.org/10.1016/0004-6981(89)90153-4.

J. Williams, V. Gros, E. Atlas, K. Maciejczyk, A. Batsaikhan, H. F. Schöler, C. Forster, B. Quack, N. Yassaa, R. Sander, and R. Van Dingenen. Possible evidence for a connection between methyl iodide emissions and Saharan dust. *Journal of Geophysical Research: Atmospheres*, 112(D7):n/a–n/a, 2007. ISSN 2156-2202. doi:10.1029/2005JD006702. URL http://dx.doi.org/10.1029/2005JD006702. D07302.

10  M. Yang, R. Beale, P. Liss, M. Johnson, B. Blomquist, and P. Nightingale. Air-sea fluxes of oxygenated volatile organic compounds across the Atlantic Ocean. *Atmospheric Chemistry and Physics*, 14(14):7499–7517, 2014. doi:10.5194/acp-14-7499-2014. URL http://www.atmos-chem-phys.net/14/7499/2014/.

X. Yang, R. A. Cox, N. J. Warwick, J. A. Pyle, G. D. Carver, F. M. O'Connor, and N. H. Savage. Tropospheric bromine chemistry and its impacts on ozone: A model study. *Journal of Geophysical Research: Atmospheres*, 110(D23):n/a–n/a, 2005. ISSN 2156-2202. 15  doi:10.1029/2005JD006244. URL http://dx.doi.org/10.1029/2005JD006244. D23311.

R. A. Zaveri, R. C. Easter, J. D. Fast, and L. K. Peters. Model for Simulating Aerosol Interactions and Chemistry (MOSAIC). *Journal of Geophysical Research: Atmospheres*, 113(D13):n/a–n/a, 2008. ISSN 2156-2202. doi:10.1029/2007JD008782. URL http://dx.doi.org/10.1029/2007JD008782. D13204.

F. Ziska, B. Quack, K. Abrahamsson, S. D. Archer, E. Atlas, T. Bell, J. H. Butler, L. J. Carpenter, C. E. Jones, N. R. P. Harris, H. Hepach, 20  K. G. Heumann, C. Hughes, J. Kuss, K. Krüger, P. Liss, R. M. Moore, A. Orlikowska, S. Raimund, C. E. Reeves, W. Reifenhäuser, A. D. Robinson, C. Schall, T. Tanhua, S. Tegtmeier, S. Turner, L. Wang, D. Wallace, J. Williams, H. Yamamoto, S. Yvon-Lewis, and Y. Yokouchi. Global sea-to-air flux climatology for bromoform, dibromomethane and methyl iodide. *Atmospheric Chemistry and Physics*, 13(17):8915–8934, 2013. doi:10.5194/acp-13-8915-2013. URL http://www.atmos-chem-phys.net/13/8915/2013/.

---

## Author Response (AR2)

**Review of Alba Badia et al.: ', Co-Editor report, 03 Jan 2019 Minor Revision**

Review of "Importance of reactive halogens in the tropical marine atmosphere: A regional modelling study using WRF-Chem" by Alba Badia et al.

**The authors have addressed all of the reviewer comments, except for the comments by both reviewers regarding the mass transfer expression used. Given the importance of gas-aerosol interactions for halogen chemistry, the authors must address this point.**

**The authors should review the recent book from Brasseur and Jacob, Modeling of Atmospheric Chemistry, 2017, Section 5.5.2. The issue of mass transport limitation by gas diffusion is very well described in this section. Specifically, Brasseur and Jacob point out that "The free molecular regime generally applies to stratospheric aerosols. ... Tropospheric aerosols are often in the transition regime." This section then goes on to describe how to treat mass transfer using the expression of Schwartz (1986) for the transition regime, which is generally within 10% of the exact solution.**

The authors must demonstrate that they are in the free molecular regime in order to generally use their equation 2 throughout their model runs. The expression for the mass transfer rate coefficient in the free molecular regime, which the authors use, is given in Equation 5.117 (k_T in uptake rate expression given by 5.114) in Brasseur and Jacob. If the runs the authors present are not in the free molecular regime, then it is necessary to re-complete at least one model run with the more correct mass transfer rate coefficient, given in equation 5.121 in Brasseur and Jacob. The authors should discuss how the model results change (if they do) upon using the more correct expression. The authors argue that this will add additional computational cost in large scale models and as a result the mass transfer expression requires simplification in their runs. This is not convincing as the more correct expression will not add significant computational costs. However, I recognize that it is likely unfeasible to re-complete all the model runs for this study with the updated expression due to the large number of runs. This paper can be accepted if they can demonstrate how using the correct expression impacts the results in their case and discuss this briefly in the paper.

Even if other large scale model studies have used the the same expression (free molecular regime), we should not continue to generally use an expression that we know is incorrect for aerosol-gas interactions in the troposphere.

**Response:** We have reviewed Brasseur and Jacob and the reviewer is correct that our system is not in the free molecular range so strictly specking we should use a mass transfer rate coefficient that takes account of the diffusion limitation. We have therefore performed a new run (WRF-DIFF) where we have taken this into account and as we suspected the uncertainties introduced are relatively small. The results from this new run are actually very similar to those from the additional sensitivity run (WRF-GAMMADV2) that we did previously in response to the reviewer's comments on this point. Anyway as requested by the reviewer we have now completed "*at least one model run with the more correct mass transfer rate coefficient, given in equation 5.121 in Brasseur and Jacob*", have included the results from this run in the figures with the other model runs and have discussed "*how the model results change (if they do) upon using the more correct expression.*". We trust the paper can now be accepted for publication.

[revised manuscript text omitted]